# DELTA: DIVERSE CLIENT SAMPLING FOR FASTING FEDERATED LEARNING

## ABSTRACT

Partial client participation has been widely adopted in Federated Learning (FL) to efficiently reduce the communication burden. However, an improper client sampling scheme will select unrepresentative subsets, which will cause a large variance in the model update and slows down the convergence. Existing sampling methods are either biased or can be further improved to accelerate the convergence. In this paper, we propose an unbiased sampling scheme, termed DELTA, to alleviate this problem. In particular, DELTA characterizes the impact of client diversity and local variance and samples the representative clients who carry valuable information for global model updates. Moreover, DELTA is a provably optimal unbiased sampling scheme that minimizes the variance caused by partial client participation and achieves better convergence than other unbiased sampling schemes. We corroborate our results with experiments on both synthetic and real data sets.

## 1 INTRODUCTION

Federated Learning (FL) has recently emerged as a critical distributed learning paradigm where a number of clients collaborate with a central server to train a model. Edge clients finish the update locally without any data sharing, thus preserving client privacy. Communication can become the primary bottleneck of FL since edge devices have limited bandwidth and connection availability (Wang et al., 2021). In order to reduce the communication burden, only a portion of clients will be chosen for training in practice. However, an improper client sampling strategy, such as uniform client sampling adopted in FedAvg (McMahan et al., 2017), might exacerbate the issues of data heterogeneity in FL, as the randomly-selected unrepresentative subsets can increase the variance introduced by client sampling and directly slow down the convergence.

Existing sampling strategies can usually be categorized into two classes: biased and unbiased. Considering the crucial unbiased client sampling that may preserve the optimization objective, only a few strategies are proposed, e.g., in terms of multinomial distribution (MD) sampling and cluster sampling, including clustering based on sample size and clustering based on similarity methods. However, these sampling methods usually suffer from a slow convergence with large variance and computation overhead problems (Balakrishnan et al., 2021; Fraboni et al., 2021b).

To accelerate the convergence of FL with partial client participation, Importance Sampling (IS), another unbiased sampling strategy, is proposed in recent literature (Chen et al., 2020; Rizk et al., 2020). IS will select clients with the large gradient norm, as shown in Fig 1(a). As for another sampling method in Figure 1(a), cluster-based IS will first cluster the clients according to the gradient norm and then use IS to select the clients with a large gradient norm within each cluster.

Though IS, and cluster-based IS have their advantages, **1) IS suffers from learning inefficiency due to the transmission of excessive important yet similar updates from clients to the server**. This problem has been pointed out in recent works (Fraboni et al., 2021a; Shen et al., 2022), and some efforts are being conducted to solve this problem. One of them is cluster-based IS, which avoids redundant sampling of clients by first clustering similar clients into groups. Though clustering operation can somewhat alleviate this problem, **2) vanilla cluster-based IS does not work well because the high-dimensional gradient is too complicated to be a good clustering feature and can bring about poor clustering results, as pointed out by Shen et al. (2022). In addition, clustering is known to be susceptible to biased performance if the samples are chosen from a group that is clustered based on a biased opinion, as shown in Sharma (2017); Thompson (1990).** From the above discussion, we know though IS and cluster-based IS have their own advantages in

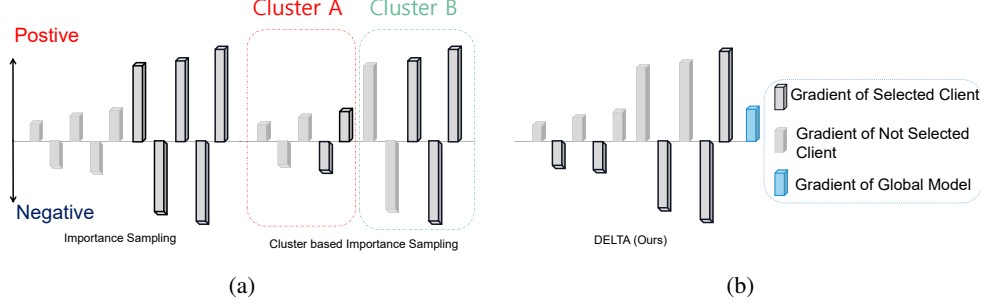

Figure 1: **Difference between IS. cluster-based IS, and our sampling scheme DELTA.**

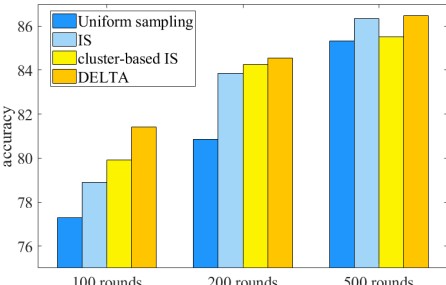

Figure 2: We use a logistic regression model to show the performance of different methods on non-iid MNIST. We sample 10 out of 200 clients and run 500 communication rounds. We report the average of the best 10 accuracies under 100, 300, and 500 rounds, which shows the accuracy performance from the initial training state to convergence.

sampling, they both face their own limitations as well. Specifically, IS has utilized the large gradient norm to accelerate convergence while meeting redundant sampling problems due to excessive similar updates, and cluster-based IS can alleviate the similar update problem but face a slow convergence due to poor clustering effect and biased performance. Figure 2 illustrates both these two sampling methods have times when they perform poorly.

To address the above challenges of IS and cluster-based IS, namely excessive similar updates and poor performance due to poor cluster effect and biased grouping, we propose a novel sampling method for Federated Learning, termed **D**iv**E**rse c**L**ien**T** s**A**mpling (DELTA). To simplify the notion, in this paper, we term FL with IS as FedIS. Compared with FedIS and cluster-based IS methods, we show in Figure 1(b) that DELTA tends to select clients with diverse gradient w.r.t global gradient. In this way, DELTA not only utilizes the advantages of a large gradient norm for convergence acceleration but also overcomes the gradient similarity issue.

## 1.1 CONTRIBUTIONS

In this paper, we propose an efficient unbiased sampling scheme based on gradient diversity and local variance, in the sense that (i) it can effectively solve the excessive similar gradient problem without additional clustering operation, while taking advantage of the accelerated convergence of gradient-norm-based IS and (ii) is provable better than uniform sampling or gradient norm based sampling. The sampling scheme is completely generic and can be easily compatible with other advanced optimization methods, like Fedprox (Li et al., 2018) and momentum (Karimireddy et al., 2020a).

As our key contributions,

- we present an unbiased sampling scheme for FL based on gradient diversity and local variance, a.k.a. DELTA. It can take advantage of the clients who select a large gradient norm and solve the problem of over-selection of clients with similar gradients at the beginning of training when that gradient of the global model is relatively large. Compared with the SOTA rate of FedAvg, its convergence rate removes the term $\mathcal{O}(1/T^{2/3})$ as well as a $\sigma_G^2$-related term in the numerator of $\mathcal{O}(1/T^{1/2})$.
- We provide theoretical proof of convergence for nonconvex FedIS. Unlike existing work, our analysis is based on a more relaxed assumption and yields no worse results than the existing convergence rates. Its rate removes the term $\mathcal{O}(1/T^{2/3})$ from that of FedAvg.

## 2 RELATED WORK

FedAvg is proposed by McMahan et al. (2017) as a de facto algorithm of FL, in which multiple local SGD steps are executed on the available clients to alleviate the communication bottleneck. While communication efficient, heterogeneity, such as system heterogeneity (Li et al., 2018; Wang et al.,

2020; Mitra et al., 2021; Diao et al., 2020), and statistical/objective heterogeneity (Lin et al., 2020; Karimireddy et al., 2020b; Li et al., 2018; Wang et al., 2020; Guo et al., 2021), results in inconsistent optimization objectives and drifted clients models, impeding federated optimization considerably.

**Objective inconsistency in FL.** Objective inconsistency is not rare in FL due to the heterogeneity of clients' data and the difference in computing ability. For instance, Wang et al. (2020) first identify an objective inconsistency caused by heterogeneous local updates. There also exist several works that encounter the difficulty from the objective inconsistency caused by partial client participation (Li et al., 2019; Cho et al., 2020; Balakrishnan et al., 2021). Li et al. (2019); Cho et al. (2020) use local-global gap $f^* - \frac{1}{m}\sum_{i=1}^{m} F_i^*$ to measure the distance between global optimum and average of all local personal optimum, where the local-global gap results from objective inconsistency at the final optimal point. In fact, objective inconsistency occurs in each training round, not only at the final optimal point. Balakrishnan et al. (2021) also encounter objective inconsistency caused by partial client participation. However, they use $\|\frac{1}{n}\sum_{i=1}^{n}\nabla F_i(x_t) - \nabla f(x_t)\| \leq \epsilon$ as an assumption to describe such update inconsistency caused by objective inconsistency without any analysis on it. So far, the objective inconsistency caused by partial client participation has not been analyzed though it is prevalent in FL, even in homogeneous local updates. Our work gives the fundamental convergence analysis on the influence of the objective inconsistency of partial client participation.

**Client selection in FL.** In general, the sampling method can be divided into biased and unbiased sampling. Note that unbiased sampling guarantees the same expected value of the client aggregation as the global deterministic aggregation with all clients' participation. In contrast, biased sampling will lead to converging to sub-optimal. The most famous unbiased sampling strategy in FL is multinomial sampling (MD), that samples according to client data ratio (Wang et al., 2020; Fraboni et al., 2021a). Besides, IS, an unbiased sampling method, is recently used in FL to reduce the convergence variance. Chen et al. (2020) uses update norm as importance to sampling clients, Rizk et al. (2020) samples clients based on data variability and Mohammed et al. (2021) uses test accuracy as an estimation of importance. Meanwhile, many biased sampling strategies have been proposed for accelerating training, such as sampling clients with higher loss (Cho et al., 2020), sampling clients as many as possible under the limitation of threshold (Qu et al., 2021), sampling clients with larger updates (Ribero & Vikalo, 2020) and greedy sampling according to gradient diversity (Balakrishnan et al., 2021). However, all these biased sampling methods can exacerbate the negative effects of objective inconsistency and promise to converge to only a neighbor of optimum. Recently, cluster-based client selection has draw some attentions in FL (Fraboni et al., 2021a; Xu et al., 2021; Muhammad et al., 2020; Shen et al., 2022). Though cluster operation needs additional clustering operation, and causes computation and memory overhead, Fraboni et al. (2021a); Shen et al. (2022) show clustering is helpful for sampling diverse clients and benefits for reducing variance. The proposed DELTA in Algorithm 1 can be viewed as a muted version of the diverse client clustering algorithm while promising to be unbiased.

## 3 THEORETICAL ANALYSIS AND AN IMPROVED FL SAMPLING STRATEGY

In FL, the objective of the global model is a sum-structured optimization problem:

$$f^* = \min_{x \in R^d}\left[f(x) := \sum_{i=1}^{m} w_i F_i(x)\right], \tag{1}$$

where $F_i(x) = \mathbb{E}_{\xi_i \sim D_i}[F_i(x, \xi_i)]$ represents the local objective function of client $i$ over data distribution $D_i$, and $\xi_i$ means the sampled data of client $i$. $m$ is the total number of clients and $w_i$ represents the weight of client $i$. With partial client participation, FedAvg (McMahan et al., 2017) randomly selects $|S_t| = n$ clients ($n \leq m$) to communicate and update model. Then the loss function of actual participating users in each round can be expressed as:

$$f_{S_t}(x_t) = \frac{1}{n}\sum_{i \in S_t} F_i(x_t). \tag{2}$$

To ease the theoretical analysis of our work, we use the following widely used assumptions.

### 3.1 ASSUMPTIONS

**Assumption 1** (L-Smooth). *The client's local objective function is Lipschitz smooth, i.e., there is a constant $L > 0$, such that $\|\nabla F_i(x) - \nabla F_i(y)\| \leq L\|x - y\|, \forall x, y \in \mathbb{R}^d$, and $i = 1, 2, \ldots, m$.*

Table 1: **Number of communication rounds required to reach $\epsilon$ or $\epsilon + \varphi$ ($\epsilon$ for unbiased sampling and $\epsilon + \varphi$ for biased sampling, where $\varphi$ is a non-convergent constant term) accuracy for FL.** $\sigma_L$ is local variance bound, and $\sigma_G$ bound is $E\|\nabla F_i(x) - \nabla f(x)\|^2 \leq \sigma_G^2$. $\Gamma$ is the distance of global optimum and the average of local optimum(Heterogeneity bound), $\mu$ corresponds to $\mu$ strongly convex. $G$ is the client's gradient bound, and $\zeta_G$ means the gradient diversity.

| Algorithm | Convexity | Partial Worker | Unbiased Sampling | Convergence rate | Assumption |
|---|---|---|---|---|---|
| SGD | Strongly/Nonconvex | ✓ | ✓ | $\frac{\sigma_L^2}{\mu m K \epsilon} + (\frac{1}{\mu}) / \frac{\sigma_L^2}{m K \epsilon^2} + \frac{1}{\epsilon}$ | $\sigma_L$ bound |
| **DELTA** | Nonconvex | ✓ | ✓ | $\frac{\sigma_L^2}{n K \epsilon^2} + \frac{\hat{M}^2}{K \epsilon}$ | Assumption 3 |
| **FedIS** (ours) | Nonconvex | ✓ | ✓ | $\frac{\sigma_L^2 + K \sigma_G^2}{\eta K \epsilon^2} + \frac{M^2}{K \epsilon}$ | Assumption 3 |
| FedIS (others) (Chen et al., 2020) | Nonconvex | ✓ | ✓ | $\frac{M^2}{n K \epsilon^2} + \frac{A^2+1}{\epsilon} + \frac{\sigma_G}{\epsilon^{3/2}}$ | Assumption 3 and $\rho$ bound |
| Yang et al. (2021) | Nonconvex | ✓ | ✓ | $\frac{\sigma_L^2}{n K \epsilon^2} + \frac{4 K \sigma_G^2}{n K \epsilon^2} + \frac{\hat{M}^2}{K \epsilon} + \frac{K^{1/3}\bar{M}^2}{n^{1/3}\epsilon^{2/3}}$ | $\sigma_G$ bound |
| Karimireddy et al. (2020b) | Nonconvex | ✓ | ✓ | $\frac{M^2}{n K \epsilon^2} + \frac{A^2+1}{\epsilon} + \frac{\sigma_G}{\epsilon^{3/2}}$ | Assumption 3 |
| Balakrishnan et al. (2021) | Strongly convex | ✓ | ✗ | $\frac{1}{\epsilon} + \frac{1}{\varphi}$ | Heterogeneity Gap |
| Cho et al. (2020) | Strongly convex | ✓ | ✗ | $\frac{\sigma_L^2 + G^2}{\epsilon + \varphi} + \frac{\Gamma}{\mu}$ | Heterogeneity Gap |
| Yang et al. (2021) | Nonconvex | ✗ | ✓ | $\frac{\sigma_L^2}{m K \epsilon^2} + \frac{\sigma_L^2/(4K)+\sigma_G^2}{\epsilon}$ | $\sigma_G$ bound |
| Karimireddy et al. (2020b) | Strongly Convex | ✗ | ✓ | $\frac{\sigma_L^2 + \sigma_G}{\mu m K \epsilon} + \frac{\sigma_L + \sigma_G}{\mu \sqrt{\epsilon}} + \frac{m(A^2+1)}{\mu}$ | Assumption 3 |

$M = \sigma_L^2 + 4K\sigma_G^2$, $\hat{M}^2 = \sigma_L^2 + K(1 - {}^n/m)\sigma_G^2$, $\bar{M}^2 = \sigma_L^2 + 6K\sigma_G^2$, $\acute{M}^2 = \sigma_L^2 + 4K\zeta_G^2$.

$\rho$ **assumption**: A bound of the similarity among local gradients in Chen et al. (2020) Another FedIS(others) (Chen et al., 2020) has the same convergence rate as Karimireddy et al. (2020b) under the $\rho$ assumption. While FedIS(ours) uses a looser Assumption 3 and achieves a faster rate than Chen et al. (2020).

## Algorithm 1 DELTA

**Require:** initial weights $x_0$, global learning rate $\eta$, local learning rate $\eta_l$, number of training rounds $T$
**Ensure:** trained weights $x_T$
1: **for** round $t = 1, \dots, T$ **do**
2:    Select a subset of clients according to the proposed sampling probability of **DELTA** (11)
3:    **for** each worker $i \in S_t$, in parallel **do**
4:      $x_{t,0}^i = x_t$
5:      **for** $k = 0, \cdots, K - 1$ **do**
6:        compute $g_{t,k}^i = \nabla F_i(x_{t,k}^i, \xi_{t,k}^i)$
7:        Local update:$x_{t,k+1}^i = x_{t,k}^i - \eta_L g_{t,k}^i$
8:      Let $\Delta_t^i = x_{t,K}^i - x_{t,0}^i = -\eta_L \sum_{k=0}^{K-1} g_{t,k}^i$
9:      Send gradient to server
10:    At Server:
11:    Receive $\Delta_t^i, i \in S_t$
12:    let $\Delta_t = \frac{1}{|S_t|} \sum_{i \in S_t} \frac{n_i}{n p_i^t} \Delta_t^i$
13:    Server update: $x_{t+1} = x_t + \eta \Delta_t$
14:    Broadcast $x_{t+1}$ to clients

**Assumption 2** (Unbiased Local Gradient Estimator and Local Variance). *let $\xi_t^i$ be a random local data sample in the round $t$ at client $i$: $\mathbb{E}\left[\nabla F_i(x_t, \xi_t^i)\right] = \nabla F_i(x_t), \forall i \in [m]$, where the expectation is over the local datasets sample. The function $F_i(x_t, \xi_t^i)$ has $\sigma_{L,i} > 0$ bounded local variance, i.e.,$\mathbb{E}\left[\left\|\nabla F_i(x_t, \xi_t^i) - \nabla F_i(x_t)\right\|^2\right] = \sigma_{L,i}^2 \leq \sigma_L^2$.*

**Assumption 3** (Bound Dissimilarity). *There exists constant $\sigma_G \geq 0$ and $A \geq 0$ s.t. $\mathbb{E}\|\nabla F_i(x)\|^2 \leq (A^2 + 1)\|\nabla f(x)\|^2 + \sigma_G^2$. When all local loss functions are identical, $A^2 = 0$ and $\sigma_G^2 = 0$.*

The above assumptions are commonly used in both non-convex optimization and FL literature, see e.g. Karimireddy et al. (2020b); Yang et al. (2021); Koloskova et al. (2020); Wang et al. (2020); Cho et al. (2020); Li et al. (2019). For Assumption 3, if all local loss functions are identical, then we have $A = 0$ and $\sigma_G = 0$.

### 3.2 CONVERGENCE RATE OF FEDIS

As discussed in the introduction, IS has an excessive gradient similarity problem, which may cause redundant sampling resulting in training inefficiency. As discussed in the introduction, IS has the issue of high gradient similarity, requiring us to design a new diversity sampling method. Before going to the details of our new sampling strategy, we first provide the convergence rate of FL under standard IS analysis in this section; the analysis itself is not well explored, especially for the nonconvex setting.

**Theorem 3.1** (Convergence rate of FedIS). *Under Assumptions 1–3, and sampling strategy FedIS $p_i^t = \frac{\|\hat{g_i^t}\|}{\sum_{j=1}^m \|\hat{g_j^t}\|}$, where $\hat{g_i^t} = \sum_{k=0}^{K-1} g_i^t = \sum_{k=0}^{K-1} \nabla F_i(x_{k,t}^i, \xi_{k,t}^i)$ is the sum of the gradient updates*

*of multiple local updates. Let constant local and global learning rates $\eta_L$ and $\eta$ be chosen as such that $\eta_L < min\left(1/(8LK), C\right)$, where $C$ is obtained from the condition that $\frac{1}{2} - 10L^2K^2(A^2+1)\eta_L^2 - \frac{L^2\eta K(A^2+1)}{2n}\eta_L > 0$, and $\eta \leq 1/(\eta_L L)$, the expected gradient norm will be bounded as follows:*

$$\min_{t \in [T]} E\|\nabla f(x_t)\|^2 \leq \mathcal{O}\left(\frac{f^0 - f^*}{\sqrt{nKT}}\right) + \underbrace{\mathcal{O}\left(\frac{\sigma_L^2}{\sqrt{nKT}}\right) + \mathcal{O}\left(\frac{M^2}{T}\right) + \mathcal{O}\left(\frac{K\sigma_G^2}{\sqrt{nKT}}\right)}_{order\ of\ \Phi}. \tag{3}$$

*where $f^0 = f(x_0)$, $f^* = f(x_*)$, $M = \sigma_L^2 + 4K\sigma_G^2$ and the expectation is over the local dataset samples among workers.*

The FedIS sampling probability $p_i^t = \frac{\|\hat{g}_i^t\|}{\sum_{j=1}^m \|\hat{g}_j^t\|}$ is derived from minimizing the variance of convergence w.r.t. $p_i^t$. The variance is

$$\Phi = \frac{5\eta_L^2 KL^2}{2}M^2 + \frac{\eta\eta_L L}{2m}\sigma_L^2 + \frac{L\eta\eta_L}{2nK}\text{Var}(\frac{1}{mp_i^t}\hat{g}_i^t), \tag{4}$$

where $\text{Var}(1/(mp_i^t)\hat{g}_i^t)$ is called *update variance*. The proof details of Theorem 3.1 and derivation of sampling probability FedIS are detailed in Appendix C and Appendix E.1.

**Remark 3.2.** *It is worth mentioning that although a few works provide the convergence upper bound for FedIS, several limitations exist in these analyses and results.*
*1) Rizk et al. (2020); Luo et al. (2022) applied IS in FL to solve a convex/strongly convex problem, while we solved a nonconvex problem.*
*2) In Rizk et al. (2020), their analysis result and sampling probability rely on the assumption of knowing the optimum $x_*$, which is not feasible in practice.*
*3) Our analysis uses the common Assumption 1–3, while Chen et al. (2020) provides the convergence rate of nonconvex FL under a stronger assumption of gradient similarity bound. Compared with Chen et al. (2020), we prove a tighter convergence upper bound for FedIS. Specifically, our convergence rate for FedIS improves from $\mathcal{O}(\frac{1}{\sqrt{nKT}} + \frac{1}{T} + \frac{1}{T^{2/3}})$ to $\mathcal{O}\left(\frac{1}{\sqrt{nKT}} + \frac{1}{T}\right)$ (c.f. Table 1).*

Despite the success of FedIS in reducing the variance term in the convergence rate, it is far from optimal, due to the issue of high gradient similarity and the improvement space of further minimizing the variance term (i.e., global variance $\sigma_G$ and local variance $\sigma_L$ in $\Phi$). We will discuss how to address this challenging variance term in the next section.

### 3.3 AN IMPROVED CONVERGENCE ANALYSIS

To ease the understanding of the theoretical difference between FedIS and DELTA, as well as a better illustration of our design choice, we include an analysis flowchart in Figure 3 to help understand the difference between FedIS and DELTA while strengthening the motivation. Specifically, based on the convergence variance of FedIS, we find it is important to reduce the variance beyond $\text{Var}(1/(mp_i^t)\hat{g}_i^t)$. Furthermore, we connect the important variance with the convergence of surrogate objective $\tilde{f}(x_t)$. Unlike FedIS, which analyzes the global objective, DELTA focuses on analyzing the surrogate objective and therefore obtains a different convergence variance and sampling probabilities than FedIS.

**The limitations of FedIS.** As identified by the Theorem 3.1 discussed above, IS suffers from excessive similar gradient selection. The variance $\Phi$ in (4) shows that the standard IS strategy can only control the update variance $\text{Var}(1/(mp_i^t)\hat{g}_i^t)$, while leaving other terms in $\Phi$ untouched, i.e., $\sigma_L$ and $\sigma_G$. Thus, the standard IS fails to handle the excessive similar gradient selection problem, and it motivates us to give a new sampling strategy below to address the issue of $\sigma_L$ and $\sigma_G$.

**The decomposition of the global objective.** As inspired by the proof of Theorem 3.1 as well as the corresponding Lemma B.1 (stated in Appendix) proposed for unbiased sampling, the global objective can be decomposed into surrogate objective and update gap,

$$\mathbb{E}\|\nabla f(x_t)\|^2 = \mathbb{E}\left\|\nabla \tilde{f}_{S_t}(x_t)\right\|^2 + \chi_t^2, \tag{5}$$

where $\chi_t = \mathbb{E}\left\|\nabla \tilde{f}_{S_t}(x_t) - \nabla f(x_t)\right\|$ is the update gap.

Intuitively, the surrogate objective is the practical objective of the participating clients in each round, while the update gap $\chi_t$ means the update distance between partial client participation and full client participation. The convergence behavior of the update gap $\chi_t^2$ corresponds to the update variance in $\Phi$, and the convergence of surrogate objective $\mathbb{E}\left\|\nabla \tilde{f}_{S_t}(x_t)\right\|^2$ is dependent on the other variance terms in $\Phi$, i.e., local variance and global variance.

Minimizing the surrogate objective allows us further to reduce the variance term in the convergence rate, and we focus on the convergence analysis of the surrogate objective below. For the purpose of analysis, we use IS property to formulate the surrogate objective with an arbitrary unbiased sampling probability.

**Surrogate objective formulation.** The expression of the surrogate objective relies on the property of IS. In detail, IS aims to substitute the original sampling distribution $p(z)$ with another arbitrary sampling distribution $q(z)$ while keeping the expectation unchanged: $\mathbb{E}_{q(z)}[F_i(z)] = \mathbb{E}_{p(z)}[q_i(z)/p_i(z)F_i(z)]$. According to the Monte Carlo method, when $q(z)$ follows the uniform distribution, we can estimate $\mathbb{E}_{q(z)}[F_i(z)]$ by $1/m \sum_{i=1}^{m} F_i(z)$ and $\mathbb{E}_{p(z)}[q_i(z)/p_i(z)F_i(z)]$ by $1/n \sum_{i \in S_t} 1/mp_i F_i(z)$, respectively, where $m$ and $|S_t| = n$ are sample sizes.

Based on the IS property, we formulate the surrogate objective as below:

$$\tilde{f}_{S_t}(x_t) = \tfrac{1}{n} \sum_{i \in S_t} \tfrac{1}{mp_i^t} F_i(x_t), \tag{6}$$

where $m$ is the total number of clients, $|S_t| = n$ is the number of participating clients in each round, and $p_t^i$ is the probability that client $i$ is selected at round $t$.

**An improved rate for the global objective.** Following the fact (c.f. Lemma B.2 in appendix) that[1]:

$$\min_{t \in [T]} \mathbb{E}\|\nabla f(x_t)\|^2 = \min_{t \in [T]} \mathbb{E}\|\nabla \tilde{f}(x_t)\|^2 + \mathbb{E}\|\chi_t^2\| \le \min_{t \in [T]} 2\mathbb{E}\|\nabla \tilde{f}(x_t)\|^2, \tag{7}$$

the convergence rate of the global objective can be formulated as follows:

**Theorem 3.3** (Convergence rate). *Under Assumption 1–3 and let local and global learning rates $\eta$ and $\eta_L$ satisfy $\eta_L < 1/(\sqrt{20K}L\sqrt{\tfrac{1}{n}\sum_{l=1}^{m}\tfrac{1}{mp_l^t}})$ and $\eta\eta_L \le 1/KL$, the minimal gradient norm will be bounded as below:*

$$\min_{t \in [T]} \mathbb{E}\left\|\nabla f(x_t)\right\|^2 \le \tfrac{f^0 - f^*}{c\eta\eta_L KT} + \tfrac{\tilde{\Phi}}{c}, \tag{8}$$

*where $f^0 = f(x_0)$, $f^* = f(x_*)$, $c$ is a constant, and the expectation is over the local dataset samples among all workers. The combination of variance $\tilde{\Phi}$ represents combinations of local variance and client gradient diversity.*

We derive the convergence rates for both sampling with replacement and sampling without replacement. For sampling without replacement:

$$\tilde{\Phi} = \tfrac{5L^2K\eta_L^2}{2mn} \sum_{i=1}^{m} \tfrac{1}{p_i^t}(\sigma_{L,i}^2 + 4K\zeta_{G,i}^2) + \tfrac{L\eta_L\eta}{2n} \sum_{i=1}^{m} \tfrac{1}{m^2p_i^t}\sigma_{L,i}^2, \tag{9}$$

For sampling with replacement,

$$\tilde{\Phi} = \tfrac{5L^2K\eta_L^2}{2m^2} \sum_{i=1}^{m} \tfrac{1}{p_i^t}(\sigma_{L,i}^2 + 4K\zeta_{G,i}^2) + \tfrac{L\eta_L\eta}{2n} \sum_{i=1}^{m} \tfrac{1}{m^2p_i^t}\sigma_{L,i}^2 \tag{10}$$

where $\zeta_{G,i} = \|\nabla F_i(x_t) - \nabla f(x_t)\|$ and let $\zeta_G$ be a upper bound for all $i$, i.e., $\zeta_{G,i} \le \zeta_G$. The proof details of Theorem 3.3 can be found in Appendix D.

## 3.4 OUR PROPOSED SAMPLING STRATEGY: DELTA

The update difference between the surrogate objective and the global objective can be defined as *objective inconsistency*. As demonstrated in Figure 4, different sampling methods lead to different degrees of objective inconsistency, and such inconsistency can be alleviated by choosing clients with a small updating gap. Figure 4(a) uses a toy example of square functions to illustrate the objective inconsistency when two out of three clients are selected for training, where DELTA would sample

---

[1]With a slight abuse of notations, we use the $\tilde{f}(x_t)$ for $\tilde{f}_{S_t}(x_t)$ in this paper.

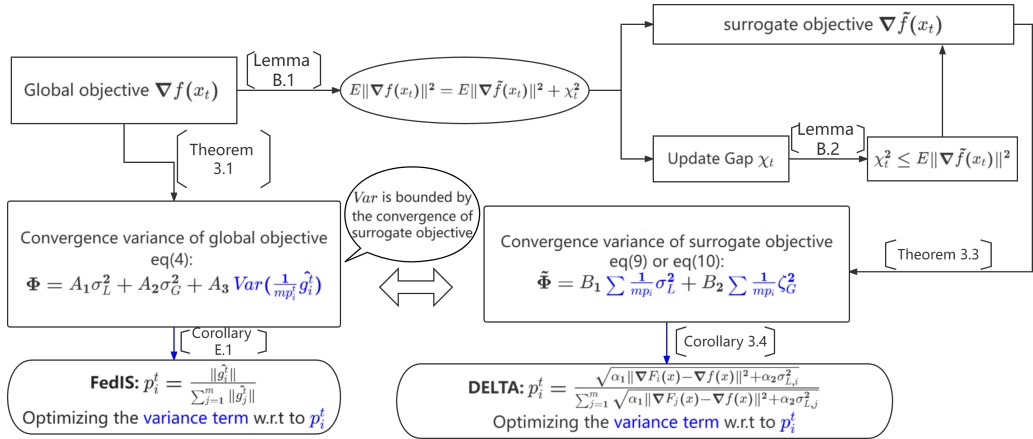

Figure 3: Sketch of theoretical analysis flow (Compared with FedIS). The left side represents the analysis flow of FedIS, while the analysis of DELTA is shown on the right. The sampling probability difference comes from the difference in variance.

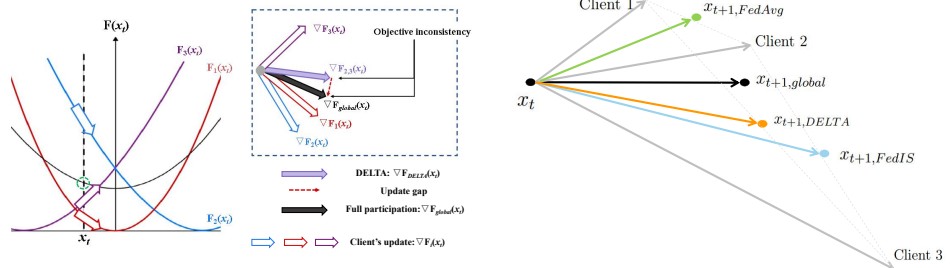

(a) Objective inconsistency and update gap.  (b) Illustration of different sampling methods.

Figure 4: **(a): Overview of objective inconsistency and update gap.** Here is three square functions with expression $y = 10x^2$ and $y = 3(x \pm 8)^2$, and gradient is calculated at $x = -2$. The detail enlargement shows the objective inconsistency. **(b): Illustration of the different sampling methods.** The client's update is shown by the grey arrow and the ideal global update is the black arrow. It shows our DELTA is better than FedIS and FedAvg.

diverse clients, leading to a small update gap. Figure 4(b) shows the one single round update process of different sampling schemes: IS tends to select client 2 and client 3 whose gradient norm is large, while diversity sampling DELTA tends to select client 1 and client 3. Therefore, compared with IS, the sampled clients of DELTA have a smaller bias from the global objective, illustrating a better sampling scheme of DELTA.

To derive our sampling strategy DELTA, it is equivalent to solving an optimization problem that minimizes the variance $\tilde{\Phi}$ w.r.t the proposed sampling probability $p_i^t$:

$$\min_{p_i^t} \tilde{\Phi} \quad \text{s.t.} \quad \sum_{i=1}^m p_i^t = 1 \,,$$

where $\tilde{\Phi}$ is a linear combination of local variance $\sigma_{L,i}$ and gradient diversity $\zeta_{G,i}$ (cf. Theorem 3.3).

**Corollary 3.4** (Optimal sampling probability for DELTA). *By solving the above optimization problem, the optimal sampling probability can be formulated as:*

$$p_i^t = \frac{\sqrt{\alpha_1 \|\nabla F_i(x) - \nabla f(x)\|^2 + \alpha_2 \sigma_{L,i}^2}}{\sum_{j=1}^m \sqrt{\alpha_1 \|\nabla F_j(x) - \nabla f(x)\|^2 + \alpha_2 \sigma_{L,j}^2}} \,, \tag{11}$$

*where $\alpha_1$ and $\alpha_2$ are constants defined as $\alpha_1 = 20K^2 L\eta_L$ and $\alpha_2 = 5KL\eta_L + \frac{\eta}{n}$.*

Let $\eta_L = \mathcal{O}\left(\frac{1}{\sqrt{TKL}}\right)$, $\eta = \mathcal{O}\left(\sqrt{Kn}\right)$ and substitute the optimal sampling probability (11) back to $\tilde{\Phi}$. Then for sufficiently large T, the iterates of Theorem 3.3 satisfy:

$$\min_{t \in [T]} \mathbb{E}\|\nabla f(x_t)\|^2 \leq \mathcal{O}\left(\frac{f^0 - f^*}{\sqrt{nKT}}\right) + \underbrace{\mathcal{O}\left(\frac{\sigma_L^2}{\sqrt{nKT}}\right) + \mathcal{O}\left(\frac{\sigma_L^2 + 4K\zeta_G^2}{KT}\right)}_{\text{order of } \tilde{\Phi}} . \tag{12}$$

### 3.5 DISCUSSIONS

**Difference between DELTA and FedIS.** The difference between DELTA and FedIS comes mainly from the difference between $\tilde{\Phi}$ and $\Phi$. FedIS aims to reduce the update variance term $\text{Var}(1/(mp_i^t)\hat{g}_i^t)$ in $\Phi$, while DELTA aims to reduce the whole $\tilde{\Phi}$ which is composed of the gradient diversity and the local variance. Minimizing $\tilde{\Phi}$ corresponds to further minimizing the terms of $\Phi$ that can not be minimized by FedIS. Solving different optimization problems leads to different sampling probability expressions. As shown in Figure 4, DELTA selects the more diverse Client 1 and Client 3 for participation, while FedIS tends to select Client 2 and Client 3 which have large gradient norms. It can be seen that the selection of DELTA leads to a smaller bias than FedIS. Moreover, as shown in Table 1, based on our convergence rate results, DELTA achieves a better convergence rate with $\mathcal{O}(G^2/\epsilon^2)$ higher than other unbiased sampling algorithms.

**Compare DELTA with uniform sampling.** According to the Cauchy-Schwarz inequality, DELTA is at least better than uniform sampling by reducing variance: $\frac{\tilde{\Phi}_{\text{uniform}}}{\tilde{\Phi}_{\text{DELTA}}} = \frac{m\sum_{i=1}^m\left(\sqrt{\alpha_1\sigma_L^2+\alpha_2\zeta_{G,i}^2}\right)^2}{\left(\sum_{i=1}^m\sqrt{\alpha_1\sigma_L^2+\alpha_2\zeta_{G,i}^2}\right)^2} \geq 1$.

This implies that DELTA does reduce the variance, especially when $\frac{\left(\sum_{i=1}^m\sqrt{\alpha_1\sigma_L^2+\alpha_2\zeta_{G,i}^2}\right)^2}{\sum_{i=1}^m\left(\sqrt{\alpha_1\sigma_L^2+\alpha_2\zeta_{G,i}^2}\right)^2} \ll m$.

**Remark 3.5.** *DELTA ensures the convergence of FL with partial client participation to a stationary point without any gap. Our results can be considered as a theoretical explanation for the heuristic of gradient diversity sampling algorithm in FL, and DELTA encourages the global model to acquire more knowledge in each round. Specifically, the server will give more weight to the clients with larger gradient diversity and local variance. These clients are representative, and sampling these clients can accelerate training given the more diverse and informative data to reflect the global data distribution. However, DELTA may fail to identify the attacked clients and even tends to select them when it comes to user attack scenarios. We will leave the solution for this scenario in our future work.*

## 4 PRACTICAL IMPLEMENTATION FOR DELTA AND FEDIS

Gradient-norm-based sampling method requires the computation of the full gradient in each iteration (Elvira & Martino, 2021; Zhao & Zhang, 2015). However, obtaining each client's gradient in advance is generally inadmissible in FL. For practical purposes, a series of IS algorithms estimate the current round's gradient by the historical gradient (Cho et al., 2020; Katharopoulos & Fleuret, 2017). Similarly, we utilize the gradient from the previous training iteration to estimate the gradient of the current round to reduce the computing resources (Rizk et al., 2020), where the previous iteration refers to the one in which the client participates.

In particular, at iteration 0, all probabilities are set to $1/m$, then during the $i_{th}$ iteration, after the participating clients $i \in S_t$ send the server their updated gradients, the sampling probabilities are updated as: $p_{i,t+1}^* = \frac{\|g_{\hat{i},t}\|}{\sum_{i\in S_t}\|g_{\hat{i},t}\|}(1 - \sum_{i\in S_t^c}p_{i,t}^*)$, where the multiplicative factor follows from ensuring all the probabilities sum to 1. Specifically, we use the average of the latest participated clients' gradients to approximate the true gradient of the global model for DELTA. In this way, it is not necessary to obtain all clients' gradients in each round. The convergence analysis of our practical algorithm is provided in Appendix F.

## 5 EXPERIMENTS

In this section, we use both synthetic dataset and split FEMNIST to demonstrate our theoretical results. To show the validity of the practical algorithm, we run experiments on FEMNIST and CIFAR-10, and show that DELTA converges faster and achieve higher accuracy than other baselines.

**Synthetic datasets.** We first examine our theoretical results through logistic regression on synthetic datasets. In details, we randomly generate $(x, y)$ by $y = log((Ax-b)^2/2)$ with given $A_i$ and $b_i$ as training data for clients, and each client's local dataset contain 1000 samples. In each round, 10 out of 20 clients are selected to participate in training (we also provide the results of 10 out of 200 clients in Appendix G). To simulate the gradient noise, in each training step, we calculate the gradient of client $i$ by $g_i = \nabla f_i(A_i, b_i, D_i) + \nu_i$, where $A_i$ and $b_i$ are model parameters, $D_i$ is the local dataset of client $i$, and $\nu_i$ is a zero-mean random variable which control the heterogeneity of client $i$. The larger the $\mathbb{E}\|\nu_i\|^2$, the larger the heterogeneity of client $i$.

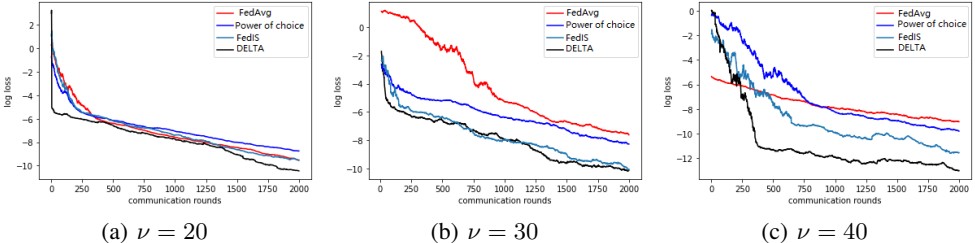

| (a) $\nu = 20$ | (b) $\nu = 30$ | (c) $\nu = 40$ |

Figure 5: **Performance of different algorithms on the regression model.** The loss is calculated by $f(x, y) = \left\| y - log^{((A_i x - b_i)^2 / 2)} \right\|^2$, $A = 10$, $b = 1$. We report the logarithm of global loss with different degrees of gradient noise $\nu$. All methods are well-tuned, and we report the best result of each algorithm under each setting.

Table 2: **Performance of algorithms.** We run 500 communication rounds on FEMNIST and CIFAR10 for each algorithm. We report the mean of maximum 5 accuracies for test datasets and the average number of communication rounds to reach the threshold accuracy.

| Algorithm | FEMNIST $\alpha = 0.1$ | | | CIFAR10 $\alpha = 0.5$ | | |
|---|---|---|---|---|---|---|
| | Acc (%) | Rounds for 70% | Time(s) for 70% | Acc (%) | Rounds for 54% | Time(s) for 54% |
| FedAvg (w/ uniform sampling) | $70.35 \pm 0.51$ | 426 (1.0×) | 1795.12 (1.0×) | $54.28 \pm 0.29$ | 338 (1.0×) | 3283.14 (1.0×) |
| Cluster-based IS | $71.21 \pm 0.24$ | 362 (1.17×) | 1547.41 (1.16×) | $54.83 \pm 0.02$ | 323 (1.05×) | 3188.54 (1.03×) |
| FedIS | $71.69 \pm 0.43$ | 404 (1.05×) | 1719.26 (1.04×) | $55.05 \pm 0.27$ | 313 (1.08×) | 3085.05 (1.06×) |
| DELTA | **$72.10 \pm 0.49$** | **322 (1.32×)** | **1372.33 (1.31×)** | **$55.20 \pm 0.26$** | **303 (1.12×)** | **2989.98 (1.1×)** |

Figure 5 demonstrates that these empirical results align with our theoretical analysis. Additional experiments of different functions and different settings, and the detailed sampling strategies of these different sampling algorithms can be found in Appendix G.

- **DELTA and FedIS outperform other biased and unbiased methods in convergence speed.** We can see both DELTA and FedIS converge faster than both FedAvg and Power-of-choice sampling. The larger the noise (variance), the more obvious the convergence speed advantage of DELTA and FedIS. For $\nu = 30$, FedIS can achieve near twice faster than FedAvg, and for $\nu = 40$, DELTA can achieve nearly $4\times$ times faster than FedAvg.
- **DELTA outperforms FedIS.** In experiments, DELTA converges about twice faster as FedIS in Figure 5(a). As all results show, DELTA can reduce more variance than FedIS and thus converge a smaller loss.

**Split FEMNIST** In this section, we consider the split FEMNIST. We let $10\%$ clients own $90\%$ data and the detailed split algorithm is provided in Appendix G. Figure 6 shows that when the data distribution is highly heterogeneous, Our DELTA algorithm converges faster than other baselines.

**FEMNIST and CIFAR-10.** We also verify our practical algorithm on FEMNIST and CIFAR-10. We summarize our numerical results in Table 2: Compared with other baselines, DELTA achieves higher accuracy and has an improvement in convergence rate both in terms of the number of iterations and the wall-clock time.

We also test different choices of the number of participated clients $n$ and test on different heterogeneity $\alpha$, and observe the consistent improvement of DELTA. The detailed setting and additional experiments are in Appendix G.

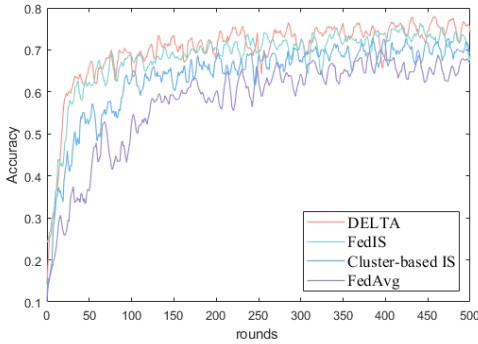

Figure 6: Performance of different sampling methods on Split FEMNIST dataset

# 6 CONCLUSION AND FUTURE WORK

In this work, we studied the optimal client sampling strategy that addresses the data heterogeneity to accelerate the convergence speed of FL. We obtain a new tractable convergence rate for nonconvex FL algorithms with arbitrary client sampling probabilities. Based on the bound, we solve an optimization problem with respect to sampling probability and thus develop a novel unbiased sampling scheme that characterizes the impact of client diversity and local variance on the sampling design. Experimental results validated the superiority of our theoretical and practical algorithms compared to several baselines.

As we point out, when user attacks occur, DELTA requires some changes to be able to identify and avoid selecting users from these attacks.

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

# A    TOY CASE

In Figure 7, we give a detailed toy case to show that DELTA is more effective than FedIS.

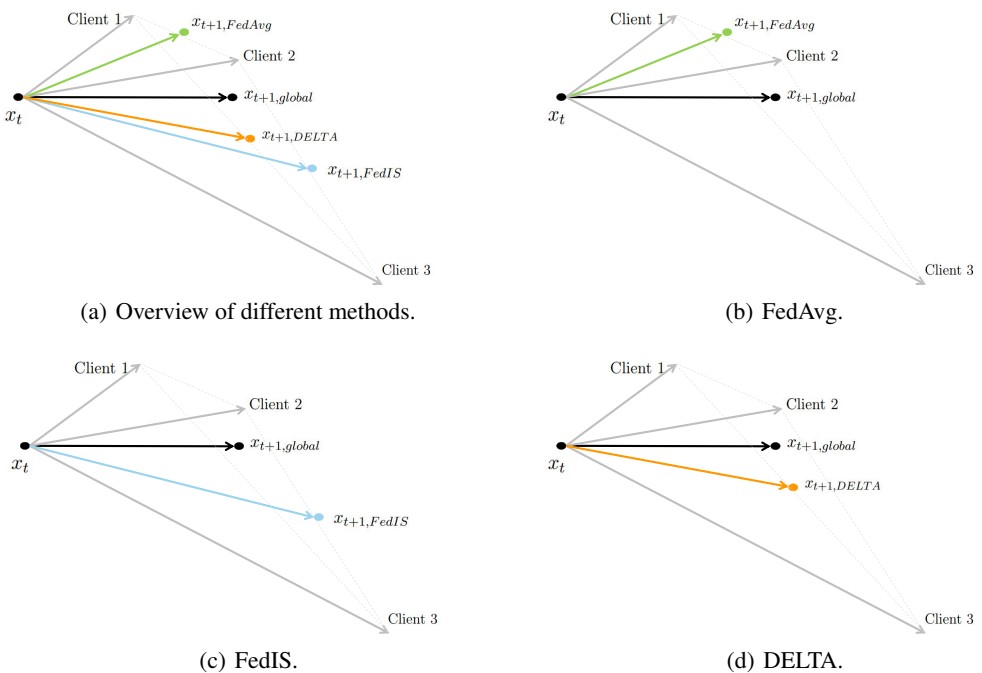

(a) Overview of different methods.

(b) FedAvg.

(c) FedIS.

(d) DELTA.

Figure 7: **Overview of objective inconsistency.** The intuition of objective inconsistency in FL is caused by client sampling. When Client 1 & 2, are selected to participate the training, then the model $x^{t+1}$ becomes $x_{FedAvg}^{t+1}$ instead of $x_{global}^{t+1}$, resulting in *objective inconsistency*. Different sampling strategies can cause different surrogate objectives, thus causing different biases. From Fig 7(a) we can see DELTA achieves minimal bias among the three unbiased sampling methods.

**Experiments for illustrating our observation.**    For the experiments to illustrate our observation of the introduction, we apply a logistic regression model on the non-iid MNIST dataset. 10 clients are selected from 200 clients to participate in training in each round. We set 2 cluster centers for cluster-based IS. And we set the mini batch-size to 32, the learning rate to 0.01, and the local update time to 5 for all methods. We run 500 communication rounds for each algorithm. We report the average of each round's selected clients' gradient norm and the minimum of each round's selected clients' gradient norm.

We report the gradient norm performance of cluster-based IS and IS to show that cluster-based IS selects clients with small gradients. As we mentioned in the introduction, the cluster-based IS always selects some clients from the cluster with small gradients, which will slow the convergence in some cases. We provide the average gradient norm comparison between IS and cluster-based IS in Figure 8(a). Besides, we also provide the minimal gradient norm comparison between IS and cluster-based IS in Figure 8(b).

We report the comparison of accuracy and loss performance between vanilla cluster-based IS and the removal of cluster-based IS with small gradient clusters. Specifically, we consider the setting with two cluster centers. And after 250 rounds, we replace the clients in the cluster containing the smaller gradient with the clients in the cluster containing the larger gradient while keeping the total number of the participated clients the same. The experiment result is shown in Figure 9. We can observe that the vanilla cluster-based IS performs worse than cluster-based IS without small gradients, which indicates that the small gradient is one reason for poor performance.

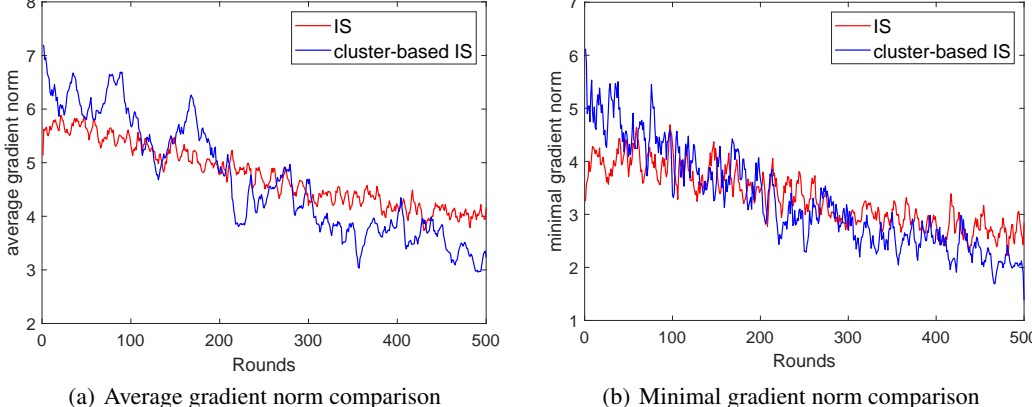

(a) Average gradient norm comparison      (b) Minimal gradient norm comparison

Figure 8: **The gradient norm comparison.** Both results indicate that cluster-based IS selects clients with small gradients after about half of the training rounds compared to IS.

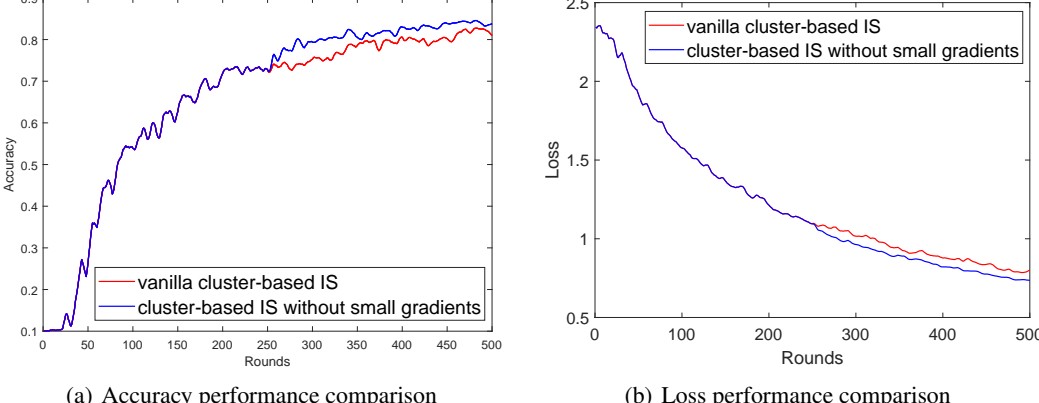

(a) Accuracy performance comparison      (b) Loss performance comparison

Figure 9: **An illustration that cluster-based IS sampling from the cluster with small gradients will slow convergence.** When the small gradient-norm cluster's clients are replaced by the clients from the large gradient-norm cluster, we see the performance improvement of cluster-based IS.

## B  TECHNIQUES

Here we show some technical lemmas which are helpful in the theoretical proof. We substitute $\frac{1}{m}$ for $\frac{n_i}{N}$ to simplify writing in all following proofs. $\frac{n_i}{N}$ is the data ratio of client $i$. All our proof can be easily extended from $f(x_t) = \frac{1}{m} \sum_{i=1}^m F_i(x_t)$ to $f(x_t) = \sum_{i=1}^m \frac{n_i}{N} F_i(x_t)$.

**Lemma B.1.** *(Unbiased Sampling). Importance sampling is unbiased sampling.* $\mathbb{E}(\frac{1}{n} \sum_{i \in S_t} \frac{1}{mp_i} \nabla F_i(x_t)) = \frac{1}{m} \sum_{i=1}^m \nabla F_i(x_t)$ *, no matter whether the sampling is with replacement or without replacement.*

Lemma B.1 proves that the importance sampling is an unbiased sampling strategy, either in sampling with replacement or sampling without replacement.

*Proof.* with replacement:

$$\mathbb{E}\left(\frac{1}{n} \sum_{i \in S_t} \frac{1}{mp_i^t} \nabla F_i(x_t)\right) = \frac{1}{n} \sum_{i \in S_t} \mathbb{E}\left(\frac{1}{mp_i^t} \nabla F_i(x_t)\right) = \frac{1}{n} \sum_{i \in S_t} \mathbb{E}\left(\mathbb{E}\left(\frac{1}{mp_i^t} \nabla F_i(x_t) \mid S\right)\right)$$

$$= \frac{1}{n} \sum_{i \in S_t} \mathbb{E}\left(\sum_{l=1}^m p_l^t \frac{1}{mp_l^t} \nabla F_l(x_t)\right) = \frac{1}{n} \sum_{i \in S_t} \nabla f(x_t) = \nabla f(x_t) \,,$$

$$(13)$$

without replacement:

$$
\begin{aligned}
\mathbb{E}\left(\frac{1}{n}\sum_{i\in S_t}\frac{1}{mp_i}\nabla F_i(x_t)\right) &= \frac{1}{n}\sum_{l=1}^{m}\mathbb{E}\left(\mathbb{I}_m\frac{1}{mp_l^t}\nabla F_l(x_t)\right) = \frac{1}{n}\sum_{l=1}^{m}\mathbb{E}(\mathbb{I}_m)\times\mathbb{E}(\frac{1}{mp_l^t}\nabla F_l(x_t)) \\
&= \frac{1}{n}\mathbb{E}(\sum_{l=1}^{m}\mathbb{I}_m)\times\mathbb{E}(\frac{1}{mp_l^t}\nabla F_l(x_t)) = \frac{1}{n}n\times\sum_{l=1}^{m}p_l^t\frac{1}{mp_l^t}\nabla F_l(x_t) \\
&= \frac{1}{n}\sum_{l=1}^{m}np_l^t\times\frac{1}{mp_l^t}\nabla F_l(x_t) = \frac{1}{m}\sum_{l=1}^{m}\nabla F_l(x_t) = \nabla f(x_t)\,, \quad (14)
\end{aligned}
$$

where $\mathbb{I}_m \triangleq \begin{cases} 1 & if\ x_l \in S_t\,, \\ 0 & otherwise\,. \end{cases}$

In the expectation, there are three origins of stochasticity. They are client sampling, local data SGD, and filtration of $x_t$. Therefore, the expectation is over all these randomnesses. Here, $S$ represents the origins of stochasticity except for client sampling. Rigorously, $S$ represents the filtration of the stochastic process $\{x_j, j=1,2,3\dots\}$ at time $t$ and the stochasticity of local SGD. $\qquad\square$

**Lemma B.2** (update gap bound).

$$
\chi^2 = \mathbb{E}\|\frac{1}{n}\sum_{i\in S_t}\frac{1}{mp_i^t}\nabla F_i(x_t) - \nabla f(x_t)\|^2 = \mathbb{E}\|\nabla\tilde{f}(x_t)\|^2 - \|\nabla f(x_t)\|^2 \le \mathbb{E}\|\nabla\tilde{f}(x_t)\|^2\,. \quad (15)
$$

*where the first equation follows from* $\mathbb{E}[x - \mathbb{E}(x)]^2 = \mathbb{E}[x^2] - [\mathbb{E}(x)]^2$ *and Lemma B.1.*

To increase readability, we give a detailed derivation of the Lemma B.2.

$$
\mathbb{E}\left(\|\nabla\tilde{f}(x_t) - \nabla f(x_t)\|^2 \mid S\right) = \mathbb{E}\left(\|\nabla\tilde{f}(x_t)\|^2 \mid S\right) - 2\mathbb{E}\left(\|\nabla\tilde{f}(x_t)\|\|\nabla f(x_t)\| \mid S\right) + \mathbb{E}\left(\|\nabla f(x_t)\|^2 \mid S\right)\,,
$$
$$(16)$$

where $\mathbb{E}(x \mid S)$ means the expectation on $x$ over the sampling space. And we use $\mathbb{E}\left(\|\nabla\tilde{f}(x_t) \mid S\right) = \nabla f(x_t)$ and $\mathbb{E}\left(\|\nabla f(x_t)\|^2 \mid S\right) = \|\nabla f(x_t)\|^2$ ($\|\nabla f(x)\|$ is a constant for stochasticity $S$ and the expectation over a constant is the constant itself.)
Therefore, we conclude

$$
\mathbb{E}\left(\|\nabla\tilde{f}(x_t) - \nabla f(x_t)\|^2 \mid S\right) = \mathbb{E}\left(\|\nabla\tilde{f}(x_t)\|^2 \mid S\right) - \|\nabla f(x_t)\|^2 \le \mathbb{E}\left(\|\nabla\tilde{f}(x_t)\|^2 \mid S\right)\,.
$$
$$(17)$$

We can further take the expectation on both sides of the inequality according to our needs, but without changing the relationship.

The following lemma follows from Lemma 4 of Reddi et al. (2020), but with a looser condition Assumption 3, instead of $\sigma_G^2$ bound. With some effort, we can derive the following lemma:

**Lemma B.3** (Local updates bound.). *For any step-size satisfying* $\eta_L \le \frac{1}{8LK}$, *we can have the following results:*

$$
\mathbb{E}\|x_{i,k}^t - x_t\|^2 \le 5K(\eta_L^2\sigma_L^2 + 4K\eta_L^2\sigma_G^2) + 20K^2(A^2+1)\eta_L^2\|\nabla f(x_t)\|^2\,. \quad (18)
$$

*Proof.*

$$\mathbb{E}_t \|x_{t,k}^i - x_t\|^2$$

$$= \mathbb{E}_t \|x_{t,k-1}^i - x_t - \eta_L g_{t,k-1}^t\|^2$$

$$= \mathbb{E}_t \|x_{t,k-1}^i - x_t - \eta_L (g_{t,k-1}^t - \nabla F_i(x_{t,k-1}^i) + \nabla F_i(x_{t,k-1}^i) - \nabla F_i(x_t) + \nabla F_i(x_t))\|^2$$

$$\leq (1 + \frac{1}{2K-1}) \mathbb{E}_t \|x_{t,k-1}^i - x_t\|^2 + \mathbb{E}_t \|\eta_L (g_{t,k-1}^t - \nabla F_i(x_{t,k}^i))\|^2$$

$$+ 4K \mathbb{E}_t [\|\eta_L (\nabla F_i(x_{t,K-1}^i) - \nabla F_i(x_t))\|^2] + 4K \eta_L^2 \mathbb{E}_t \|\nabla F_i(x_t)\|^2$$

$$\leq (1 + \frac{1}{2K-1}) \mathbb{E}_t \|x_{t,k-1}^i - x_t\|^2 + \eta_L^2 \sigma_L^2 + 4K \eta_L^2 L^2 \mathbb{E}_t \|x_{t,k-1}^i - x_t\|^2$$

$$+ 4K \eta_L^2 \sigma_{G,i}^2 + 4K \eta_L^2 (A^2 + 1) \|\nabla f(x_t)\|^2$$

$$\leq (1 + \frac{1}{K-1}) \mathbb{E} \|x_{t,k-1}^i - x_t\|^2 + \eta_L^2 \sigma_L^2 + 4K \eta_L^2 \sigma_G^2 + 4K (A^2 + 1) \|\eta_L \nabla f(x_t)\|^2. \quad (19)$$

$$(20)$$

Unrolling the recursion, we get:

$$\mathbb{E}_t \|x_{t,k}^i - x_t\|^2 \leq \sum_{p=0}^{k-1} (1 + \frac{1}{K-1})^p \left[ \eta_L^2 \sigma_L^2 + 4K \eta_L^2 \sigma_G^2 + 4K (A^2 + 1) \|\eta_L \nabla f(x_t)\|^2 \right] \quad (21)$$

$$\leq (K-1) \left[ (1 + \frac{1}{K-1})^K - 1 \right] \left[ \eta_L^2 \sigma_L^2 + 4K \eta_L^2 \sigma_G^2 + 4K (A^2 + 1) \|\eta_L \nabla f(x_t)\|^2 \right] \quad (22)$$

$$\leq 5K (\eta_L^2 \sigma_L^2 + 4K \eta_L^2 \sigma_G^2) + 20K^2 (A^2 + 1) \eta_L^2 \|\nabla f(x_t)\|^2. \quad (23)$$

$\square$

## C  CONVERGENCE OF FEDIS, PROOF OF THEOREM 3.1

We first restate the convergence theorem (Theorem 3.1) more formally, then prove the result for nonconvex case.

**Theorem C.1.** *Under Assumptions 1–3 and sampling strategy FedIS, the expected gradient norm will converge to a stationary point of the global objective. More specifically, if communication rounds T is pre-determined and the learning rate $\eta$ and $\eta_L$ is constant learning rates, then the expected gradient norm will be bounded as follows:*

$$\min_{t \in [T]} \mathbb{E} \|\nabla f(x_t)\|^2 \leq \frac{F}{c \eta \eta_L K T} + \Phi, \quad (24)$$

*where $F = f(x_0) - f(x_*)$, $M^2 = \sigma_L^2 + 4K \sigma_G^2$, and the expectation is over the local datasets samples among workers.*

*Let $\eta_L < \min(1/(8LK), C)$, where C is obtained from the condition that $\frac{1}{2} - 10L^2 K^2 (A^2 + 1) \eta_L^2 - \frac{L^2 \eta K (A^2+1)}{2n} \eta_L > 0$, and $\eta \leq 1/(\eta_L L)$, it then holds that:*

$$\Phi = \frac{1}{c} \left[ \frac{5\eta_L^2 L^2 K}{2m} \sum_{i=1}^{m} (\sigma_L^2 + 4K \sigma_G^2) + \frac{\eta \eta_L L}{2m} \sigma_L^2 + \frac{L \eta \eta_L}{2nK} V(\frac{1}{mp_i^t} \hat{g}_i^t) \right]. \quad (25)$$

*where c is a constant that satisfies $\frac{1}{2} - 10L^2 K^2 (A^2 + 1) \eta_L^2 - \frac{L^2 \eta K (A^2+1)}{2n} \eta_L > c > 0$, and $V(\frac{1}{mp_i^t} \hat{g}_i^t) = E \|\frac{1}{mp_i^t} \hat{g}_i^t - \frac{1}{m} \sum_{i=1}^{m} \hat{g}_i^t\|^2$.*

**Corollary C.2.** *Suppose $\eta_L$ and $\eta$ are such that the conditions mentioned above are satisfied, $\eta_L = \mathcal{O}\left(\frac{1}{\sqrt{T}KL}\right)$ and $\eta = \mathcal{O}\left(\sqrt{Kn}\right)$, and let the sampling probability be FedIS (82). Then for sufficiently large T, the iterates of Theorem 3.1 satisfy:*

$$\min_{t \in [T]} \mathbb{E} \|\nabla f(x_t)\|^2 = \mathcal{O}\left( \frac{\sigma_L^2}{\sqrt{nKT}} + \frac{K \sigma_G^2}{\sqrt{nKT}} + \frac{\sigma_L^2 + 4K \sigma_G^2}{KT} \right). \quad (26)$$

*Proof.*

$$\mathbb{E}_t[f(x_{t+1})] \overset{(a1)}{\leq} f(x_t) + \langle \nabla f(x_t), \mathbb{E}_t[x_{t+1} - x_t] \rangle + \frac{L}{2} \mathbb{E}_t[\|x_{t+1} - x_t\|^2]$$

$$= f(x_t) + \langle \nabla f(x_t), \mathbb{E}_t[\eta \Delta_t + \eta \eta_L K \nabla f(x_t) - \eta \eta_L K \nabla f(x_t)] \rangle + \frac{L}{2} \eta^2 \mathbb{E}_t[\|\Delta_t\|^2]$$

$$= f(x_t) - \eta \eta_L K \|\nabla f(x_t)\|^2 + \eta \underbrace{\langle \nabla f(x_t), \mathbb{E}_t[\Delta_t + \eta_L K \nabla f(x_t)] \rangle}_{A_1} + \frac{L}{2} \eta^2 \underbrace{\mathbb{E}_t \|\Delta_t\|^2}_{A_2},$$

$$\tag{27}$$

where (a1) follows from Lipschitz continuous condition. The expectation is conditioned over everything before current step $k$ of round $t$. Specifically, it is over clients' sampling, local data sampling, and the current round's model $x_t$.

Firstly we consider $A_1$:

$$A_1 = \langle \nabla f(x_t), \mathbb{E}_t[\Delta_t + \eta_L K \nabla f(x_t)] \rangle$$

$$= \left\langle \nabla f(x_t), \mathbb{E}_t[-\frac{1}{|S_t|} \sum_{i \in S_t} \frac{1}{mp_i^t} \sum_{k=0}^{K-1} \eta_L g_{t,k}^i + \eta_L \nabla f(x_t)] \right\rangle$$

$$\overset{(a2)}{=} \left\langle \nabla f(x_t), \mathbb{E}_t[-\frac{1}{m} \sum_{i=1}^{m} \sum_{k=0}^{K-1} \eta_L \nabla F_i(x_{t,k}^i) + \eta_L \nabla f(x_t)] \right\rangle$$

$$= \left\langle \sqrt{\eta_L K} \nabla f(x_t), -\frac{\sqrt{\eta_L}}{\sqrt{K}} \mathbb{E}_t[\frac{1}{m} \sum_{i=1}^{m} \sum_{k=0}^{K-1} (\nabla F_i(x_{t,k}^i) - \nabla F_i(x_t))] \right\rangle$$

$$\overset{(a3)}{=} \frac{\eta_L K}{2} \|\nabla f(x_t)\|^2 + \frac{\eta_L}{2K} \mathbb{E}_t \left\| \frac{1}{m} \sum_{i=1}^{m} \sum_{k=0}^{K-1} (\nabla F_i(x_{t,k}^i) - \nabla F_i(x_t)) \right\|^2$$

$$- \frac{\eta_L}{2K} \mathbb{E}_t \| \frac{1}{m} \sum_{i=1}^{m} \sum_{k=0}^{K-1} \nabla F_i(x_{t,k}^i) \|^2$$

$$\overset{(a4)}{\leq} \frac{\eta_L K}{2} \|\nabla f(x_t)\|^2 + \frac{\eta_L L^2}{2m} \sum_{i=1}^{m} \sum_{k=0}^{K-1} \mathbb{E}_t \left\| x_{t,k}^i - x_t \right\|^2 - \frac{\eta_L}{2K} \mathbb{E}_t \| \frac{1}{m} \sum_{i=1}^{m} \sum_{k=0}^{K-1} \nabla F_i(x_{t,k}^i) \|^2$$

$$\leq \left( \frac{\eta_L K}{2} + 10 K^3 L^2 \eta_L^3 (A^2 + 1) \right) \|\nabla f(x_t)\|^2 + \frac{5L^2 \eta_L^3}{2} K^2 \sigma_L^2 + 10 \eta_L^3 L^2 K^3 \sigma_G^2$$

$$- \frac{\eta_L}{2K} \mathbb{E}_t \| \frac{1}{m} \sum_{i=1}^{m} \sum_{k=0}^{K-1} \nabla F_i(x_{t,k}^i) \|^2, \tag{28}$$

where (a2) follows from Assumption 2 and LemmaB.1. (a3) is due to $\langle x, y \rangle = \frac{1}{2}\left[\|x\|^2 + \|y\|^2 - \|x - y\|^2\right]$ and (a4) comes from Assumption 1.

Next consider $A_2$. Let $\hat{g}_i^t = \sum_{k=0}^{K-1} g_{i,k}^t = \sum_{k=0}^{K-1} \nabla F_i(x_{t,k}^i, \xi_{t,k}^i)$

$$
\begin{aligned}
A_2 &= \mathbb{E}_t \|\Delta_t\|^2 \\
&= \mathbb{E}_t \left\| \eta_L \frac{1}{n} \sum_{i \in S_t} \frac{1}{mp_i^t} \sum_{k=0}^{K-1} g_{t,k}^i \right\|^2 \\
&= \eta_L^2 \frac{1}{n} \mathbb{E}_t \left\| \frac{1}{mp_i^t} \sum_{k=0}^{K-1} g_{t,k}^i - \frac{1}{m} \sum_{i=1}^{m} \sum_{k=0}^{K-1} g_{t,k}^i \right\|^2 \\
&+ \eta_L^2 \mathbb{E}_t \left\| \frac{1}{m} \sum_{i=1}^{m} \sum_{k=0}^{K-1} g_i(x_{t,k}^i) \right\|^2 \\
&= \frac{\eta_L^2}{n} V\left( \frac{1}{mp_i^t} \hat{g}_i^t \right) \\
&+ \eta_L^2 \mathbb{E} \| \frac{1}{m} \sum_{i=1}^{m} \sum_{k=0}^{K-1} [g_i(x_{t,k}^i) - \nabla F_i(x_{t,k}^i) + \nabla F_i(x_{t,k}^i)] \|^2 \\
&\leq \frac{\eta_L^2}{n} V\left( \frac{1}{mp_i} \hat{g}_i^t \right) \\
&+ \eta_L^2 \frac{1}{m^2} \sum_{i=1}^{m} \sum_{k=0}^{K-1} \mathbb{E} \|g_i(x_{t,k}^i) - \nabla F_i(x_{t,k}^i)\|^2 + \eta_L^2 \mathbb{E} \| \frac{1}{m} \sum_{i=1}^{m} \sum_{k=0}^{K-1} \nabla F_i(x_{t,k}^i) \|^2 \\
&\leq \frac{\eta_L^2}{n} V\left( \frac{1}{mp_i^t} \hat{g}_i^t \right) + \eta_L^2 \frac{K}{m} \sigma_L^2 + \eta_L^2 \mathbb{E} \| \frac{1}{m} \sum_{i=1}^{m} \sum_{k=0}^{K-1} \nabla F_i(x_{t,k}^i) \|^2 .
\end{aligned}
\tag{29}
$$

The third equality follows from independent sampling. Specifically, for sampling with replacement, due to every index being independent, we utilize $\mathbb{E} \|x_1^2 + ... + x_n\|^2 = \mathbb{E}[\|x_1\|^2 + ... + \|x_n\|^2]$.

For sampling without replacement:

$$
\mathbb{E} \| \frac{1}{n} \sum_{i \in S_t} \left( \frac{1}{mp_i^t} \hat{g}_i^t - \frac{1}{m} \sum_{i=1}^{m} \hat{g}_i^t \right) \|^2
\tag{30}
$$

$$
= \frac{1}{n^2} \mathbb{E} \| \sum_{i=1}^{m} \mathbb{I}_i \left( \frac{1}{mp_i^t} \hat{g}_i^t - \frac{1}{m} \sum_{i=1}^{m} \hat{g}_i^t \right) \|^2
\tag{31}
$$

$$
= \frac{1}{n^2} \mathbb{E} \left( \| \sum_{i=1}^{m} \mathbb{I}_i \left( \frac{1}{mp_i^t} \hat{g}_i^t - \frac{1}{m} \sum_{i=1}^{m} \hat{g}_i^t \right) \|^2 \mid \mathbb{I}_i = 1 \right) \times \mathbb{P}(\mathbb{I}_i = 1)
\tag{32}
$$

$$
+ \frac{1}{n^2} \mathbb{E} \left( \| \sum_{i=1}^{m} \mathbb{I}_i \left( \frac{1}{mp_i^t} \hat{g}_i^t - \frac{1}{m} \sum_{i=1}^{m} \hat{g}_i^t \right) \|^2 \mid \mathbb{I}_i = 0 \right) \times \mathbb{P}(\mathbb{I}_i = 0)
\tag{33}
$$

$$
= \frac{1}{n} \sum_{i=1}^{m} p_i^t \| \frac{1}{mp_i^t} \hat{g}_i^t - \frac{1}{m} \sum_{i=1}^{m} \hat{g}_i^t \|^2
\tag{34}
$$

$$
= \frac{1}{n} E \| \frac{1}{mp_i^t} \hat{g}_i^t - \frac{1}{m} \sum_{i=1}^{m} \hat{g}_i^t \|^2 .
\tag{35}
$$

From above, we observe that it is possible to gain a speedup by sampling from the distribution that minimizes $V(\frac{1}{mp_i^t} \hat{g}_i^t)$. Moreover, as we have discussed before, the optimal sampling probability is $p_i^* = \frac{\|\hat{g}_i^t\|}{\sum_{i=1}^{m} \|\hat{g}_i^t\|}$. For MD sampling (Li et al., 2019), which samples according to date ratio, the optimal sampling probability is $p_{i,t}^* = \frac{q_i \|\hat{g}_i^t\|}{\sum_{i=1}^{m} q_i \|\hat{g}_i^t\|}$, where $q_i = \frac{n_i}{N}$

Now substitute the expression of $A_1$ and $A_2$:

$$\mathbb{E}_t[f(x_{t+1})] \leq f(x_t) - \eta\eta_L K \|\nabla f(x_t)\|^2 + \eta \langle \nabla f(x_t), \mathbb{E}_t[\Delta_t + \eta_L K \nabla f(x_t)] \rangle + \frac{L}{2}\eta^2 \mathbb{E}_t\|\Delta_t\|^2$$

$$\leq f(x_t) - \eta\eta_L K \left(\frac{1}{2} - 10L^2 K^2 \eta_L^2 (A^2 + 1)\right)\|\nabla f(x_t)\|^2 + \frac{5\eta\eta_L^3 L^2 K^2}{2}(\sigma_L^2 + 4K\sigma_G^2)$$

$$+ \frac{\eta^2\eta_L^2 KL}{2m}\sigma_L^2 + \frac{L\eta^2\eta_L^2}{2n}V(\frac{1}{mp_i^t}\hat{g_i^t}) - \left(\frac{\eta\eta_L}{2K} - \frac{L\eta^2\eta_L^2}{2}\right)\mathbb{E}_t \left\|\frac{1}{m}\sum_{i=1}^{m}\sum_{k=0}^{K-1}\nabla F_i(x_{t,k}^i)\right\|^2$$

$$\leq f(x_t) - c\eta\eta_L K\|\nabla f(x_t)\|^2 + \frac{5\eta\eta_L^3 L^2 K^2}{2}(\sigma_L^2 + 4K\sigma_G^2) + \frac{\eta^2\eta_L^2 KL}{2m}\sigma_L^2 + \frac{L\eta^2\eta_L^2}{2n}V(\frac{1}{mp_i^t}\hat{g_i^t}),$$

$$(36)$$

where the last inequality follows from $\left(\frac{\eta\eta_L}{2K} - \frac{L\eta^2\eta_L^2}{2}\right) \geq 0$ if $\eta\eta_l \leq \frac{1}{KL}$, and (a9) holds because there exists a constant $c > 0$ (with some $\eta_L$) satisfying $\frac{1}{2} - 10L^2 \frac{1}{m}\sum_{i-1}^{m} K^2\eta_L^2(A^2 + 1) > c > 0$

Rearranging and summing from $t = 0, \ldots, T - 1$, we have:

$$\sum_{t=1}^{T-1} c\eta\eta_L K\mathbb{E}\|\nabla f(x_t)\|^2 \leq f(x_0) - f(x_T) + T(\eta\eta_L K)\Phi. \tag{37}$$

Which implies:

$$\min_{t \in [T]} \mathbb{E}\|\nabla f(x_t)\|^2 \leq \frac{f_0 - f_*}{c\eta\eta_L KT} + \Phi, \tag{38}$$

where

$$\Phi = \frac{1}{c}\left[\frac{5\eta_L^2 KL^2}{2}(\sigma_L^2 + 4K\sigma_G^2) + \frac{\eta\eta_L L}{2m}\sigma_L^2 + \frac{L\eta\eta_L}{2nK}V(\frac{1}{mp_i^t}\hat{g_i^t})\right]. \tag{39}$$

$\square$

### C.1 PROOF FOR CONVERGENCE OF FEDIS (THEOREM 3.1) UNDER ASSUMPTION 1–3.

For comparison, we first provide the convergence result under Assumption 4. The Assumption 4 is formally defined below:

**Assumption 4** (Gradient bound). *The stochastic gradient's expected squared norm is uniformly bounded, i.e.,* $E\|\nabla F_i(x_{t,k}, \xi_{k,t})\|^2 \leq G^2$ *for all $i$ and $k$.*

First we show Assumption 4 can be used to bound the update variance $V\left(\frac{1}{mp_i^t}\hat{g_i^t}\right)$, and under the sampling probability FedIS (80):

$$V\left(\frac{1}{mp_i^t}\hat{g_i^t}\right) \leq \frac{1}{m^2}\mathbb{E}\|\sum_{i=1}^{m}\sum_{k=1}^{K}\nabla F_i(x_{t,k}, \xi_{k,t})\|^2 \leq \frac{1}{m}\sum_{i=1}^{m} K\sum_{k=1}^{K}\mathbb{E}\|\nabla F_i(x_{t,k}, \xi_{k,t})\|^2 \leq K^2 G^2$$

$$(40)$$

While for using Assumption 3 instead of additional Assumption 4, we can also bound the update variance:

$$V\left(\frac{1}{mp_i^t}\hat{g_i^t}\right) \leq \frac{1}{m^2}\mathbb{E}\|\sum_{i=1}^{m}\sum_{k=1}^{K}\nabla F_i(x_{t,k}, \xi_{k,t})\|^2 \leq \frac{1}{m}\sum_{i=1}^{m} K\sum_{k=1}^{K}\mathbb{E}\|\nabla F_i(x_{t,k}, \xi_{k,t})\|^2$$

$$\leq K^2\sigma_G^2 + K^2(A^2 + 1)\|\nabla f(x_t)\|^2 \tag{41}$$

We replace the variance back to equation (36):

$$\mathbb{E}_t[f(x_{t+1})] \leq f(x_t) - \eta\eta_L K \|\nabla f(x_t)\|^2 + \eta \langle \nabla f(x_t), \mathbb{E}_t[\Delta_t + \eta_L K \nabla f(x_t)]\rangle + \frac{L}{2}\eta^2 \mathbb{E}_t\|\Delta_t\|^2$$

$$\leq f(x_t) - \eta\eta_L K \left(\frac{1}{2} - 10L^2K^2\eta_L^2(A^2+1)\right) \|\nabla f(x_t)\|^2 + \frac{5\eta\eta_L^3 L^2 K^2}{2}(\sigma_L^2 + 4K\sigma_G^2)$$

$$+ \frac{\eta^2\eta_L^2 KL}{2m}\sigma_L^2 + \frac{L\eta^2\eta_L^2}{2n}V(\frac{1}{mp_i^t}\hat{g}_i^t) - \left(\frac{\eta\eta_L}{2K} - \frac{L\eta^2\eta_L^2}{2}\right)\mathbb{E}_t\left\|\frac{1}{m}\sum_{i=1}^{m}\sum_{k=0}^{K-1}\nabla F_i(x_{t,k}^i)\right\|^2$$

$$\leq f(x_t) - \eta\eta_L K \left(\frac{1}{2} - 10L^2K^2\eta_L^2(A^2+1) - \frac{L\eta\eta_L K(A^2+1)}{2n}\right)\|\nabla f(x_t)\|^2 + \frac{5\eta\eta_L^3 L^2 K^2}{2}(\sigma_L^2 + 4K\sigma_G^2)$$

$$+ \frac{\eta^2\eta_L^2 KL}{2m}\sigma_L^2 + \frac{L\eta^2\eta_L^2}{2n}K^2\sigma_G^2 - \left(\frac{\eta\eta_L}{2K} - \frac{L\eta^2\eta_L^2}{2}\right)\mathbb{E}_t\left\|\frac{1}{m}\sum_{i=1}^{m}\sum_{k=0}^{K-1}\nabla F_i(x_{t,k}^i)\right\|^2$$

(42)

This shows that the requirement for $\eta_L$ is different. It needs that there exists a constant $c > 0$(with some $\eta_L$) satisfying $\frac{1}{2} - 10L^2K^2\eta_L^2(A^2+1) - \frac{L\eta\eta_L K(A^2+1)}{2n} > c > 0$. One can still guarantee that there exists a constant for $\eta_L$ to satisfy this inequality according to the properties of quadratic functions. Specifically, for quadratic equations $-10L^2K^2(A^2+1)\eta_L^2 - \frac{L\eta K(A^2+1)}{2n}\eta_L + \frac{1}{2}$, we know $-10L^2K^2(A^2+1) < 0$, $-\frac{L\eta K(A^2+1)}{2n}$ and $\frac{1}{2} > 0$. According to the solution of quadratic equations, we can make sure there exists a $\eta_L > 0$ solution.

Then we can substitute equation (36) by equation (42) and let $\eta_L = \mathcal{O}\left(\frac{1}{\sqrt{T}KL}\right)$ and $\eta = \mathcal{O}\left(\sqrt{Kn}\right)$, we get the convergence rate of FedIS under Assumption 1– 3:

$$\min_{t\in[T]} E\|\nabla f(x_t)\|^2 \leq \mathcal{O}\left(\frac{f^0 - f^*}{\sqrt{nKT}}\right) + \underbrace{\mathcal{O}\left(\frac{\sigma_L^2}{\sqrt{nKT}}\right) + \mathcal{O}\left(\frac{M^2}{T}\right) + \mathcal{O}\left(\frac{K\sigma_G^2}{\sqrt{nKT}}\right)}_{\text{order of }\Phi}. \qquad (43)$$

# D  CONVERGENCE OF DELTA. PROOF OF THEOREM 3.3

## D.1  CONVERGENCE RATE WITH IMPROVED ANALYSIS METHOD FOR GETTING DELTA

As we see FedIS can only reduce the update variance term in $\Phi$. Since we want to reduce the convergence variance as much as possible, the other term $\sigma_L$ and $\sigma_G$ still needs to be optimized. However, it is not straightforward to derive the optimization problem from $\Phi$. In order to further reduce the variance in $\Phi$ (cf. 4), i.e., local variance ($\sigma_L$) and global variance ($\sigma_G$), we divide the convergence of the global objective into a surrogate objective and an update gap, and analyze each term separately. The analysis framework is shown in Figure 10.

While for the update gap, as inspired by the expression form of update variance, we formally define it as below.

**Definition D.1** (Update gap). *In order to measure the update inconsistency, we define the update gap:*

$$\chi_t = \mathbb{E}\left[\left\|\nabla\tilde{f}(x_t) - \nabla f(x_t)\right\|\right]. \qquad (44)$$

*Here the expectation is over all clients' distribution. When full clients participate, we have $\chi_t^2 = 0$. The update inconsistency exists as long as partial client participation.*

The update gap is a direct embodiment of the objective inconsistency in the update process. The existence of update gap makes the analysis of global objective different from the analysis of surrogate objective. However, once we promise the convergence of the update gap, we can re-derive the convergence result for the global objective. Formally, the update gap can help us to connect global objective convergence and surrogate objective convergence as follows:

$$\mathbb{E}\|\nabla f(x_t)\|^2 = \mathbb{E}\|\nabla\tilde{f}(x_t)\|^2 + \chi_t^2. \qquad (45)$$

The equation follows from the property of unbiasedness, see Lemma B.1.

In order to deduce the convergence rate of the global objective, we start from the convergence analysis of the surrogate objective.

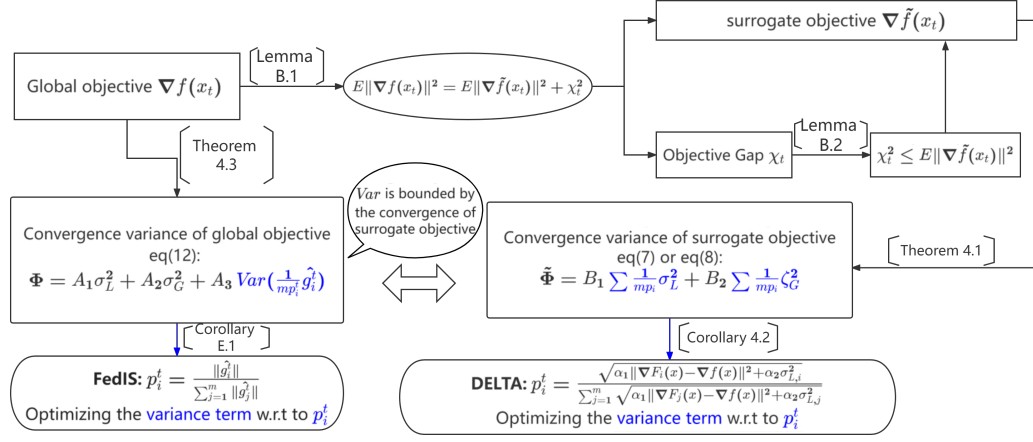

Figure 10: Sketch of theoretical analysis flow (Compared with FedIS). The left side represents the analysis flow of FedIS, while the analysis of DELTA is shown on the right. The sampling probability difference comes from the difference in variance.

**Theorem D.2** (Convergence rate of surrogate objective). *Under Assumption 1–3 and let local and global learning rates $\eta$ and $\eta_L$ satisfy $\eta_L < 1/(\sqrt{20K}L\sqrt{\frac{1}{n}\sum_{l=1}^{m}\frac{1}{mp_l^t}})$ and $\eta\eta_L \leq 1/KL$, the minimal gradient norm of surrogate objective will be bounded as below:*

$$\min_{t\in[T]}\mathbb{E}\left\|\nabla\tilde{f}(x_t)\right\|^2 \leq \frac{f^0-f^*}{\tilde{c}\eta\eta_L KT} + \frac{\tilde{\Phi}}{\tilde{c}}, \tag{46}$$

*where $f^0 = f(x_0)$, $f^* = f(x_*)$, the expectation is over the local dataset samples among workers.*

$\tilde{\Phi}$ **is the new combination of variance**, representing combinations of local variance and client gradient diversity.

For sample without replacement:

$$\tilde{\Phi} = \frac{5L^2K\eta_L^2}{2mn}\sum_{i=1}^{m}\frac{1}{p_i^t}(\sigma_{L,i}^2 + 4K\zeta_{G,i}^2) + \frac{L\eta_L\eta}{2n}\sum_{i=1}^{m}\frac{1}{m^2p_i^t}\sigma_{L,i}^2, \tag{47}$$

For sampling with replacement:

$$\tilde{\Phi} = \frac{5L^2K\eta_L^2}{2m^2}\sum_{i=1}^{m}\frac{1}{p_i^t}(\sigma_{L,i}^2 + 4K\zeta_{G,i}^2) + \frac{L\eta_L\eta}{2n}\sum_{i=1}^{m}\frac{1}{m^2p_i^t}\sigma_{L,i}^2 \tag{48}$$

where $\zeta_{G,i}$ represents client gradient diversity: $\zeta_{G,i} = \|\nabla F_i(x_t) - \nabla f(x_t)\|$, $\tilde{c}$ is a constant. The proof of Theorem D.2 is shown in Appendix D.2 and Appendix D.3. Specifically, the proof for sampling with replacement is shown in Appendix D.2 while the proof for sampling without replacement is shown in Appendix D.3.

**Remark D.3.** *We notice that there is no update variance in $\tilde{\Phi}$, but the local variance and global variance remain in it. Furthermore, the new combination of variance $\tilde{\Phi}$ can be minimized by optimizing w.r.t sampling probability, as is shown later.*

**Derive the convergence from surrogate objective to global objective.** As shown in Lemma B.1, unbiased sampling promises partial client updates in expectation are equal to the participation of all clients. With enough training rounds, unbiased sampling can guarantee that the update gap $\chi^2$ will converge to zero. However, we still need the convergence speed of $\chi_t^2$ to recover the convergence rate of the global objective. Fortunately, we can bound the convergence behavior of $\chi_t^2$ by the convergence rate of surrogate objective according to Definition D.1 and Lemma B.2. Therefore, the update gap can achieve at least the same convergence rate as the surrogate objective.

**Corollary D.4** (New convergence rate of global objective). *Under Assumption 1–3 and based on the above analysis that update variance is bounded, the global objective will converge to a stationary point. Its gradient is bounded as:*

$$\min_{t\in[T]}\mathbb{E}\|\nabla f(x_t)\|^2 = \min_{t\in[T]}\mathbb{E}\|\nabla\tilde{f}(x_t)\|^2 + \mathbb{E}\|\chi_t^2\| \leq \min_{t\in[T]}2\mathbb{E}\|\nabla\tilde{f}(x_t)\|^2 \leq \frac{f^0-f^*}{c\eta\eta_L KT} + \frac{\tilde{\Phi}}{c}. \tag{49}$$

**Theorem D.5** (Restate of Theorem 3.3). *Under Assumption 1-3 and the same conditions as theorem3.1, the minimal gradient norm of surrogate objective will be bounded as follows by setting $\eta_L = \frac{1}{\sqrt{T}KL}$ and $\eta\sqrt{Kn}$ Let local and global learning rates $\eta$ and $\eta_L$ satisfy $\eta_L < \frac{1}{\sqrt{20KL}\sqrt{\frac{1}{n}\sum_{l=1}^{m}\frac{1}{mp_l^t}}}$*

*and $\eta\eta_L \le \frac{1}{KL}$. Under Assumption 1-3 and with partial worker participation, the sequence of outputs $x_k$ generated by Algorithm 1 satisfies:*

$$\min_{t\in[T]}\mathbb{E}\|\nabla f(x_t)\|^2 \le \frac{F}{c\eta\eta_L KT} + \frac{1}{c}\tilde{\Phi},\tag{50}$$

*where $F = f(x_0) - f(x_*)$, and the expectation is over the local dataset samplings among workers. $c$ is a constant. $\zeta_{G,i}$ is defined as client gradient diversity: $\zeta_{G,i} = \|\nabla F_i(x_t) - \nabla f(x_t)\|$.*

*For sample with replacement: $\tilde{\Phi} = \frac{5L^2 K\eta_L^2}{2m^2}\sum_{l=1}^{m}\frac{1}{p_l^t}(\sigma_{L,l}^2 + 4K\zeta_{G,l}^2) + \frac{L\eta_L\eta}{2n}\sum_{l=1}^{m}\frac{1}{m^2 p_l^t}\sigma_{L,i}^2$.*

*For sampling without replacement: $\tilde{\Phi} = \frac{5L^2 K\eta_L^2}{2mn}\sum_{l=1}^{m}\frac{1}{p_l^t}(\sigma_{L,l}^2 + 4K\zeta_{G,l}^2) + \frac{L\eta_L\eta}{2n}\sum_{l=1}^{m}\frac{1}{m^2 p_l^t}\sigma_{L,l}^2$.*

**Remark D.6** (Condition of $\eta_L$). *Here, though the condition expression for $\eta_L$ relies on a dynamic sampling probability $p_l^t$, we can still guarantee that there a constant $\eta_L$ satisfies this condition.*

*Specifically, one can substitute the optimal sampling probability $\frac{1}{p_i^t} = \frac{\sum_{j=1}^{m}\sqrt{\alpha_1\zeta_{G,j}^2 + \alpha_2\sigma_{L,j}^2}}{\sqrt{\alpha_1\zeta_{G,i}^2 + \alpha_2\sigma_{L,i}^2}}$ back to the above inequality condition. As long as the gradient $\nabla F_i(x_t)$ is bounded, we can ensure $\frac{1}{m^2}\sum_{i=1}^{m}\frac{\sum_{j=1}^{m}\sqrt{\alpha_1\zeta_{G,j}^2 + \alpha_2\sigma_{L,j}^2}}{\sqrt{\alpha_1\zeta_{G,i}^2 + \alpha_2\sigma_{L,i}^2}} \le \frac{\max_j\sqrt{\alpha_1\zeta_{G,j}^2 + \alpha_2\sigma_{L,j}^2}}{\min_i\sqrt{\alpha_1\zeta_{G,i}^2 + \alpha_1\sigma_{L,i}^2}} \le \tilde{G}$, therefore $\frac{1}{\sqrt{20(A^2+1)}KL\sqrt{\frac{1}{m^2}\sum_{i=1}^{m}\frac{\sum_{j=1}^{m}\sqrt{\alpha_1\zeta_{G,j}^2 + \alpha_2\sigma_{L,j}^2}}{\sqrt{\alpha_1\zeta_{G,i}^2 + \alpha_2\sigma_{L,i}^2}}}} \ge \frac{1}{\sqrt{20(A^2+1)}KL\sqrt{\tilde{G}}} \ge C$, where $\tilde{G}$ and $C$ are positive constants. Thus, we can always find a constant $\eta_L$ to satisfy this inequality under dynamic sampling probability $p_i^t$.*

**Corollary D.7.** *Suppose $\eta_L$ and $\eta$ are such that the conditions mentioned above are satisfied, $\eta_L = \mathcal{O}\left(\frac{1}{\sqrt{T}KL}\right)$ and $\eta = \mathcal{O}\left(\sqrt{Kn}\right)$. Then for sufficiently large T, the iterates of Theorem 3.3 satisfy:*

$$\min_{t\in[T]}\mathbb{E}\|\nabla f(x_t)\|^2 \le \mathcal{O}\left(\frac{F}{\sqrt{nKT}}\right) + \mathcal{O}\left(\frac{\sigma_L^2}{\sqrt{nKT}}\right) + \mathcal{O}\left(\frac{\sigma_L^2 + 4K\zeta_G^2}{KT}\right).\tag{51}$$

**Lemma D.8.** *For any step-size satisfying $\eta_L \le \frac{1}{8LK}$, we can have the following results:*

$$\mathbb{E}\|x_{i,k}^t - x_t\|^2 \le 5K(\eta_L^2\sigma_L^2 + 4K\eta_L^2\zeta_{G,i}^2) + 20K^2(A^2+1)\eta_L^2\|\nabla f(x_t)\|^2.\tag{52}$$

*where $\zeta_{G,i} = \|\nabla F(x_t) - \nabla f(x_t)\|$, and the expectation is over local SGD and filtration of $x_t$, without the stochasticity of client sampling.*

*Proof.*

$$\mathbb{E}_t\|x_{t,k}^i - x_t\|^2$$
$$= \mathbb{E}_t\|x_{t,k-1}^i - x_t - \eta_L g_{t,k-1}^t\|^2$$
$$= \mathbb{E}_t\|x_{t,k-1}^i - x_t - \eta_L(g_{t,k-1}^t - \nabla F_i(x_{t,k-1}^i) + \nabla F_i(x_{t,k-1}^i) - \nabla F_i(x_t) + \nabla F_i(x_t))\|^2$$
$$\le (1 + \frac{1}{2K-1})\mathbb{E}_t\|x_{t,k-1}^i - x_t\|^2 + \mathbb{E}_t\|\eta_L(g_{t,k-1}^t - \nabla F_i(x_{t,k}^i))\|^2$$
$$+ 4K\mathbb{E}_t[\|\eta_L(\nabla F_i(x_{t,K-1}^i) - \nabla F_i(x_t))\|^2] + 4K\eta_L^2\mathbb{E}_t\|\nabla F_i(x_t)\|^2$$
$$\le (1 + \frac{1}{2K-1})\mathbb{E}_t\|x_{t,k-1}^i - x_t\|^2 + \eta_L^2\sigma_L^2 + 4K\eta_L^2 L^2\mathbb{E}_t\|x_{t,k-1}^i - x_t\|^2$$
$$+ 4K\eta_L^2\zeta_{G,i}^2 + 4K\eta_L^2(A^2+1)\|\nabla f(x_t)\|^2$$
$$\le (1 + \frac{1}{K-1})\mathbb{E}\|x_{t,k-1}^i - x_t\|^2 + \eta_L^2\sigma_L^2 + 4K\eta_L^2\sigma_G^2 + 4K(A^2+1)\|\eta_L\nabla f(x_t)\|^2.\tag{53}$$

$$\tag{54}$$

Unrolling the recursion, we get:

$$\mathbb{E}_t\|x_{t,k}^i - x_t\|^2 \leq \sum_{p=0}^{k-1}(1 + \frac{1}{K-1})^p \left[\eta_L^2\sigma_L^2 + 4K\eta_L^2\zeta_{G,i}^2 + 4K(A^2+1)\|\eta_L\nabla f(x_t)\|^2\right] \quad (55)$$

$$\leq (K-1)\left[(1+\frac{1}{K-1})^K - 1\right]\left[\eta_L^2\sigma_L^2 + 4K\eta_L^2\sigma_G^2 + 4K(A^2+1)\|\eta_L\nabla f(x_t)\|^2\right] \quad (56)$$

$$\leq 5K(\eta_L^2\sigma_L^2 + 4K\eta_L^2\sigma_G^2) + 20K^2(A^2+1)\eta_L^2\|\nabla f(x_t)\|^2 . \quad (57)$$

$$\square$$

**In Section D.2 and Section D.3, we provide the proof for Theorem D.2.** Specifically, the proof for sampling with replacement is shown in Appendix D.2 while the proof for sampling without replacement is shown in Appendix D.3.

### D.2 SAMPLE WITH REPLACEMENT

$$\min_{t\in[T]}\mathbb{E}\|\nabla\tilde{f}(x_t)\|^2 \leq \frac{f_0 - f_*}{c\eta\eta_L KT} + \frac{1}{c}\tilde{\Phi}, \quad (58)$$

where $\tilde{\Phi} = \frac{5L^2K\eta_L^2}{2m^2}\sum_{l=1}^m \frac{1}{p_l^t}(\sigma_L^2 + 4K\zeta_{G,i}^2) + \frac{L\eta_L\eta}{2n}\sum_{l=1}^m \frac{1}{m^2p_l^t}\sigma_L^2.$

*Proof.*

$$\tilde{f}(x_{t+1}) \overset{(a1)}{\leq} \tilde{f}(x_t) + \left\langle\nabla\tilde{f}(x_t), \mathbb{E}_t[x_{t+1} - x_t]\right\rangle + \frac{L}{2}\mathbb{E}_t[\|x_{t+1} - x_t\|^2]$$

$$= \tilde{f}(x_t) + \left\langle\nabla\tilde{f}(x_t), \mathbb{E}_t[\eta\Delta_t + \eta\eta_L K\nabla\tilde{f}(x_t) - \eta\eta_L K\nabla\tilde{f}(x_t)]\right\rangle + \frac{L}{2}\eta^2\mathbb{E}_t[\|\Delta_t\|^2]$$

$$= \tilde{f}(x_t) - \eta\eta_L K\left\|\nabla\tilde{f}(x_t)\right\|^2 + \eta\underbrace{\left\langle\nabla\tilde{f}(x_t), \mathbb{E}_t[\Delta_t + \eta_L K\nabla\tilde{f}(x_t)]\right\rangle}_{A_1} + \frac{L}{2}\eta^2\underbrace{\mathbb{E}_t\|\Delta_t\|^2}_{A_2} . \quad (59)$$

Where (a1) follows from Lipschitz continuous condition. Here the expectation is over local data SGD and filtration of $x_t$. However, in the next analysis, the expectation is over all randomness, i.e., client sampling is included.

Firstly consider $A_1$:

$$A_1 = \left\langle\nabla\tilde{f}(x_t), \mathbb{E}_t[\Delta_t + \eta_L K\nabla\tilde{f}(x_t)]\right\rangle$$

$$= \left\langle\nabla\tilde{f}(x_t), \mathbb{E}_t[-\frac{1}{|S_t|}\sum_{i\in S_t}\frac{1}{mp_i^t}\sum_{k=0}^{K-1}\eta_L g_{t,k}^i + \eta_L K\nabla\tilde{f}(x_t)]\right\rangle$$

$$\overset{(a2)}{=} \left\langle\nabla\tilde{f}(x_t), \mathbb{E}_t[-\frac{1}{|S_t|}\sum_{i\in S_t}\frac{1}{mp_i^t}\sum_{k=0}^{K-1}\eta_L\nabla F_i(x_{t,k}^i) + \eta_L K\nabla\tilde{f}(x_t)]\right\rangle$$

$$= \left\langle\sqrt{K\eta_L}\nabla\tilde{f}(x_t), \frac{\sqrt{\eta_L}}{\sqrt{K}}\mathbb{E}_t[-\frac{1}{n}\sum_{i\in S_t}\frac{1}{mp_i^t}\sum_{k=0}^{K-1}\nabla F_i(x_{t,k}^i) + K\nabla\tilde{f}(x_t)]\right\rangle$$

$$\overset{(a3)}{=} \frac{K\eta_L}{2}\|\nabla\tilde{f}(x_t)\|^2 + \frac{\eta_L}{2K}\mathbb{E}_t\left(\|-\frac{1}{n}\sum_{i\in S_t}\frac{1}{mp_i^t}\sum_{k=0}^{K-1}\nabla F_i(x_{t,k}^i) + K\nabla\tilde{f}(x_t)\|^2\right)$$

$$- \frac{\eta_L}{2K}\mathbb{E}_t\|-\frac{1}{n}\sum_{i\in S_t}\frac{1}{mp_i^t}\sum_{k=0}^{K-1}\nabla F_i(x_{t,k}^i)\|^2 , \quad (60)$$

where (a2) follows from Assumption 2, and (a3) is due to $\langle x, y \rangle = \frac{1}{2} \left[ \|x\|^2 + \|y\|^2 - \|x - y\|^2 \right]$ for $x = \sqrt{K\eta_L} \nabla \tilde{f}(x_t)$ and $y = \frac{\sqrt{\eta_L}}{K} [-\frac{1}{n} \sum_{i \in S_t} \frac{1}{mp_i^t} \sum_{k=0}^{K-1} \nabla F_i(x_{t,k}^i) + K\nabla \tilde{f}(x_t)]$.

In order to bound $A_1$, we need to bound the following part:

$$
\mathbb{E}_t \| \frac{1}{n} \sum_{i \in S_t} \frac{1}{mp_i^t} \sum_{k=0}^{K-1} \nabla F_i(x_{t,k}^i) - K\nabla \tilde{f}(x_t) \|^2
$$

$$
= \mathbb{E}_t \| \frac{1}{n} \sum_{i \in S_t} \frac{1}{mp_i^t} \sum_{k=0}^{K-1} \nabla F_i(x_{t,k}^i) - \frac{1}{n} \sum_{i \in S_t} \frac{1}{mp_i^t} \sum_{k=0}^{K-1} \nabla F_i(x_t) \|^2
$$

$$
\overset{(a4)}{\leq} \frac{K}{n} \sum_{i \in S_t} \sum_{k=0}^{K-1} \mathbb{E}_t \| \frac{1}{mp_i^t} (\nabla F_i(x_{t,k}^i) - \nabla F_i(x_t)) \|^2
$$

$$
= \frac{K}{n} \sum_{i \in S_t} \sum_{k=0}^{K-1} \mathbb{E}_t \{ \mathbb{E}_t ( \| \frac{1}{mp_i^t} (\nabla F_i(x_{t,k}^i) - \nabla F_i(x_t)) \|^2 \mid S) \}
$$

$$
= \frac{K}{n} \sum_{i \in S_t} \sum_{k=0}^{K-1} \mathbb{E}_t ( \sum_{l=1}^{m} \frac{1}{m^2 p_l^t} \| \nabla F_l(x_{t,k}^l) - \nabla F_l(x_t) \|^2 )
$$

$$
= K \sum_{k=0}^{K-1} \sum_{l=1}^{m} \frac{1}{m^2 p_l^t} \mathbb{E}_t \| \nabla F_l(x_{t,k}^l) - \nabla F_l(x_t) \|^2
$$

$$
\overset{(a5)}{\leq} \frac{K^2}{m^2} \sum_{l=1}^{m} \frac{L^2}{p_l^t} \mathbb{E} \| x_{t,k}^l - x_t \|^2
$$

$$
\overset{(a6)}{\leq} \frac{L^2 K^2}{m^2} \sum_{l=1}^{m} \frac{1}{p_l^t} \left( 5K(\eta_L^2 \sigma_L^2 + 4K\eta_L^2 \zeta_{G,i}^2) + 20K^2(A^2 + 1)\eta_L^2 \| \nabla f(x_t) \|^2 \right)
$$

$$
= \frac{5L^2 K^3 \eta_L^2}{m^2} \sum_{l=1}^{m} \frac{1}{p_l^t} (\sigma_L^2 + 4K\sigma_G^2) + \frac{20L^2 K^4 \eta_L^2 (A^2 + 1)}{m^2} \sum_{l=1}^{m} \frac{1}{p_l^t} \| \nabla f(x_t) \|^2 , \qquad (61)
$$

where (a4) follows from the fact that $\mathbb{E}\|x_1 + \cdots + x_n\|^2 \leq n\mathbb{E}\left( \|x_1\|^2 + \cdots + \|x_n\|^2 \right)$, (a5) is due to Assumption 1, and (a6) is due to Lemma D.8.

Combining the above formulations, we have:

$$
A_1 \leq \frac{K\eta_L}{2} \| \nabla \tilde{f}(x_t) \|^2 + \frac{\eta_L}{2K} \left[ \frac{5L^2 K^3 \eta_L^2}{m^2} \sum_{l=1}^{m} \frac{1}{p_l^t} (\sigma_L + 4K\zeta_{G,i}^2) + \frac{20L^2 K^4 \eta_L^2 (A^2 + 1)}{m^2} \sum_{l=1}^{m} \frac{1}{p_l^t} \| \nabla f(x_t) \|^2 \right]
$$

$$
- \frac{\eta_L}{2K} \mathbb{E}_t \| - \frac{1}{n} \sum_{i \in S_t} \frac{1}{mp_i^t} \sum_{k=0}^{K-1} \nabla F_i(x_{t,k}^i) \|^2 . \qquad (62)
$$

Next we consider to bound $A_2$:

$$A_2 = \mathbb{E}_t \|\Delta_t\|^2$$

$$= \mathbb{E}_t \left\| -\eta_L \frac{1}{n} \sum_{i \in S_t} \frac{1}{mp_i^t} \sum_{k=0}^{K-1} g_{t,k}^i \right\|^2$$

$$= \eta_L^2 \mathbb{E}_t \left\| \frac{1}{n} \sum_{i \in S_t} \sum_{k=0}^{K-1} (\frac{1}{mp_i^t} g_{t,k}^i - \frac{1}{mp_i^t} \nabla F_i(x_{t,k}^i)) \right\|^2 + \eta_L^2 \mathbb{E}_t \left\| -\frac{1}{n} \sum_{i \in S_t} \frac{1}{mp_i^t} \sum_{k=0}^{K-1} \nabla F_i(x_{t,k}^i) \right\|^2$$

$$= \eta_L^2 \frac{1}{n^2} \sum_{i \in S_t} \sum_{k=0}^{K-1} \mathbb{E}_t \left\| \frac{1}{mp_i^t} g_{t,k}^i - \frac{1}{mp_i^t} \nabla F_i(x_{t,k}^i) \right\|^2 + \eta_L^2 \mathbb{E}_t \left\| -\frac{1}{n} \sum_{i \in S_t} \frac{1}{mp_i^t} \sum_{k=0}^{K-1} \nabla F_i(x_{t,k}^i) \right\|^2$$

$$= \eta_L^2 \frac{1}{n^2} \sum_{k=0}^{K-1} \mathbb{E}_t \left( \mathbb{E} \left\| \frac{1}{mp_i^t} (g_{t,k}^i - \nabla F_i(x_{t,k}^i)) \right\|^2 \mid S \right) + \eta_L^2 \mathbb{E}_t \left\| -\frac{1}{n} \sum_{i \in S_t} \frac{1}{mp_i^t} \sum_{k=0}^{K-1} \nabla F_i(x_{t,k}^i) \right\|^2$$

$$= \eta_L^2 \frac{1}{n^2} \sum_{k=0}^{K-1} \mathbb{E}_t \left( \sum_{l=1}^{m} \frac{1}{m^2 p_l^t} \left\| g_{t,k}^i - \nabla F_i(x_{t,k}^i) \right\|^2 \right) + \eta_L^2 \mathbb{E}_t \left\| -\frac{1}{n} \sum_{i \in S_t} \frac{1}{mp_i^t} \sum_{k=0}^{K-1} \nabla F_i(x_{t,k}^i) \right\|^2$$

$$\overset{(a7)}{\leq} \eta_L^2 \frac{K}{n} \sum_{l=1}^{m} \frac{1}{m^2 p_l^t} \sigma_L^2 + \eta_L^2 \mathbb{E}_t \left\| -\frac{1}{n} \sum_{i \in S_t} \frac{1}{mp_i^t} \sum_{k=0}^{K-1} \nabla F_i(x_{t,k}^i) \right\|^2, \tag{63}$$

where $S$ represents the whole sample space and (a7) is due to Assumption 2.

Now substitute the expression of $A_1$ and $A_2$ and take the expectation over client sampling distribution on both sides. It should be noted that the derivation of $A_1$ and $A_2$ in above is based on considering the expectation over sampling distribution:

$$f(x_{t+1}) \leq f(x_t) - \eta\eta_L K \mathbb{E}_t \left\| \nabla \tilde{f}(x_t) \right\|^2 + \eta \mathbb{E}_t \left\langle \nabla \tilde{f}(x_t), \Delta_t + \eta_L K \nabla \tilde{f}(x_t) \right\rangle + \frac{L}{2} \eta^2 \mathbb{E}_t \|\Delta_t\|^2$$

$$\leq f(x_t) - K\eta\eta_L \left( \frac{1}{2} - \frac{10K^2 \eta_L^2 L^2 (A^2+1)}{m^2} \sum_{l=1}^{m} \frac{1}{p_l^t} \right) \mathbb{E}_t \left\| \nabla \tilde{f}(x_t) \right\|^2 + \frac{5L^2 K^2 \eta_L^3 \eta}{2m^2} \sum_{l=1}^{m} \frac{1}{p_l^t} \left( \sigma_L + 4K\zeta_{G,i}^2 \right) \tag{64}$$

$$+ \frac{L\eta_L^2 \eta^2 K}{2n} \sum_{l=1}^{m} \frac{1}{m^2 p_l^t} \sigma_L^2 - \left( \frac{\eta\eta_L}{2K} - \frac{L\eta^2 \eta_L^2}{2} \right) \mathbb{E}_t \left\| -\frac{1}{n} \sum_{i \in S_t} \frac{1}{mp_i^t} \sum_{k=0}^{K-1} \nabla f_i(x_{t,k}^i) \right\|^2$$

$$\overset{(a8)}{\leq} f(x_t) - K\eta\eta_L \left( \frac{1}{2} - \frac{10K^2 \eta_L^2 L^2 (A^2+1)}{m^2} \sum_{l=1}^{m} \frac{1}{p_l^t} \right) \mathbb{E}_t \|\nabla \tilde{f}(x_t)\|^2 + \frac{5L^2 K^2 \eta_L^3 \eta}{2m^2} \sum_{l=1}^{m} \frac{1}{p_l^t} (\sigma_L + 4K\zeta_{G,i}^2) \tag{65}$$

$$+ \frac{L\eta_L^2 \eta^2 K}{2n} \sum_{l=1}^{m} \frac{1}{m^2 p_l^t} \sigma_L^2$$

$$\overset{(a9)}{\leq} f(x_t) - cK\eta\eta_L \mathbb{E}_t \|\nabla \tilde{f}(x_t)\|^2 + \frac{5L^2 K^2 \eta_L^3 \eta}{2m^2} \sum_{l=1}^{m} \frac{1}{p_l^t} (\sigma_L^2 + 4K\zeta_{G,i}^2) + \frac{L\eta_L^2 \eta^2 K}{2n} \sum_{l=1}^{m} \frac{1}{m^2 p_l^t} \sigma_L^2, \tag{66}$$

where (a8) follows from $\left( \frac{\eta\eta_L}{2K} - \frac{L\eta^2 \eta_L^2}{2} \right) \geq 0$ if $\eta\eta_l \leq \frac{1}{KL}$, and (a9) holds because there exists a constant $c > 0$ satisfying $(\frac{1}{2} - \frac{10K^2 \eta_L^2 L^2 (A^2+1)}{m^2} \sum_{l=1}^{m} \frac{1}{p_l^t}) > c > 0$ if $\eta_L < \frac{1}{\sqrt{20(A^2+1)}KL\sqrt{\frac{1}{m} \sum_{l=1}^{m} \frac{1}{mp_l^t}}}$.

Rearranging and summing from $t = 0, \ldots, T-1$, we have:

$$\sum_{t=1}^{T-1} c\eta\eta_L K \mathbb{E}\|\nabla \tilde{f}(x_t)\|^2 \leq f(x_0) - f(x_T) + T(\eta\eta_L K) \left( \frac{5L^2 K\eta_L^2}{2m^2} \sum_{l=1}^{m} \frac{1}{p_l^t} (\sigma_L^2 + 4K\zeta_{G,i}^2) + \frac{L\eta_L \eta}{2n} \sum_{l=1}^{m} \frac{1}{m^2 p_l^t} \sigma_L^2 \right). \tag{67}$$

Which implies:

$$\min_{t \in [T]} \mathbb{E} \|\nabla \tilde{f}(x_t)\|^2 \leq \frac{f_0 - f_*}{c \eta \eta_L K T} + \frac{1}{c} \tilde{\Phi} \,, \tag{68}$$

where $\tilde{\Phi} = \frac{5L^2 K \eta_L^2}{2m^2} \sum_{l=1}^{m} \frac{1}{p_l^t} (\sigma_L^2 + 4K\zeta_{G,i}^2) + \frac{L\eta_L\eta}{2n} \sum_{l=1}^{m} \frac{1}{m^2 p_l^t} \sigma_L^2.$

$\square$

### D.3 Sample without replacement

$$\min_{t \in [T]} \mathbb{E} \|\nabla \tilde{f}(x_t)\|^2 \leq \frac{f_0 - f_*}{c \eta \eta_L K T} + \frac{1}{c} \tilde{\Phi} \,, \tag{69}$$

where $\tilde{\Phi} = \frac{5L^2 K \eta_L^2}{2mn} \sum_{l=1}^{m} \frac{1}{p_l^t} (\sigma_L^2 + 4K\zeta_{G,i}^2) + \frac{L\eta_L\eta}{2n} \sum_{l=1}^{m} \frac{1}{m^2 p_l^t} \sigma_L^2.$

*Proof.*

$$\tilde{f}(x_{t+1}) \leq \tilde{f}(x_t) + \left\langle \nabla \tilde{f}(x_t), \mathbb{E}[x_{t+1} - x_t] \right\rangle + \frac{L}{2} \mathbb{E}_t[\|x_{t+1} - x_t\|]$$

$$= \tilde{f}(x_t) + \left\langle \nabla \tilde{f}(x_t), \mathbb{E}_t[\eta \Delta_t + \eta \eta_L K \nabla \tilde{f}(x_t) - \eta \eta_L K \nabla \tilde{f}(x_t)] \right\rangle + \frac{L}{2} \eta^2 \mathbb{E}_t[\|\Delta_t\|^2]$$

$$= \tilde{f}(x_t) - \eta \eta_L K \left\| \nabla \tilde{f}(x_t) \right\|^2 + \eta \underbrace{\left\langle \nabla \tilde{f}(x_t), \mathbb{E}_t[\Delta_t + \eta_L K \nabla \tilde{f}(x_t)] \right\rangle}_{A_1} + \frac{L}{2} \eta^2 \underbrace{\mathbb{E}_t \|\Delta_t\|^2}_{A_2} \,. \tag{70}$$

Where the first inequality follows from Lipschitz continuous condition. Here the expectation is over local data SGD and filtration of $x_t$. However, in the next analysis, the expectation is over all randomness, i.e., client sampling is included.

Similarly, we consider $A_1$ first:

$$A_1 = \left\langle \nabla \tilde{f}(x_t), \mathbb{E}_t[\Delta_t + \eta_L K \nabla \tilde{f}(x_t)] \right\rangle$$

$$= \left\langle \nabla \tilde{f}(x_t), \mathbb{E}_t \left[ -\frac{1}{|S_t|} \sum_{i \in S_t} \frac{1}{m p_i^t} \sum_{k=0}^{K-1} \eta_L g_{t,k}^i + \eta_L K \nabla \tilde{f}(x_t) \right] \right\rangle$$

$$= \left\langle \nabla \tilde{f}(x_t), \mathbb{E}_t \left[ -\frac{1}{|S_t|} \sum_{i \in S_t} \frac{1}{m p_i^t} \sum_{k=0}^{K-1} \eta_L \nabla F_i(x_{t,k}^i) + \eta_L K \nabla \tilde{f}(x_t) \right] \right\rangle$$

$$= \left\langle \sqrt{K\eta_L} \nabla \tilde{f}(x_t), \frac{\sqrt{\eta_L}}{\sqrt{K}} \mathbb{E}_t \left[ -\frac{1}{n} \sum_{i \in S_t} \frac{1}{m p_i^t} \sum_{k=0}^{K-1} \nabla F_i(x_{t,k}^i) + K \nabla \tilde{f}(x_t) \right] \right\rangle$$

$$= \frac{K\eta_L}{2} \left\| \nabla \tilde{f}(x_t) \right\|^2 + \frac{\eta_L}{2K} \mathbb{E}_t \left\| -\frac{1}{n} \sum_{i \in S_t} \frac{1}{m p_i^t} \sum_{k=0}^{K-1} \nabla F_i(x_{t,k}^i) + K \nabla \tilde{f}(x_t) \right\|^2$$

$$- \frac{\eta_L}{2K} \mathbb{E}_t \left\| -\frac{1}{n} \sum_{i \in S_t} \frac{1}{m p_i^t} \sum_{k=0}^{K-1} \nabla F_i(x_{t,k}^i) \right\|^2 \,. \tag{71}$$

Since $x_i$ are sampled from $S_t$ without replacement, this causes pairs $x_{i1}, x_{i2}$ to no longer be independent. We introduce the activation function by:

$$\mathbb{I}_m \triangleq \begin{cases} 1 & if \ x \in S_t \,, \\ 0 & \text{otherwise} \,. \end{cases} \tag{72}$$

Then we get the following bound:

$$\mathbb{E}_t \left\| \frac{1}{n} \sum_{i \in S_t} \frac{1}{mp_i^t} \sum_{k=0}^{K-1} \nabla F_i(x_{t,k}^i) - K\nabla \tilde{f}(x_t) \right\|^2$$

$$= \mathbb{E}_t \left\| \frac{1}{n} \sum_{l=1}^{m} \mathbb{I}_m \frac{1}{mp_l^t} \sum_{k=0}^{K-1} \nabla F_l(x_{t,k}^l) - \frac{1}{n} \sum_{l=1}^{m} \mathbb{I}_m \frac{1}{mp_l^t} \sum_{k=0}^{K-1} \nabla F_l(x_t) \right\|^2$$

$$\stackrel{(b1)}{\leq} \frac{m}{n^2} \sum_{l=1}^{m} \mathbb{E}_t \left\| \mathbb{I}_m \frac{1}{mp_l^t} \sum_{k=0}^{K-1} \left( \nabla F_l(x_{t,k}^l) - \nabla F_l(x_t) \right) \right\|^2$$

$$- \frac{1}{n^2} \sum_{l_1 \neq l_2} \mathbb{E}_t \left\| \left\{ \mathbb{I}_m \frac{1}{mp_{l_1}} \sum_{k=0}^{K-1} \left( \nabla F_{l_1}(x_{t,k}^{l_1}) - \nabla F_{l_1}(x_t) \right) \right\} - \left\{ \mathbb{I}_m \frac{1}{mp_{l_2}} \sum_{k=0}^{K-1} \left( \nabla F_{l_2}(x_{t,k}^{l_2}) - \nabla F_{l_2}(x_t) \right) \right\} \right\|^2$$

$$\leq \frac{m}{n^2} \sum_{l=1}^{m} \mathbb{E}_t \left\| \mathbb{I}_m \frac{1}{mp_l^t} \sum_{k=0}^{K-1} \left( \nabla F_l(x_{t,k}^l) - \frac{1}{mp_l^t} \nabla F_l(x_t) \right) \right\|^2$$

$$= \frac{m}{n^2} \sum_{l=1}^{m} \mathbb{E}_t \left\{ \left\| \mathbb{I}_m \frac{1}{mp_l^t} \sum_{k=0}^{K-1} \left( \nabla F_l(x_{t,k}^l) - \frac{1}{mp_l^t} \nabla F_l(x_t) \right) \right\|^2 \mid \mathbb{I}_m = 1 \right\} \times P(\mathbb{I}_m = 1) \qquad (73)$$

$$+ \mathbb{E}_t \left\{ \left\| \mathbb{I}_m \left( \frac{1}{mp_l^t} \sum_{k=0}^{K-1} \nabla F_l(x_{t,k}^l) - \frac{1}{mp_l^t} \nabla F_l(x_t) \right\|^2 \mid \mathbb{I}_m = 0 \right\} \times P(\mathbb{I}_m = 0) \right)$$

$$= \frac{m}{n^2} \sum_{l=1}^{m} np_l^t \mathbb{E} \left\| \frac{1}{mp_l^t} \sum_{k=0}^{K-1} \nabla F_l(x_{t,k}^l) - \frac{1}{mp_l^t} \sum_{k=0}^{K-1} \nabla F_l(x_t) \right\|^2$$

$$\stackrel{(b2)}{\leq} \frac{L^2 K}{mn} \sum_{k=0}^{K-1} \sum_{l=1}^{m} \frac{1}{p_l^t} \mathbb{E} \| x_{t,k}^l - x_t \|^2$$

$$\stackrel{(b3)}{\leq} \frac{L^2 K^2}{n} \left( 5K \frac{\eta_L^2}{m} \sum_{l=1}^{m} \frac{1}{p_l^t} (\sigma_L^2 + 4K\zeta_{G,i}^2) + 20K^2(A^2+1)\eta_L^2 \|\nabla f(x_t)\|^2 \frac{1}{m} \sum_{l=1}^{m} \frac{1}{p_l^t} \right), \qquad (74)$$

where (b1) follows from $\| \sum_{i=1}^{m} t_i \|^2 = \sum_{i \in [m]} \|t_i\|^2 + \sum_{i \neq j} \langle t_i, t_j \rangle \stackrel{c1}{=} \sum_{i \in [m]} m\|t_i\|^2 - \frac{1}{2} \sum_{i \neq j} \|t_i - t_j\|^2$ (where (c1) is due to $\langle x, y \rangle = \frac{1}{2} \left[ \|x\|^2 + \|y\|^2 - \|x-y\|^2 \right]$), and (b2) is due to $\mathbb{E}\|x_1 + \cdots + x_n\|^2 \leq n\mathbb{E} \left( \|x_1\|^2 + \cdots + \|x_n\|^2 \right)$, and (b3) is from Lemma D.8.
Therefore, we have the bound of $A_1$:

$$A_1 \leq \frac{K\eta_L}{2} \|\nabla \tilde{f}(x_t)\|^2 + \frac{\eta_L L^2 K}{2n} \left( 5K \frac{\eta_L^2}{m} \sum_{l=1}^{m} \frac{1}{p_l^t} (\sigma_L^2 + 4K\zeta_{G,i}^2) + 20K^2(A^2+1)\eta_L^2 \|\nabla f(x_t)\|^2 \frac{1}{m} \sum_{l=1}^{m} \frac{1}{p_l^t} \right)$$

$$- \frac{\eta_L}{2K} \mathbb{E}_t \left\| -\frac{1}{n} \sum_{i \in S_t} \frac{1}{mp_i^t} \sum_{k=0}^{K-1} \nabla F_i(x_{t,k}^i) \right\|^2. \qquad (75)$$

And $A_2$ has the following expression:

$$A_2 = \mathbb{E}_t \|\Delta_t\|^2$$

$$= \mathbb{E}_t \left\| -\eta_L \frac{1}{n} \sum_{i \in S_t} \frac{1}{mp_i^t} \sum_{k=0}^{K-1} g_{t,k}^i \right\|^2$$

$$= \eta_L^2 \mathbb{E}_t \left\| \frac{1}{n} \sum_{i \in S_t} \sum_{k=0}^{K-1} (\frac{1}{mp_i^t} g_{t,k}^i - \frac{1}{mp_i^t} \nabla F_i(x_{t,k}^i)) \right\|^2 + \eta_L^2 \mathbb{E}_t \left\| -\frac{1}{n} \sum_{i \in S_t} \frac{1}{mp_i^t} \sum_{k=0}^{K-1} \nabla F_i(x_{t,k}^i) \right\|^2$$

$$= \eta_L^2 \frac{1}{n^2} \mathbb{E}_t \left\| \sum_{l=1}^{m} \mathbb{I}_m \sum_{k=0}^{K-1} \frac{1}{mp_l^t} (g_{t,k}^l - \nabla F_i(x_{t,k}^i)) \right\|^2 + \eta_L^2 \mathbb{E}_t \left\| -\frac{1}{n} \sum_{i \in S_t} \frac{1}{mp_i^t} \sum_{k=0}^{K-1} \nabla F_i(x_{t,k}^i) \right\|^2$$

$$= \eta_L^2 \frac{1}{n^2} \sum_{l=1}^{m} \mathbb{E}_t \left\| \sum_{l=1}^{m} \mathbb{I}_m \sum_{k=0}^{K-1} \frac{1}{mp_l^t} (g_{t,k}^l - \nabla F_i(x_{t,k}^i)) \right\|^2 + \eta_L^2 \mathbb{E}_t \left\| -\frac{1}{n} \sum_{i \in S_t} \frac{1}{mp_i^t} \sum_{k=0}^{K-1} \nabla F_i(x_{t,k}^i) \right\|^2$$

$$= \eta_L^2 \frac{1}{n^2} \sum_{l=1}^{m} np_l^t \mathbb{E}_t \left\| \sum_{k=0}^{K-1} \frac{1}{mp_l^t} (g_{t,k}^l - \nabla F_i(x_{t,k}^i)) \right\|^2 + \eta_L^2 \mathbb{E}_t \left\| -\frac{1}{n} \sum_{i \in S_t} \frac{1}{mp_i^t} \sum_{k=0}^{K-1} \nabla F_i(x_{t,k}^i) \right\|^2$$

$$\leq \eta_L^2 \frac{K}{n} \sum_{l=1}^{m} \frac{1}{m^2 p_l^t} \sigma_L^2 + \eta_L^2 \mathbb{E}_t \left\| -\frac{1}{n} \sum_{i \in S_t} \frac{1}{mp_i^t} \sum_{k=0}^{K-1} \nabla F_i(x_{t,k}^i) \right\|^2. \tag{76}$$

Now substitute the expression of $A_1$ and $A_2$ and take the expectation over client sampling distribution on both sides. It should be noted that the derivation of $A_1$ and $A_2$ in above is based on considering the expectation over sampling distribution:

$$f(x_{t+1}) \leq f(x_t) - \eta\eta_L K \mathbb{E}_t \left\| \nabla \tilde{f}(x_t) \right\|^2 + \eta \mathbb{E}_t \left\langle \nabla \tilde{f}(x_t), \Delta_t + \eta_L K \nabla \tilde{f}(x_t) \right\rangle + \frac{L}{2} \eta^2 \mathbb{E}_t \|\Delta_t\|^2$$

$$\overset{(b4)}{\leq} f(x_t) - \eta\eta_L K \left( \frac{1}{2} - \frac{10 L^2 K^2 (A^2+1)\eta_L^2}{nm} \sum_{l=1}^{m} \frac{1}{p_l^t} \right) \mathbb{E}_t \|\nabla \tilde{f}(x_t)\|^2 + \frac{2K^2 \eta\eta_L^3 L^2}{2nm} \sum_{l=1}^{m} \frac{1}{p_l^t} (\sigma_L^2 + 4K\zeta_{G,i}^2)$$

$$+ \frac{L\eta^2 \eta_L^2 K}{2n} \sum_{l=1}^{m} \frac{1}{p_l^t} \sigma_L^2 - \left( \frac{\eta\eta_L}{2K} - \frac{L\eta^2\eta_L^2}{2} \right) \mathbb{E}_t \left\| -\frac{1}{n} \sum_{i \in S_t} \frac{1}{mp_i^t} \sum_{k=0}^{K-1} \nabla F_i(x_{t,k}^i) \right\|^2$$

$$\leq f(x_t) - c\eta\eta_L K \mathbb{E}_t \|\nabla \tilde{f}(x_t)\|^2 + \frac{2K^2 \eta\eta_L^3 L^2}{2nm} \sum_{l=1}^{m} \frac{1}{p_l^t} (\sigma_L^2 + 4K\zeta_{G,i}^2) + \frac{L\eta^2 \eta_L^2 K}{2n} \sum_{l=1}^{m} \frac{1}{p_l^t} \sigma_L^2. \tag{77}$$

Also, for (b4), step sizes need to satisfy $\left( \frac{\eta\eta_L}{2K} - \frac{L\eta^2\eta_L^2}{2} \right) \geq 0$ if $\eta\eta_l \leq \frac{1}{KL}$, and there exists a constant $c > 0$ satisfying $(\frac{1}{2} - \frac{10K^2\eta_L^2 L^2(A^2+1)}{mn} \sum_{l=1}^{m} \frac{1}{p_l^t}) > c > 0$ if $\eta_L < \frac{1}{\sqrt{20(A^2+1)}KL\sqrt{\frac{1}{n}\sum_{l=1}^{m}\frac{1}{mp_l^t}}}$.

Rearranging and summing from $t = 0, \ldots, T-1$, we have:

$$\sum_{t=1}^{T-1} c\eta\eta_L K \mathbb{E} \|\nabla \tilde{f}(x_t)\|^2 \leq f(x_0) - f(x_T) + T(\eta\eta_L K)\tilde{\Phi}. \tag{78}$$

Which implies:

$$\min_{t \in [T]} \mathbb{E} \|\nabla \tilde{f}(x_t)\|^2 \leq \frac{f_0 - f_*}{c\eta\eta_L KT} + \frac{1}{c}\tilde{\Phi}, \tag{79}$$

where $\tilde{\Phi} = \frac{5L^2 K\eta_L^2}{2mn} \sum_{l=1}^{m} \frac{1}{p_l^t} (\sigma_L^2 + 4K\zeta_{G,i}^2) + \frac{L\eta_L\eta}{2n} \sum_{l=1}^{m} \frac{1}{m^2 p_l^t} \sigma_L^2$.

$$\square$$

# E PROOF OF OPTIMAL SAMPLING PROBABILITY

## E.1 SAMPLING PROBABILITY FEDIS

**Corollary E.1** (Optimal sampling probability for FedIS).

$$\min_{p_l^t} \Phi \qquad s.t. \sum_{l=1}^{m} p_l^t = 1 \,.$$

*Solving the above optimization problem, we give the expression of optimal sampling probability:*

$$p_i^t = \frac{\|\hat{g_i^t}\|}{\sum_{j=1}^{m} \|\hat{g_j^t}\|} \,, \tag{80}$$

*where $\hat{g_i^t} = \sum_{k=0}^{K-1} g_k^i$ is the gradient updates sum of multiple updates.*

Recall theorem 3.1, only the last variance term in the convergence term $\Phi$ is affected by sampling. In other words, we need to minimize the variance term with respect to probability. We formalized it as below:

$$\min_{p_i^t \in [0,1], \sum_{i=1}^{m} p_i^t = 1} V\left(\frac{1}{mp_i^t}\hat{g_i^t}\right) \Leftrightarrow \min_{p_i^t \in [0,1], \sum_{i=1}^{m} p_i^t = 1} \frac{1}{m^2} \sum_{i=1}^{m} \frac{1}{p_i^t} \|\hat{g_i^t}\|^2 \,. \tag{81}$$

This problem can be solved in closed form by the KKT condition. It is easy to verify that the solution of the above optimization is :

$$p_{i,t}^* = \frac{\|\sum_{k=0}^{K-1} g_{t,k}^i\|}{\sum_{i=1}^{m} \|\sum_{k=0}^{K-1} g_{t,k}^i\|}, \forall i \in 1, 2, ..., m \,. \tag{82}$$

Under optimal sampling probability, the variance will be:

$$V\left(\frac{1}{mp_i^t}\hat{g^t}\right) \leq \mathbb{E}\left\|\frac{\sum_{i=1}^{m} \hat{g_i^t}}{m}\right\|^2 = \frac{1}{m^2}\mathbb{E}\|\sum_{i=1}^{m} \sum_{k=1}^{K} \nabla F_i(x_{t,k}, \xi_{k,t})\|^2 \tag{83}$$

Therefore, the variance term is bounded by:

$$V\left(\frac{1}{mp_i^t}\hat{g^t}\right) \leq \frac{1}{m} \sum_{i=1}^{m} K \sum_{k=1}^{K} \mathbb{E}\|\nabla F_i(x_{t,k}, \xi_{k,t})\|^2 \leq K^2 G^2 \tag{84}$$

**Remark:** If the uniform distribution is adopted $p_i^t = \frac{1}{m}$, it is easy to observe that the variance of the stochastic gradient is bounded by $\frac{\sum_{i=1}^{m} \|g_i\|^2}{m}$.
According to Cauchy-Schwarz inequality,

$$\frac{\sum_{i=1}^{m} \|\hat{g_i^t}\|^2}{m} \bigg/ \left(\frac{\sum_{i=1}^{m} \|\hat{g_i}\|}{m}\right)^2 = \frac{m \sum_{i=1}^{m} \|\hat{g_i}\|^2}{\left(\sum_{i=1}^{m} \|\hat{g_i}\|\right)^2} \geq 1 \,, \tag{85}$$

This implies that importance sampling does improve convergence rate, especially when $\frac{\left(\sum_{i=1}^{m} \|g_i\|\right)^2}{\sum_{i=1}^{m} \|g_i\|^2} << m$.

## E.2 SAMPLING PROBABILITY OF DELTA

Our result is of the following form:

$$\min_{t \in [T]} \mathbb{E}\|\nabla f(x_t)\|^2 \leq \frac{f_0 - f_*}{c\eta\eta_L KT} + \tilde{\Phi} \,, \tag{86}$$

it's easy to see that the sampling strategy only affects $\tilde{\Phi}$, for enhancing the convergence rate, we need to minimize $\tilde{\Phi}$ with respect to $p_l^t$. As is shown, the expression of $\tilde{\Phi}$ in with and without replacement

are similar, only differ in number $n$ and $m$. Here we just consider with replacement case. Specifically, we need to solve this optimization problem:

$$\min_{p_l^t} \tilde{\Phi} = \frac{1}{c}\left(\frac{5L^2 K\eta_L^2}{2m^2}\sum_{l=1}^m \frac{1}{p_l^t}(\sigma_{L,l}^2 + 4K\zeta_{G,i}^2) + \frac{L\eta_L\eta}{2n}\sum_{l=1}^m \frac{1}{m^2 p_l^t}\sigma_{L,i}^2\right) \qquad s.t. \sum_{l=1}^m p_l^t = 1\,.$$

Solving this optimization problem, we can find the optimal sampling probability to be:

$$p_{i,t}^* = \frac{\sqrt{5KL\eta_L(\sigma_{L,i}^2 + 4K\zeta_{G,i}^2) + \frac{\eta}{n}\sigma_{L,l}^2}}{\sum_{l=1}^m \sqrt{5KL\eta_L(\sigma_{L,l}^2 + 4K\zeta_{G,l}^2) + \frac{\eta}{n}\sigma_{L,l}^2}}\,. \tag{87}$$

For simplicity's sake, we rewrote the optimal sampling probability as :

$$p_{i,t}^* = \frac{\sqrt{\alpha_1\zeta_{G,i}^2 + \alpha_2\sigma_{L,i}^2}}{\sum_{l=1}^m \sqrt{\alpha_1\zeta_{G,l}^2 + \alpha_2\sigma_{L,l}^2}}\,, \tag{88}$$

where $\alpha_1 = 20K^2 L\eta_L$, $\alpha_2 = 5KL\eta_L + \frac{\eta}{n}$.

**Remark:** Now we compare with the uniform sampling strategy:

$$\Phi_{DELTA} = \frac{L\eta_L}{2c}\left(\frac{\sum_{l=1}^m \sqrt{\alpha_1\zeta_{G,l}^2 + \alpha_2\sigma_{L,l}^2}}{m}\right)^2\,. \tag{89}$$

For uniform $p_l = \frac{1}{m}$:

$$\Phi_{uniform} = \frac{L\eta_L}{2c}\frac{\sum_{l=1}^m \left(\sqrt{\alpha_1\zeta_{G,l}^2 + \alpha_2\sigma_{L,l}^2}\right)^2}{m}\,. \tag{90}$$

According to Cauchy-Schwarz inequality:

$$\frac{\sum_{l=1}^m \left(\sqrt{\alpha_1\zeta_{G,l}^2 + \alpha_2\sigma_{L,l}^2}\right)^2}{m}\Bigg/\left(\frac{\sum_{l=1}^m \sqrt{\alpha_1\zeta_{G,l}^2 + \alpha_2\sigma_{L,l}^2}}{m}\right)^2 = \frac{m\sum_{l=1}^m \left(\sqrt{\alpha_1\zeta_{G,l}^2 + \alpha_2\sigma_{L,l}^2}\right)^2}{\left(\sum_{l=1}^m \sqrt{\alpha_1\zeta_{G,l}^2 + \alpha_2\sigma_{L,l}^2}\right)^2} \geq 1\,, \tag{91}$$

implies that importance sampling does improve convergence rate (importance sampling-based approach might be $n$-times faster in convergence than uniform), especially when $\frac{\left(\sum_{l=1}^m \sqrt{\alpha_1\zeta_{G,l}^2 + \alpha_2\sigma_{L,l}^2}\right)^2}{\sum_{l=1}^m \left(\sqrt{\alpha_1\zeta_{G,l}^2 + \alpha_2\sigma_{L,l}^2}\right)^2} << m$.

## F  CONVERGENCE ANALYSIS OF THE PRACTICAL ALGORITHM

For providing the convergence rate of applying the practical algorithm, we need an additional Assumption:

**Assumption 5** (Local gradient norm bound). *The gradients $\nabla F_i(x)$ are uniformly upper bounded (by a constant $G > 0$) $\|\nabla F_i(x)\|^2 \leq G^2, \forall i$.*

Assumption 5 is a general assumption in IS community to bound the gradient norm (Zhao & Zhang, 2015; Elvira & Martino, 2021; Katharopoulos & Fleuret, 2018), and it is also used in the FL community to analyze convergence (Balakrishnan et al., 2021; Zhang et al., 2020). This assumption tells us a useful fact that will be used later:

$|\nabla F_i(x_{t,k}, \xi_{t,k})/\nabla F_i(x_{s,k}, \xi_{s,k})| \leq U$ for all $i$ and $k$, where subscribe $s$ refers to the lasted participated round of client $i$, and $U$ is a constant upper bound. It tells us that the client's gradient norm change is bounded.

In general, the gradient norm tends to be smaller as training progresses, thus leading the $|\nabla F_i(x_{t,k}, \xi_{t,k})/\nabla F_i(x_{s,k}, \xi_{s,k})|$ goes to zero. Even if there are some oscillations in the gradient norm, the gradient will vary within a limited range and will not appear to be infinite.

Based on Assumption 5 and Assumption 3, we can re-derive the convergence analysis for both convergence variance $\Phi$ (4) and $\tilde{\Phi}$ (47). As for Assumption 3 ($\mathbb{E}\|\nabla F_i(x)\|^2 \leq (A^2+1)\|\nabla f(x)\|^2 + \sigma_G^2$), we use $\sigma_{G,s}$ and $\sigma_{G,t}$ instead of a unified $\sigma_G$ for the sake of comparison.

Specifically, $\Phi = \frac{1}{c}[\frac{5\eta_L^2 L^2 K}{2m} \sum_{i=1}^{m}(\sigma_L^2 + 4K\sigma_G^2) + \frac{\eta\eta_L L}{2m}\sigma_L^2 + \frac{L\eta\eta_L}{2nK}V(\frac{1}{mp_i^t}\hat{g}_i^t)]$, where $\hat{g}_i^t = \sum_{k=1}^{K}\nabla F_i(x_{k,s}, \xi_{k,s})$. With the practical sampling probability $p_i^s$ of FedIS, the term

$$V\left(\frac{1}{mp_i^s}\hat{g}_i^t\right) = E\|\frac{1}{mp_i^s}\hat{g}_i^t - \frac{1}{m}\sum_{i=1}^{m}\hat{g}_i^t\|^2 \leq E\|\frac{1}{mp_i^t}\hat{g}_i^t\|^2 = E\|\frac{1}{m}\frac{\hat{g}_i^t}{\hat{g}_i^s}\sum_{j=1}^{m}\hat{g}_j^s\|^2. \quad (92)$$

According to Assumption 5, we know $\|\frac{\hat{g}_i^t}{\hat{g}_i^s}\|^2 = \|\frac{\sum_{k=1}^{K}\nabla F_i(x_{t,k}^i, \xi_{t,k}^i)}{\sum_{k=1}^{K}\nabla F_i(x_{s,k}^i, \xi_{s,k}^i)}\| \leq U^2$. Then we get

$$V\left(\frac{1}{mp_i^s}\hat{g}_i^t\right) \leq E\left(\|\frac{1}{m}\|^2\|\|\frac{\hat{g}_i^t}{\hat{g}_i^s}\|^2\|\sum_{j=1}^{m}\hat{g}_j^s\|^2\right) \leq \frac{1}{m^2}U^2 E\|\sum_{i=1}^{m}\sum_{k=1}^{K}\nabla F_i(x_{k,s}, \xi_{k,s})\|^2$$

$$\leq \frac{1}{m^2}U^2 m \sum_{i=1}^{m} K \sum_{k=1}^{K} E\|\nabla F_i(x_{k,s}, \xi_{k,s})\|^2 \quad (93)$$

Similar to the previous proof, based on Assumption 3. we can get the new convergence rate:

$$\min_{t\in[T]} E\|\nabla f(x_t)\|^2 \leq \mathcal{O}\left(\frac{f^0 - f^*}{\sqrt{nKT}}\right) + \underbrace{\mathcal{O}\left(\frac{\sigma_L^2}{\sqrt{nKT}}\right) + \mathcal{O}\left(\frac{M^2}{T}\right) + \mathcal{O}\left(\frac{KU^2\sigma_{G,s}^2}{\sqrt{nKT}}\right)}_{\text{order of } \Phi}. \quad (94)$$

where $M = \sigma_L^2 + 4K\sigma_{G,s}^2$.

**Remark F.1.** *It is worth noting that $|\nabla F_i(x_{t,k}, \xi_{t,k})/\nabla F_i(x_{s,k}, \xi_{s,k})|$ is usually relatively small because the gradient tends to go to zero as training processing. It means $U$ can be relatively small, more specifically, $U < 1$ in the upper bound term $\mathcal{O}\left(\frac{KU^2\sigma_{G,s}^2}{\sqrt{nKT}}\right)$. However, it does not mean the practical algorithm is better than the theoretical algorithm because the $\sigma_G$ is different, as we stated at the beginning. Usually, $\sigma_{G,s}$ of the practical algorithm is larger than $\sigma_{G,t}$, which also comes from the fact that the gradient tends to go to zero as training processing. Besides, due to the presence of the summation over both $i$ and $k$, the gap between $\sigma_{G,s}$ and $\sigma_{G,t}$ is multiplied, and $\sigma_{G,s}/\sigma_{G,t} \sim m^2 K^2 \frac{1}{U^2}$. Thus, the practical algorithm leads to a slower convergence than the theoretical algorithm.*

Similarly, as long as the gradient is consistently bounded, we can assume $|\nabla F_i(x_t) - \nabla f(x_t)|/|\nabla F_i(x_s) - \nabla f(x_s)| \leq \tilde{U}_1 \leq \tilde{U}$ and $|\sigma_{L,t}/\sigma_{L,s}| \leq \tilde{U}_2 \leq \tilde{U}$ where $\sigma_{L,s}^2 = \mathbb{E}\left[\|\nabla F_i(x_s, \xi_s^i) - \nabla F_i(x_s)\|\right]$ for all $i$. Then we can get a similar conclusion following the same analysis on $\tilde{\Phi}$.

Specifically, $\tilde{\Phi} = \frac{L\eta_L}{2m^2c}\sum_{i=1}^{m}\frac{1}{p_i^s}\left(\alpha_1\zeta_{G,i}^2 + \alpha_2\sigma_{L,i}^2\right)$, where $\alpha_1$ and $\alpha_2$ are constants defined in (11). For the sake of comparison of different participated rounds $s$ and $t$, we rewrite the symbol as $\zeta_{G,s}^i$ and $\sigma_{L,s}^i$. Then use the practical sampling probability $p_i^s$ of DELTA, and let $R_i^s = \sqrt{\alpha_1 \zeta_{G,s}^i{}^2 + \alpha_2 \sigma_{L,s}^i{}^2}$, we have

$$\tilde{\Phi} = \frac{L\eta_L}{2m^2c}\sum_{i=1}^{m}\frac{1}{p_i^s}(R_i^t)^2 = \frac{L\eta_L}{2m^2c}\sum_{i=1}^{m}\frac{(R_i^t)^2}{R_i^s}\sum_{j=1}^{m}(R_j^s)^2 = \frac{L\eta_L}{2m^2c}\sum_{i=1}^{m}\left(\frac{R_i^t}{R_i^s}\right)^2 R_i^s \sum_{j=1}^{m}R_j^s$$

$$\leq \frac{L\eta_L}{2m^2c}\tilde{U}^2 \sum_{i=1}^{m}R_i^s \sum_{j=1}^{m}R_j^s = \frac{L\eta_L}{2m^2c}\tilde{U}^2\left(\sum_{i=1}^{m}R_i^s\right)^2 \leq \frac{L\eta_L}{2m^2c}\tilde{U}^2 m \sum_{i=1}^{m}(R_i^s)^2$$

$$\leq \frac{L\eta_L}{2c}\tilde{U}^2(5KL\eta_L(\sigma_{L,s}^2 + 4K\zeta_{G,s}^2) + \frac{\eta}{n}\sigma_L^2) \quad (95)$$

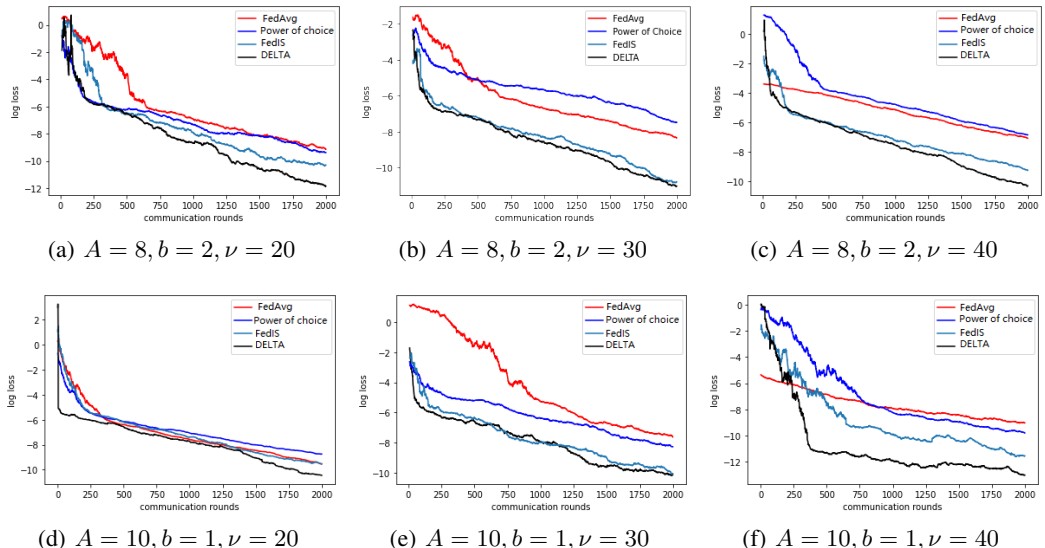

Figure 11: Performance of different algorithms on the regression model. The loss is calculated by $f(x, y) = \left\| y - log\left(\frac{(A_i x - b_i)^2}{2}\right) \right\|^2$, we report the logarithm of global loss with different degree of gradient noise $\nu$. All methods are well-tuned, and we report the best result of each algorithm under each setting.

Therefore, compared with the theoretical algorithm of DELTA, the practical algorithm of DELTA has a convergence rate as follows:

$$\min_{t \in [T]} \mathbb{E}\|\nabla f(x_t)\|^2 \leq \mathcal{O}\left(\frac{f^0 - f^*}{\sqrt{nKT}}\right) + \underbrace{\mathcal{O}\left(\frac{\tilde{U}^2 \sigma_{L,s}^2}{\sqrt{nKT}}\right) + \mathcal{O}\left(\frac{\tilde{U}^2 \sigma_{L,s}^2 + 4K\tilde{U}^2 \zeta_{G,s}^2}{KT}\right)}_{\text{order of } \tilde{\Phi}}. \quad (96)$$

This discussion of the effect $\tilde{U}$ on convergence rate is the same as $U$ in Remark F.1.

## G  EXPERIMENT DETAILS.

### G.1  ADDITIONAL EXPERIMENTS

**Synthetic dataset**   We demonstrate the experiment in different functions with different A and b. Each function is set with the noise of 20,30,40 to illustrate our theoretical results. As for constructing different functions, we assign $A = 8, 10$ and b =2, 1 respectively to see the convergence behavior of different functions.

We choose 10 out of 20 clients in each round. All the algorithms run in the same environment with a fixed learning rate of 0.001. We train each experiment for 2000 rounds to make global loss have a stable convergence performance. We display the log of global loss in Fig 11, where the Power-of-Choice is a biased sampling strategy that selects clients with higher loss (Cho et al., 2020).

We also show the convergence behavior of different sampling algorithms under small noise, as shown in Fig12.

And to be consistent with the cross-device scenario, we further expanded the number of clients from 20 to 200, keeping 10 clients selected to participate in each round. The results in Fig 13 show the effectiveness of DELTA.

**The implementation detail of different sampling algorithms**   The power-of-choice sampling method is proposed by Cho et al. (2020). The sampling strategy is that it first samples 20 clients randomly from all clients, and then chooses 10 of the 20 clients with the largest loss as selected clients. FedAvg samples clients according to their data ratio. Thus, FedAvg promises to be unbiased, which is given in Fraboni et al. (2021a); Li et al. (2019) to be an unbiased sampling method. As for FedIS, the sampling strategy follows (82). And for DELTA, the sampling probability follows (11). For practical implementation of FedIS and DELTA, the sampling probability follows the strategy we described in Section 4.

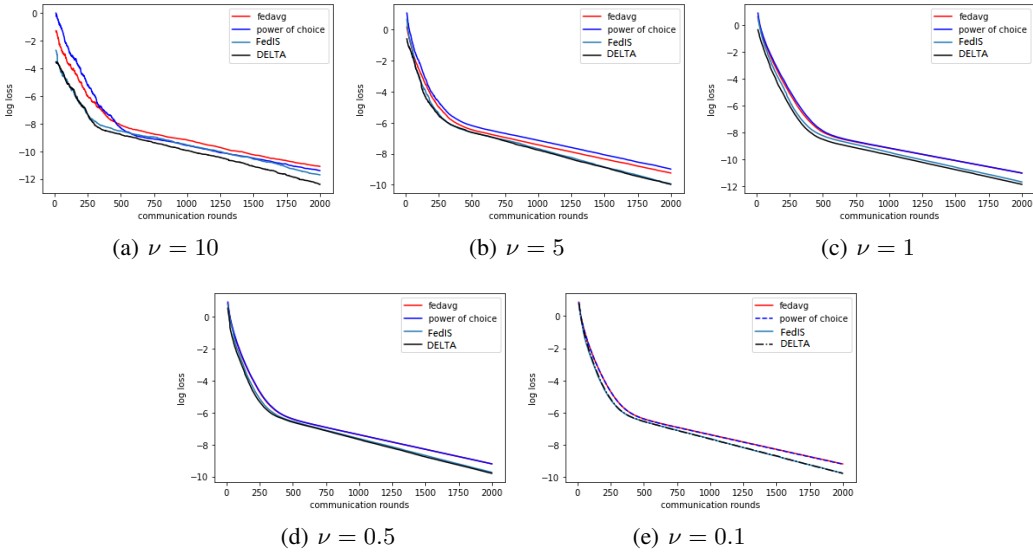

Figure 12: Performance of different algorithms on the regression model with different (small) noise setting.

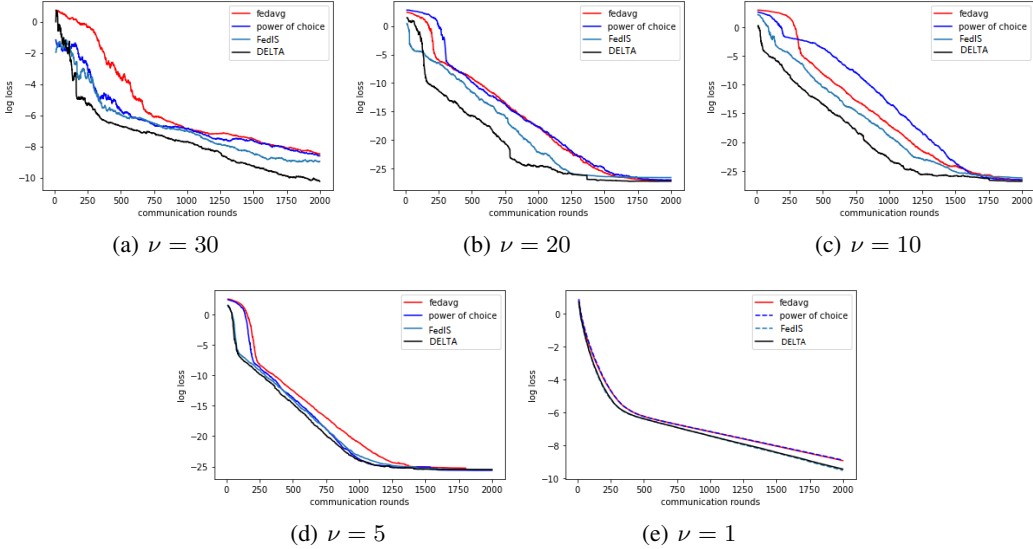

Figure 13: Performance of different algorithms on synthetic data with different noise setting. Specifically, for testing the large client number setting, each round 10 out of 200 clients are selected to participate in training.

**Split FEMNIST**  In this section, we consider the split FEMNIST. We let $10\%$ clients own $90\%$ data and the detailed split data process is shown below.

- Divide the dataset by labels, for example, divide FEMNIST into 10 groups, and assign each client one label
- Random select one client
- Reshuffle the data in the selected client
- Equally divided into 100 clients

**FEMNIST and CIFAR-10**  Specifically, we train a two-layer MLP on the split-FEMNIST and a resnet 18 on split-CIFAR-10, respectively. CIFAR10 is composed of 32x32 images with three RGB channels of 10 different classes with 60000 samples. The "split" follows the idea introduced in Yu et al. (2019); Hsu et al. (2019), where we leverage the Latent Dirichlet Allocation (LDA) to control the distribution drift with the dirichlet parameter $\alpha$. Larger $\alpha$ indicates smaller drifts. Unless otherwise stated ,we set dirichlet parameter $\alpha = 0.5$.

Unless specifically mentioned otherwise, our studies use the following protocol. All datasets are split with parameter $\alpha = 0.5$, the server choose $n = 20$ clients according to our proposed probability from

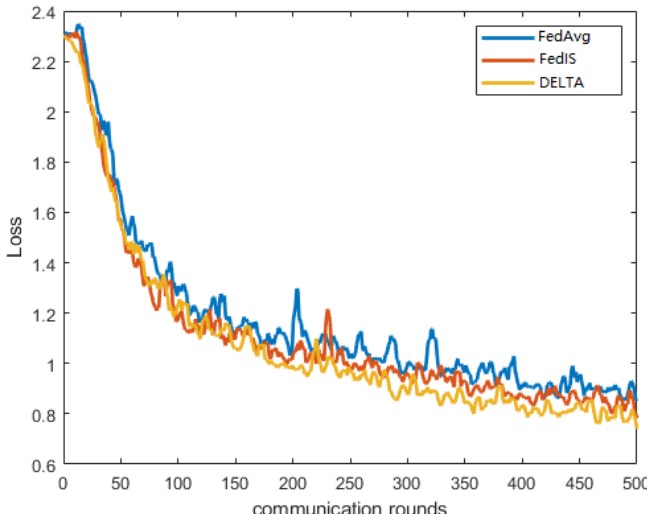

Figure 14: **Loss performance of= DELTA, FedIS and FedAvg on FEMNIST**.

the total of $m = 300$ clients, and each is trained for $T = 500$ communication rounds with $K = 5$ local epochs. The default local dataset batch size is 32. The learning rates are set the same in all algorithms, specifically $lr_{global} = 1$ and $lr_{local} = 0.01$.

All algorithms use FedAvg as the backbone. We compare DELTA and FedIS with FedAvg in different datasets with different settings.

**Loss performance of FEMNIST**    We compare the loss of DELTA, FedIS and uniform sampling on the non-iid FEMNIST dataset in Fig 14. It shows that DELTA and FedIS converges faster than FedAvg, while DELTA even achieves a lower loss than FedIS.

For CIFAR-10, we report the mean of the best 10 test accuracies on global test data here. In Table 2 we compare the performance of DELTA, FedIS, and FedAvg on non-IID FEMNIST and CIFAR-10. Specifically, we use $\alpha = 0.1$ for FEMNIST and $\alpha = 0.5$ for CIFAR-10 to split dataset. As for Multinomial Distribution (MD) sampling (Li et al., 2018), it samples based on clients' data ratio and average aggregates. It is symmetric in sampling and aggregation with FedAvg, with similar performance. It can be seen that DELTA has better accuracy than FedIS, while DELTA and FedIS both outperform FedAvg with the same communication round.

In Table 3, we demonstrate that DELTA and FedIS is compatible with other FL optimization algorithms, e.g., Fedprox (Li et al., 2018) and FedMime (Karimireddy et al., 2020a). Moreover, DELTA keeps its superiority in this setting.

Table 3: **Performance of algorithms with momentum and prox.** We run 500 communication rounds on CIFAR10 for each algorithm. We report the mean of maximum 5 accuracies for test datasets and the number of communication rounds to reach the threshold accuracy.

| Algorithm | CIFAR-10 + momentum | | CIFAR-10 + prox | |
|---|---|---|---|---|
| | Acc (%) | Rounds for 65% | Acc (%) | rounds for 65% |
| FedAvg (w/ uniform sampling) | 0.6567 | 390 | 0.6596 | 283 |
| FedIS | 0.6571 | **252** | 0.661 | 266 |
| DELTA | **0.6604** | 283 | **0.6677** | **252** |

In Table 4, we demonstrate that DELTA and FedIS is compatible with other variance reduction algorithms, like FedVARP (Jhunjhunwala et al., 2022).

It is worth noting that FedVARP utilizes the historic update to approximate the unparticipated clients' updates. However, in this setting, the improvement of the sampling strategy on the results is somewhat reduced. This is because the sampling strategy is slightly redundant when all users are involved. So when VARP and DELTA/FedIS are combined, instead of reassigning weights in the aggregation step,

we use 82 or 11 to select the current round update clients and then average aggregate the updates of all clients. One can see that the combination of DELTA/FedIS and VARP can still show the advantages of sampling.

Table 4: **Performance of DELTA/FedIS in combination with FedVARP.** We run 500 communication rounds on FEMNIST with $\alpha = 0.1$ for each algorithm. We report the mean of maximum 5 accuracies for test datasets and the number of communication rounds to reach the threshold accuracy.

| Algorithm | FEMNIST | |
| --- | --- | --- |
| | Acc (%) | Rounds for 73% |
| FedVARP | $73.81 \pm 0.18$ | 470 |
| FedIS + FedVARP | $73.96 \pm 0.14$ | 452 |
| DELTA +FedVARP | $\mathbf{74.22} \pm 0.14$ | **436** |

We also experiment with different choices of heterogeneity $\alpha$ in CIFAR-10. The parameter of heterogeneity $\alpha$ changes from 0.1 to 0.5 to 1. We observe the consistent improvement of DELTA in Table 5.

Table 5: **Performance of algorithms under different $\alpha$.** We run 500 communication rounds on CIFAR10 for each algorithm (with momentum). We report the mean of maximum 5 accuracies for test datasets and the number of communication rounds to reach the threshold accuracy.

| Algorithm | $\alpha = 0.1$ | | $\alpha = 0.5$ | | $\alpha = 1.0$ | |
| --- | --- | --- | --- | --- | --- | --- |
| | Acc (%) | Rounds for 42% | Acc (%) | rounds for 65% | Acc (%) | rounds for 71% |
| FedAvg (w/ uniform sampling) | 0.4209 | 263 | 0.6567 | 283 | 0.7183 | 246 |
| FedIS | 0.427 | 305 | 0.6571 | **252** | 0.7218 | 239 |
| DELTA | **0.4311** | **209** | **0.6604** | 283 | **0.7248** | **221** |

Besides, we also experiment with various client numbers to examine the efficiency of DELTA in FEMNIST dataset. Here we set $\alpha = 1$, and participated client number choose from $n = 10, 30, 50$. As shown in Table 6, DELTA maintains its supremacy with different participating client numbers.

Table 6: **Performance of algorithms under different participated client number $n$.** We run 500 communication rounds on FEMNIST for each algorithm. We report the mean of maximum 5 accuracies for test datasets and the number of communication rounds to reach the threshold accuracy.

| Algorithm | $n = 10$ | | $n = 30$ | | $n = 50$ | |
| --- | --- | --- | --- | --- | --- | --- |
| | Acc (%) | Rounds for 85% | Acc (%) | rounds for 85% | Acc (%) | rounds for 85% |
| FedAvg (w/ ) | 0.8717 | 263 | 0.8727 | 267 | 0.8729 | 239 |
| FedIS | 0.8739 | 305 | 0.8734 | 286 | 0.8751 | 222 |
| DELTA | **0.8741** | **209** | **0.8746** | 270 | 0.8747 | **212** |

