# OpenReview forum: "DELTA: Diverse Client Sampling for Fasting Federated Learning"
_ICLR.cc/2023/Conference — Submitted to ICLR 2023_

### Official Review · Reviewer_xCAd · 2022-10-20

**Confidence:** 3
**Correctness:** 4
**Technical Novelty And Significance:** 3
**Empirical Novelty And Significance:** 2
**Recommendation:** 6

**Clarity, Quality, Novelty And Reproducibility:**

The paper is very well written, which is impressive for a theoretically focused paper.
Experiments are well described and would not be too challenging to reproduce.


**Strength And Weaknesses:**

Strengths
- Strong theoretical analysis.
- Theory is interpretable and illuminating about the nature of the problem.

Weaknesses
- Experimental results are somewhat weak, with only a marginal improvement over FedIS in experiments on real datasets.

**Summary Of The Paper:**

The authors explore the use of importance sampling (IS) its extensions for selecting clients in FL, where IS is based off gradient norms. They first provide a convergence analysis for standard IS sampling (sampling proportional to size of gradients). They then propose an alternative IS sampling which captures both gradient magnitude, as well as gradient diversity (the difference between the client's gradient and the global objective function gradient). They provide a convergence analysis for this sampling scheme and compare the scheme with the original FedIS.

**Summary Of The Review:**

This paper extends the analysis of IS for FL, and proposes a principled extension which (at least partially) solves a well known diversity issue that is common to IS in general (though it's certainly even more pronounced in FL). The clear theoretical analysis is a valuable contribution to the field of client selection for FL.

---

> ### Author Response · Authors · 2022-11-11
> **Response to reviewer xCAd**
>
> > ### Experimental results are somewhat weak, with only a marginal improvement over FedIS in experiments on real datasets.
>
> Thank you for your appreciation of our theoretical work.\
> During the rebuttal, we added the following analysis results:
> - We have provided the additional convergence analysis without Assumption 4 for FedIS in Appendix C.
> - We have provided the theoretical analysis of the impacts of the practical algorithm and given its convergence rate in Appendix F.
> - We have also provided additional experimental results to compare the wall-clock convergence time between different sampling methods over FEMNIST and Cifar10 datasets, the result is updated in Table 2 in our revised version.
>
> We would like to add a few more observations on the findings of the experiment. Although the gain in accuracy performance is modest,  the improvement in convergence rate is more noticeable both in terms of the number of iterations and the wall-clock time.\
> In particular, our practical algorithm DELTA needs less wall-clock time to reach the threshold of accuracy because DELTA needs more less communication rounds while the computation time of DELTA in each round is comparable with FedIS and FedAvg.
>
> |  Algorithm on FEMNIST   | Accuracy (%) |  Rounds for 70%   | Time(s) for 70%   | Single round' computation time (s)   |
> |  ----  | ----  | ----  | ----  |----  |
> | FedAvg  | 70.35$\pm$ 0.51 |426 (1.0$\times$) |1795.12 (1.0$\times$) |      4.21    |
> | Cluster-based IS  | 71.21 $\pm$ 0.24 |362 (1.17$\times$) |1547.41 (1.16$\times$) | 4.27    |
> | FedIS  | 71.69$\pm$ 0.43  |404 (1.05$\times$)  |1719.26 (1.04$\times$) | 4.26    |
> | DELTA  | **72.10**$\pm$ 0.49 |**322 (1.32$\times$)** |**1372.33 (1.31$\times$)** | 4.26   |
>
>
> |  Algorithm on Cifar10  | Accuracy (%) |  Rounds for 54%   | Time(s) for 54%   |Single round' computation time  (s)   |
> |  ----  | ----  | ----  | ----  |----  |
> | FedAvg  | 54.28$\pm$ 0.29 |338 (1.0$\times$) |3283.14 (1.0$\times$) |9.71 |
> | Cluster-based IS  | 54.83$\pm$ 0.02 |323 (1.05$\times$) |3188.54 (1.03$\times$)| 9.87|
> | FedIS  | 55.05$\pm$ 0.27|313 (1.08$\times$)  |3085.05 (1.06$\times$) |9.85|
> | DELTA  | **55.20**$\pm$ 0.26 |**303 (1.12$\times$)** |**2989.98 (1.1$\times$)** |9.86|
>
>
>
> We hope the above responses address your concerns. Please let us know if you have other questions. We’re happy to further answer the questions.

---

> ### Author Response · Authors · 2022-11-15
> **Look forward to your feedback!**
>
> Dear reviewer xCAd,
>
> Thank you again for your time and efforts in reviewing. As the discussion period is ending soon, we would like to kindly remind you to check our responses and the revised version of our paper. We hope they can address your concerns and look forward to your feedback.

---

### Official Review · Reviewer_N8Mo · 2022-11-01

**Confidence:** 4
**Correctness:** 4
**Technical Novelty And Significance:** 3
**Empirical Novelty And Significance:** 2
**Recommendation:** 6

**Clarity, Quality, Novelty And Reproducibility:**

This paper is well-written in general. However, the paper could have been organized in a better way. For example, the DELTA algorithm should be placed in much earlier sections. In the current form of this paper, the DELTA algorithm appears rather late, which left quite a few notations undefined (e.g., $\eta$ and $\eta_L$) and created some difficulty/confusion in following the paper. The proposed DELTA algorithm is novel. The reproducibility of this paper is good.

**Strength And Weaknesses:**

Strengths:
1. The client sampling problem in FL is a timely and important problem in FL.
2. The proposed DELTA client sampling scheme captures diverse clients similar to the federated importance sampling (FedIS) scheme while offering unbiased client sampling performance.
3. The new and tighter convergence results for FedIS contribute to new understandings of FedIS.

Weaknesses:
1. The bounded stochastic gradient assumption in Assumption 4 is a bit restrictive and no longer needed in many state-of-the-art FL algorithms' convergence analyses. It may be interesting to see whether this assumption can be relaxed for the DELTA client sampling scheme.

2. The sampling probability $p_i^t$ in Eq. (11) requires full gradient evaluations, which are difficult to implement in practice (also admitted by the authors). To address this challenge, the authors proposed the use of stochastic gradients to approximate the computations. However, the authors didn't theoretically analyze the impacts of the approximation errors on the convergence rate performance, which is somewhat disappointing.

3. The experiment comparisons in Section 5 may be unfair. In the comparisons between DELTA, FedAvg, and FedIS, the authors only compared the convergence speeds in terms of iterations. However, the proposed DELTA method requires rather complicated calculations of $p_i^t$ in Eq. (11) compared to simple uniform sampling in FedAvg and relatively straightforward calculations of $p_i^t$ in FedIS. That is, the per-iteration complexities of these methods are quite different. Thus, it may be better to also compare the wall-clock convergence time between these methods.

**Summary Of The Paper:**

This paper considered the problem of client sampling in federated learning (FL) to improve the convergence speed of FL training. The authors proposed a new client sampling scheme called DELTA, which is unbiased and able to sample more diverse clients that carry valuable information for global model updates. The authors conducted a theoretical convergence rate performance analysis and verified the theoretical convergence performance of their proposed algorithm through simulation experiments.

**Summary Of The Review:**

This paper studied the problem of client sampling in FL. The authors proposed a new client sampling algorithm called DELTA, which could achieve both unbiasedness and capture diverse client information. The authors provided rigorous theoretical performance analysis and also provided new insights for FedIS, which is also a new contribution. However, the authors didn't provide any theoretical performance analysis for DELTA with stochastic gradient approximation. Also, some experimental results on convergence speed comparisons may be unfair.

---

> ### Author Response · Authors · 2022-11-11
> **Response to reviewer N8Mo (1/3)**
>
>
> > ### Q3: The experiment comparisons in Section 5 may be unfair. In the comparisons between DELTA, FedAvg, and FedIS, the authors only compared the convergence speeds in terms of iterations. However, the proposed DELTA method requires rather complicated calculations of pit in Eq. (11) compared to simple uniform sampling in FedAvg and relatively straightforward calculations of pit in FedIS. That is, the per-iteration complexities of these methods are quite different. Thus, it may be better to also compare the wall-clock convergence time between these methods.
>
> Thank you for your thoughtful feedback, your suggestions really help enrich our experimental results.
> - We have provided the additional experimental results of the wall-clock time comparison in Table 2 of our revised version.
> - The wall-clock time we report is the average results of multiple replicate experiments with different random seeds.
>
> We also give the comparison here:
> |  Algorithm on FEMNIST   | Accuracy (%) |  Rounds for 70%   | Time(s) for 70%   | Single round' computation time   (s)   |
> |  ----  | ----  | ----  | ----  |----  |
> | FedAvg  | 70.35$\pm$ 0.51 |426 (1.0$\times$) |1795.12 (1.0$\times$) |      4.21    |
> | Cluster-based IS  | 71.21 $\pm$ 0.24 |362 (1.17$\times$) |1547.41 (1.16$\times$) | 4.27    |
> | FedIS  | 71.69$\pm$ 0.43  |404 (1.05$\times$)  |1719.26 (1.04$\times$) | 4.26    |
> | DELTA  | **72.10**$\pm$ 0.49 |**322 (1.32$\times$)** |**1372.33 (1.31$\times$)** | 4.26   |
>
>
> |  Algorithm on Cifar10  | Accuracy (%) |  Rounds for 54%   | Time(s) for 54%   |Single round' computation time (s)   |
> |  ----  | ----  | ----  | ----  |----  |
> | FedAvg  | 54.28$\pm$ 0.29 |338 (1.0$\times$) |3283.14 (1.0$\times$) |9.71 |
> | Cluster-based IS  | 54.83$\pm$ 0.02 |323 (1.05$\times$) |3188.54 (1.03$\times$)| 9.87|
> | FedIS  | 55.05$\pm$ 0.27|313 (1.08$\times$)  |3085.05 (1.06$\times$) |9.85|
> | DELTA  | **55.20**$\pm$ 0.26 |**303 (1.12$\times$)** |**2989.98 (1.1$\times$)** |9.86|
>
> The experiment result shows **our experiment comparison is fair** because:
> - FedIS and cluster-based IS and DELTA is comparable in computation time.
> - Compared with FedAvg, though DELTA has a trival computation overhead in each round, DELTA requires less training rounds and the total computation is less than FedAvg.
>
>
>
>
>
>
> > ### Q: The DELTA algorithm should be placed in much earlier sections.
> We sincerely appreciate your suggestion, and we have relocated the algorithm in Section 3.1 in our revised version.
>
>
> We hope the above responses address your concerns. Please let us know if you have other questions. We’re happy to further answer the questions.

---

> ### Author Response · Authors · 2022-11-11
> **Response to reviewer N8Mo (2/3)**
>
> >### Q2: The sampling probability $p_i^t$ in Eq. (11) requires full gradient evaluations, which are difficult to implement in practice (also admitted by the authors). To address this challenge, the authors proposed the use of stochastic gradients to approximate the computations. However, the authors didn't theoretically analyze the impacts of the approximation errors on the convergence rate performance, which is somewhat disappointing.
>
> Thank you for sharing the potential direction for improvement. Motivated by your comment, **we have provided the convergence analysis for the practical sampling probability in Appendix F of the revised version.**
>
> The convergence result under the practical algorithm of both FedIS and DELTA is as follows:
>
> - For FedIS, the convergence rate of the practical algorithm is
>   $$\min \limits_{t\in[T]} E\|\nabla f(x_t)\|^2\leq \mathcal{O}\left(\frac{ f^0-f^*}{\sqrt{nKT}}\right) + \underbrace{\mathcal{O}\left(\frac{\sigma_L^2}{\sqrt{nKT}}\right) +  \mathcal{O}\left(\frac{M^2}{T}\right) +  \mathcal{O}\left(\frac{KU^2\sigma_G^2}{\sqrt{nKT}} \right)}_{\text{order of} \ \Phi} $$
> - For DELTA, the convergence rate of the practical algorithm is
>   $$\min_{t \in[T]} E\|\nabla f(x_t)\|^2 \leq  \mathcal{O}\left(\frac{f^0-f^*}{\sqrt{nKT}}\right) +
>         \underbrace{ \mathcal{O}\left(\frac{\tilde{U}^2\sigma_{L,s}^2}{\sqrt{nKT}}\right) + \mathcal{O}\left(\frac{\tilde{U}^2\sigma_{L,s}^2  + 4K\tilde{U}^2\zeta_{G,s}^2}{KT}\right)}_{\text{order of } \tilde{\Phi}}$$
>
>
> The $U$ and $\tilde{U}$ are two constant bounds which means the changing ratio of gradient norm. The result shows that the convergence rate of the practical algorithm is slower than that of the theoretical algorithm.
>
> **We provide the specific derivation below.** In particular, based on the additional gradient norm bound assumption, we can see that the gradient changing ratio is bounded, i.e., $|\nabla F_i(x_{t,k},\xi_{t,k}) /  \nabla F_i(x_{s,k},\xi_{s,k})| \leq U$ for all $i$ and $k$, where subscribe $s$ refers to the last participated round of client $i$, and $U$ is a constant upper bound.
> Specifically, $$\Phi = \frac{1}{c} [
>      \frac{5\eta_L^2L^2K}{2m}\sum_{i=1}^m(\sigma_L^2 +4K\sigma_G^2) + \frac{\eta\eta_LL}{2m}\sigma_L^2 + \frac{L\eta\eta_L}{2nK}V(\frac{1}{mp_i^t}\hat{g_i^t}) ]$$, where $\hat{g_i^t} = \sum_{k=1}^K\nabla F_i(x_{k,s},\xi_{k,s})$.\
> With the practical sampling probability $p_i^s$ of FedIS, the term
> $$V\left(\frac{1}{m p_{i}^{s}} \hat{g_{i}^{t}}\right) = E\|\frac{1}{mp_{i}^{s}}\hat{g_i^t} - \frac{1}{m}\sum_{i=1}^m \hat{g_i^t}\|^2 \leq E\| \frac{1}{mp_i^t}\hat{g_i^t}\|^2 = E\|\frac{1}{m}\frac{\hat{g_i^t}}{\hat{g_i^s}} \sum_{j=1}^m\hat{g_j^s} \|^{2}$$
> According to gradient changing raito bound, we know $\|\frac{\hat{g_i^t}}{\hat{g_i^s}}\|^2 = \|\frac{\sum_{k=1}^{K}\nabla F_i(x_{t,k}^i,\xi_{t,k}^i)}{\sum_{k=1}^{K}\nabla F_i(x_{s,k}^i,\xi_{s,k}^i)}\| \leq U^2$. Then we get
> $$
>     V\left(\frac{1}{m p_{i}^{s}} \hat{g_{i}^{t}}\right)  \leq  E\left(\|\frac{1}{m}\|^2\|\|\frac{\hat{g_i^t}}{\hat{g_i^s}}\|^2\|\sum_{j=1}^m\hat{g_j^s} \|^{2}\right) \leq \frac{1}{m^2}U^2E\|\sum_{i=1}^m\sum_{k=1}^K\nabla F_i(x_{k,s},\xi_{k,s})\|^2
>     \leq \frac{1}{m^2}U^2m\sum_{i=1}^mK\sum_{k=1}^KE\|\nabla F_i(x_{k,s},\xi_{k,s})\|^2
> $$
> Based on Assumption 3, we can get the new convergence rate:
> $$\min \limits_{t\in[T]} E\|\nabla f(x_t)\|^2\leq \mathcal{O}\left(\frac{ f^0-f^*}{\sqrt{nKT}}\right) + \underbrace{\mathcal{O}\left(\frac{\sigma_L^2}{\sqrt{nKT}}\right) +  \mathcal{O}\left(\frac{M^2}{T}\right) +  \mathcal{O}\left(\frac{KU^2\sigma_{G，s}^2}{\sqrt{nKT}} \right)}_{\text{order of} \ \Phi} $$
> The convergence analysis of DELTA is similar to that of FedIS, and the detailed proof and discussion for FedIS and DELTA are in Appendix F of our revised version.

---

> ### Author Response · Authors · 2022-11-11
> **Response to reviewer N8Mo (1/3)**
>
>
> Dear reviewer N8Mo, thank you for providing constructive feedback. We have fully revised our manuscript and have addressed all of the comments, as well as added new experiments to strengthen our work further. Please find our responses to your raised questions below:
>
> >### Q1: The bounded stochastic gradient assumption in Assumption 4 is a bit restrictive and no longer needed in many state-of-the-art FL algorithms' convergence analyses. It may be interesting to see whether this assumption can be relaxed for the DELTA client sampling scheme.
>
> *We would like to clarify that in our original submission, only FedIS depends on Assumption 4 while DELTA does not.* **In the revised version, we have provided the convergence rate of FedIS without Assumption 4 in Appendix C.1**
> - The convergence rate result without using Assumption 4 is as follows:
> $$\min \limits_{t\in[T]} E\|\nabla f(x_t)\|^2\leq \mathcal{O}\left(\frac{ f^0-f^*}{\sqrt{nKT}}\right) + \underbrace{\mathcal{O}\left(\frac{\sigma_L^2}{\sqrt{nKT}}\right) +  \mathcal{O}\left(\frac{M^2}{T}\right) + \mathcal{O}\left(\frac{K\sigma_G^2}{\sqrt{nKT}} \right)}_{\text{order of} \Phi}$$
> One can see the new result replaces $G$ by $\sigma_G$ while remaining the other term unchanged.
>
> - The detailed convergence rate analysis without Assumption is as below:
>
> Assumption 4 is used to bound the variance term in the final convergence result, while not used in the previous proof steps. Thus, we can focus on discussing the variance term.\
> In the final convergence result, the variance term is
> $$V\left(\frac{1}{m p_{i}^{t}} \hat{g_{i}^{t}}\right)=\frac{1}{m^2}\mathbb{E}\|\sum_{i=1}^m\sum_{k=1}^K\nabla F_i(x_{t,k},\xi_{k,t})\|^2\leq \frac{1}{m}\sum_{i=1}^mK\sum_{k=1}^K\mathbb{E}\|\nabla F_i(x_{t,k},\xi_{k,t})\|^2 \leq K^2G^2,$$ where the last inequality comes from Assumption 4.
>
> According to Assumption 3 ($\mathbb{E}\|\nabla F_i(x)\|^2 \leq (A^2+1)\|\nabla f(x)\|^2 + \sigma_G^2$), we can rewrite the variance in the convergence as follows:
> $$V\left(\frac{1}{m p_{i}^{t}} \hat{g_{i}^{t}}\right)=\frac{1}{m^2}\mathbb{E}\|\sum_{i=1}^m\sum_{k=1}^K\nabla F_i(x_{t,k},\xi_{k,t})\|^2\leq \frac{1}{m}\sum_{i=1}^mK\sum_{k=1}^K\mathbb{E}\|\nabla F_i(x_{t,k},\xi_{k,t})\|^2\leq K^2\sigma_G^2+K^2(A^2+1)\|\nabla f(x_t)\|^2.$$
> Thus in the derivation, $G^2$ can be directly substituted by $\sigma_G^2$, only the condition of $\eta_L$ changes from
> $$\frac{1}{2}-10L^2 K^2(A^2+1)\eta_L^2 > 0$$
> to
> $$\frac{1}{2}-10L^2K^2(A^2+1)\eta_L^2-\frac{L\eta K(A^2+1)}{2n}\eta_L>0.$$
> However, one can still guarantee a constant for $\eta_L$ to satisfy the new inequality according to the properties of quadratic functions. The proof and discussion are in Appendix C of our revised version.

---

> ### Author Response · Authors · 2022-11-15
> **Look forward to your feedback!**
>
> Dear reviewer N8Mo,
>
> We thank you again for your constructive comments and helpful suggestions.
> Since the discussion period is ending soon, we would like to kindly remind you to check our responses and the revised version of our paper. We hope they can address your concerns and look forward to your feedback.

---

> ### Author Response · Authors · 2022-12-10
> **To reviewer N8Mo**
>
> Dear reviewer N8Mo,
>
> Since the discussion time is ending soon, we would like to kindly remind you to check our responses and the revised version of the paper. We hope they can address your concerns, and we would be happy to know if you could reconsider your score.

---

### Official Review · Reviewer_LBTW · 2022-11-03

**Confidence:** 3
**Correctness:** 3
**Technical Novelty And Significance:** 2
**Empirical Novelty And Significance:** 2
**Recommendation:** 5

**Clarity, Quality, Novelty And Reproducibility:**

The clarity and novelty of the work should be improved and better highlighted. Details related to these concerns are mentioned in the section of strength and weaknesses.

**Strength And Weaknesses:**

Designing optimal client sampling strategies is a problem of broad interest for designing scalable federated learning methods.
The authors attempt to **design optimal client sampling strategies in theory and practice** to accelerate the convergence of commonly used federated averaging methods.

However, I would like to state the following **major concerns**:

1. **Limitations of using the optimal sampling for DELTA.**

From Corollary 3.4, the optimal sampling depends on $\Vert \nabla F_i(x) - \nabla f(x) \Vert$. This strategy cannot be implemented in practice, mainly because the gradient of the whole objective $\nabla f(x)$ cannot be accessed usually for problems over huge amounts of data.

Therefore, the authors should state how this strategy is modified to be able to implement it in their experiments.

2. **Redundant assumptions for deriving the convergence results for FedIS and DELTA.**

Assumptions 2 and 3 are commonly used for deriving convergence guarantees for federated methods. This is because Assumption 2 states how different the local stochastic gradient and the local full gradient is, and Assumption 3 implies how different the local full gradient and the whole full gradient is.

However, Assumption 4 seems to be redundant to Assumption 2, since both of them impose the property of local stochastic gradients. Therefore, the stated convergence results for FedIS and DELTA seem too restricted.

3. **Lack of motivation on how to design the optimal client sampling strategies for FedIS and DELTA.**

Since FedIS or Algorithm 3 from (Chen et al., 2020) looks similar to DELTA, the proof techniques between these methods should be similar. Hence, I am not sure how the variance from convergence guarantees for FedIS and DELTA is different. Can the authors elaborate on this and perhaps add the motivation before stating theoretical results?

4. **Numerical evaluations against other unclear existing sampling strategies.**

In the experiments, the authors compared their sampling strategies against others, e.g. the power of choice, norm, and heterogeneity. However, these strategies are not clearly stated (i.e. what is $p_i^t$)? Are they existing sampling strategies, e.g., from (Chen et al., 2020)?

Since different $p_i^t$ lead to different additional computational costs, it would be also more interesting to compare performance of the algorithms using different sampling with respect to the wall-clock time.

Furthermore, I have the following **small concerns**:

1. The step-size condition in Theorem 3.1 for $\eta_L$ and $\eta$ can be simplified to improve readability of this theorem.

The authors can write $\eta_L < \min\( 1/(8LK), C \)$ where the constant $C$ is obtained from the condition that
$1/2 - (10L^2/m)\sum_{i=1}^m K^2\eta_L^2(A^2+1)>0$. Then, the author can state the condition for $\eta$ which is $\eta \leq 1/(\eta_L L)$.

However, the constant $A$ is not clearly stated. Does it depend on the index of the client $i = 1,2,\ldots,m$?
Can the authors check this?

2. The legend in Figure 5 includes FedSRC-G and FedSRC-D. I believe that they refer to DELTA and FEDIS. Can the authors check and edit the legend?

3. Fedprox from (Li et al., 2018) does not use variance-reduction techniques to design federated optimization methods.
Therefore, the authors should revise this part of the text which is in their contribution section.

**Typo(s) I can spot:**
- the data heterogeneity to **fast** the convergence speed $\rightarrow$  the data heterogeneity to **accelerate** the convergence speed.

Note that due to limited time, I cannot check convergence proofs carefully.


**Summary Of The Paper:**

This paper proposes novel client sampling strategies to accelerate the convergence of
federated averaging methods with partial client participation. The idea is to determine
the sampling strategies to minimize variance from the worst-case convergence bounds of the methods.

**Summary Of The Review:**

I think this paper considers the problem of broad interest. However, the contributions seem to be unclear, due to unclear motivation on why and how the authors design optimal sampling strategies which are better than existing strategies. In addition, some of the proposed strategies cannot be implemented in practice, thus raising the issue on how they modify the strategies to be implementable in the experiments.

---

> ### Author Response · Authors · 2022-11-11
> **Response to LBTW (4/4)**
>
> >### Q5: Since different $p_i^t$ lead to different additional computational costs, it would be also more interesting to compare performance of the algorithms using different sampling with respect to the wall-clock time.
>
> Inspired by your comment, we designed an experiment to compare the performance with respect to the wall-clock time for different sampling methods, including FedAvg, cluster-based IS, FedIS and DELTA.
> - The experiment result is shown in Table 2 in our revised version.
> - The wall-clock time we report is the average results of multiple replicate experiments with different random seeds.
>
> We also give the comparison here:
> |  Algorithm on FEMNIST   | Accuracy (%) |  Rounds for 70%   | Time(s) for 70%   | Single round' computation time (s)  |
> |  ----  | ----  | ----  | ----  |----  |
> | FedAvg  | 70.35$\pm$ 0.51 |426 (1.0$\times$) |1795.12 (1.0$\times$) |      4.21    |
> | Cluster-based IS  | 71.21 $\pm$ 0.24 |362 (1.17$\times$) |1547.41 (1.16$\times$) | 4.27    |
> | FedIS  | 71.69$\pm$ 0.43  |404 (1.05$\times$)  |1719.26 (1.04$\times$) | 4.26    |
> | DELTA  | **72.10**$\pm$ 0.49 |**322 (1.32$\times$)** |**1372.33 (1.31$\times$)** | 4.26   |
>
>
> |  Algorithm on Cifar10  | Accuracy (%) |  Rounds for 54%   | Time(s) for 54%   |Single round' computation time (s)    |
> |  ----  | ----  | ----  | ----  |----  |
> | FedAvg  | 54.28$\pm$ 0.29 |338 (1.0$\times$) |3283.14 (1.0$\times$) |9.71 |
> | Cluster-based IS  | 54.83$\pm$ 0.02 |323 (1.05$\times$) |3188.54 (1.03$\times$)| 9.87|
> | FedIS  | 55.05$\pm$ 0.27|313 (1.08$\times$)  |3085.05 (1.06$\times$) |9.85|
> | DELTA  | **55.20**$\pm$ 0.26 |**303 (1.12$\times$)** |**2989.98 (1.1$\times$)** |9.86|
>
>
>
> The experiment result shows **DELTA needs fewer communication rounds to reach the same accuracy while each round's computation time of DELTA is comparable with other baselines.**
>
>
>
>
> >### Q: Furthermore, I have the following small concerns:
> > **Q1**. The step-size condition in Theorem $3.1$ for $\eta_L$ and $\eta$ can be simplified to improve readability of this theorem.The authors can write $\eta_L<\min (1 /(8 L K), C)$ where the constant $C$ is obtained from the condition that $1 / 2-\left(10 L^2 / m\right) \sum_{i=1}^m K^2 \eta_L^2\left(A^2+1\right)>0$. Then, the author can state the condition for $\eta$ which is $\eta \leq 1 /\left(\eta_L L\right)$.
> However, the constant $A$ is not clearly stated. Does it depend on the index of the client $i=1,2, \ldots, m$ ? Can the authors check this?\
> >**Q2**. The legend in Figure 5 includes FedSRC-G and FedSRC-D. I believe that they refer to DELTA and FEDIS. Can the authors check and edit the legend?\
> >**Q3**. Fedprox from (Li et al., 2018) does not use variance-reduction techniques to design federated optimization methods. Therefore, the authors should revise this part of the text which is in their contribution section.
>
> Thank you very much for your constructive suggestions. We would like to address all your concerns and have improved our writing following your suggestions in our revised version. Specifically,\
> **A1:** We have modified this in our updated version. As for $A$, it is a constant defined by Assumption 3. Therefore, $A$ does not depend on the client index. We have simplified the condition of $\eta$ and $\eta_L$ as your suggestion: $\eta_L < min(1/(8LK), 1/\sqrt{20(A^2+1)LK})$ , and $\eta \leq 1/(\eta_LL)$.\
> **A2:** Thank you for your kind reminder. We have changed the legend in Fig 5 in our updated version. Your understanding is completely correct that FedSRC-G and FedSRC-D refer to FedIS and DELTA, respectively. \
> **A3:** Thank you again for your helpful comment. We would like to change the expression from " compatible with other variance reduction methods, like Fedprox" to a more accurate sentence," compatible with other advanced optimization methods, like Fedprox."
>
>
> We hope the above responses address your concerns. Please let us know if you have other questions. We’re happy to further answer the questions.

---

> ### Author Response · Authors · 2022-11-11
> **Response to reviewer LBTW (3/4)**
>
> > ### Q3: **Lack of motivation on how to design the optimal client sampling strategies for FedIS and DELTA.**
> > Since FedIS or Algorithm 3 from (Chen et al., 2020) looks similar to DELTA, the proof techniques between these methods should be similar. Hence, I am not sure how the variance from convergence guarantees for FedIS and DELTA is different. Can the authors elaborate on this and perhaps add the motivation before stating theoretical results?
>
> Thanks for your suggestions. We have revised the description of the motivation and added more insights at the beginning of Section 3.3 in the updated version.
>
> Specifically, we would like to elaborate on the difference between FedIS and DELTA and give the motivation for our proposed sampling method as below:
>
> 1. The main analysis difference between FedIS and DELTA is that **FedIS focuses on analyzing the global objective** $\nabla f(x)$, while **DELTA turns to analyze surrogate objective** $\nabla \tilde{f}(x)$. In particular, the analysis of the surrogate objective $\nabla \tilde{f}(x)$ has not been explored in using vanilla IS for FL works.
> 2. The observation on the convergence variance of FedIS is $\Phi = Var(\frac{1}{mp_i^t}\hat{g_i^t}) + V$, and here we use $V$ to represent the variance term that is not related to sampling probability $p_i^t$. Thus, to further reduce the variance the convergence variance by sampling probability, **it is critical to find ways to reduce V**.
> 3. Fortunately, we can decompose the expectation of $\nabla f(x)$ as $E{\|{ \nabla f(x_t) }\|^2} = E{ \|{ \nabla \tilde{f}(x_t) } \|^2} + \chi_t^2$, as shown in equation (5) of our paper,  where $\chi_t^2 = E{\|{\nabla \tilde{f}_{S_t}(x_t) - \nabla f(x_t)}\|^2}$ corresponds to the variance $Var(\frac{1}{mp_i^t}\hat{g_i^t})$ in the convergence rate of FedIS. As we discussed, it is crucial to further reduce the convergence variance beyond $Var(\frac{1}{mp_i^t}\hat{g_i^t})$. Thus, now **it is critical to find ways to reduce the convergence variance of $\tilde{f}(x)$**.
> 4. What's more interesting is that, as we show in Lemma B.2, **the convergence of $\chi_t^2$ can be bounded by the convergence of $E\|\nabla \tilde{f}(x_t)\|^2$**. Thus, minimizing the convergence variance of $E\|\nabla \tilde{f}(x_t)\|^2$ can reduce both two terms in $E{\|{ \nabla f(x_t) }\|^2} = E{ \|{ \nabla \tilde{f}(x_t) } \|^2} + \chi_t^2$ simultaneously.
> 5. Then we focus on the analysis of $\tilde{f}(x)$ and get a different convergence variance $\tilde{\Phi}$ from $\Phi$ of FedIS. Specifically, $\tilde{\Phi}$ is related to sampling probability, thus **by minimizing the new variance $\tilde{\Phi}$, we get a different sampling probability from FedIS.**
>
>
>
> >### Q4: **Numerical evaluations against other unclear existing sampling strategies.** In the experiments, the authors compared their sampling strategies against others, e.g. the power of choice, norm, and heterogeneity. However, these strategies are not clearly stated (i.e. what is $p_i^t$ )? Are they existing sampling strategies, e.g., from (Chen et al., 2020)?
> >
> >Sorry for the statement unclarity. In the revised version, **we have added a detailed description of the sampling strategies of all these methods** in Appendix G.1. In general:
>
> - The power-of-choice sampling method is proposed by [1]. The sampling strategy of Power-of-Choice is that the server first samples a large subset of clients randomly from all clients (in our experiment, the number of the large subset is 20) and then chooses 10 of the 20 clients that have the largest loss as the selected clients.
> - The norm method refers to our FedIS. Our FedIS is a little different from [2]. In [2], clients decide whether to participate in training while the number of participated clients in expectation is equal to $n = |S_t|$. While in our FedIS, the server selects a fixed number $n = |S_t|$ to participate in each round.
> - The heterogeneity method refers to DELTA.
>
> We have fixed these notations that caused misunderstandings in our revised version.
>
> [1]Cho Y J, Wang J, Joshi G. Client selection in federated learning: Convergence analysis and power-of-choice selection strategies[J]. arXiv preprint arXiv:2010.01243, 2020.\
> [2]Chen W, Horvath S, Richtarik P. Optimal client sampling for federated learning[J]. arXiv preprint arXiv:2010.13723, 2020.

---

> ### Author Response · Authors · 2022-11-11
> **Response to reviewer LBTW (2/4)**
>
>
>
> > ### Q2: **Redundant assumptions for deriving the convergence results for FedIS and DELTA.**
> > Assumptions 2 and 3 are commonly used for deriving convergence guarantees for federated methods. This is because Assumption 2 states how different the local stochastic gradient and the local full gradient is, and Assumption 3 implies how different the local full gradient and the whole full gradient is. However, Assumption 4 seems to be redundant to Assumption 2, since both of them impose the property of local stochastic gradients. Therefore, the stated convergence results for FedIS and DELTA seem too restricted.
>
> Thanks for pointing out that, and we agree the convergence can be derived without Assumption 4.\
> *We would like to clarify that in our original submission, only FedIS depends on Assumption 4 while DELTA does not.* **In the revised version, we have provided the convergence rate of FedIS without Assumption 4 in Appendix C.1**
>
> - The reason that we use a stronger condition $G$ (Assumption 4) here is that we want to intuitively show that the convergence rate is directly related to the gradient norm, as Assumption 4 is a straightforward gradient norm bound. \
>   In fact, as we show that one can relax Assumption 4 directly to Assumption 3 and use $\sigma_G$ to replace $G$ in the convergence rate result without other changes.
> - The convergence rate of FedIS without Assumption 4 is as follows:
>   $$\min \limits_{t\in[T]} E\|\nabla f(x_t)\|^2\leq \mathcal{O}\left(\frac{ f^0-f^*}{\sqrt{nKT}}\right) + \underbrace{\mathcal{O}\left(\frac{\sigma_L^2}{\sqrt{nKT}}\right) +  \mathcal{O}\left(\frac{M^2}{T}\right) + \mathcal{O}\left(\frac{K\sigma_G^2}{\sqrt{nKT}} \right)}_{\text{order of} \ \Phi}$$
>
> - The detailed convergence rate analysis without Assumption 4 is as below:
>
> Assumption 4 is used to bound the variance term in the final convergence result, while not used in the previous proof steps. Thus, we can focus on discussing the variance term.
> In the final convergence result, the variance term is
> $$V\left(\frac{1}{m p_{i}^{t}} \hat{g_{i}^{t}}\right)=\frac{1}{m^2}\mathbb{E}\|\sum_{i=1}^m\sum_{k=1}^K\nabla F_i(x_{t,k},\xi_{k,t})\|^2\leq \frac{1}{m}\sum_{i=1}^mK\sum_{k=1}^K\mathbb{E}\|\nabla F_i(x_{t,k},\xi_{k,t})\|^2 \leq K^2G^2,$$ where the last inequality comes from Assumption 4.
>
> For relaxing Assumption 4 to the Assumption 3 ($\mathbb{E}\|\nabla F_i(x)\|^2 \leq (A^2+1)\|\nabla f(x)\|^2 + \sigma_G^2$), we can rewrite the variance in the convergence as:
> $$V\left(\frac{1}{m p_{i}^{t}} \hat{g_{i}^{t}}\right)=\frac{1}{m^2}\mathbb{E}\|\sum_{i=1}^m\sum_{k=1}^K\nabla F_i(x_{t,k},\xi_{k,t})\|^2\leq \frac{1}{m}\sum_{i=1}^mK\sum_{k=1}^K\mathbb{E}\|\nabla F_i(x_{t,k},\xi_{k,t})\|^2\leq K^2\sigma_G^2+K^2(A^2+1)\|\nabla f(x_t)\|^2.$$
> Thus in the derivation, $G^2$ can be directly substituted by $\sigma_G^2$, only the condition of $\eta_L$ changes from
> $$\frac{1}{2}-10L^2 K^2(A^2+1)\eta_L^2 > 0$$
> to
> $$\frac{1}{2}-10L^2K^2(A^2+1)\eta_L^2-\frac{L\eta K(A^2+1)}{2n}\eta_L>0.$$
> As for $\eta_L$, one can still guarantee that there exists a constant for $\eta_L$ to satisfy the new inequality according to the properties of quadratic functions. The proof and discussion are in Appendix C of our revised version.

---

> ### Author Response · Authors · 2022-11-11
> **Response to reviewer LBTW (1/4)**
>
> Dear reviewer LBTW, thank you very much for taking the time to review our paper. We really appreciate all your comments and suggestions. Please find my itemized responses to your raised questions below:
>
> > ### Q1: **Limitations of using the optimal sampling for DELTA.**
> > From Corollary 3.4, the optimal sampling depends on $\left\|\nabla F_i(x)-\nabla f(x)\right\|$. This strategy cannot be implemented in practice, mainly because the gradient of the whole objective $\nabla f(x)$ cannot be accessed usually for problems over huge amounts of data. Therefore, the authors should state how this strategy is modified to be able to implement it in their experiments.
>
> We are very grateful and thankful that you have noticed that a practical algorithm is an important basis for effective application.
>
> **We would like to clarify that the detailed practical implementation method of both FedIS and DELTA are presented in Section 4 in our original submission.**
>
> - We restate the core idea of our practical algorithm here: We use the last participated round's gradient to approximate the gradient of the current round. As for the gradient of the global objective $\nabla f(x)$, in our paper, we write, "Specifically, we use the average of the latest participated clients' gradients to approximate the true gradient of the global model for DELTA.”
>
> Moreover, we have provided the convergence analysis for the practical sampling probability in Appendix F of our revised version. In particular:
>
> - For FedIS, the convergence rate of the practical algorithm is
>   $$\min \limits_{t\in[T]} E\|\nabla f(x_t)\|^2\leq \mathcal{O}\left(\frac{ f^0-f^*}{\sqrt{nKT}}\right) \!+\! \underbrace{\mathcal{O}\left(\frac{\sigma_L^2}{\sqrt{nKT}}\right) \!+ \! \mathcal{O}\left(\frac{M^2}{T}\right) \!+ \! \mathcal{O}\left(\frac{KU^2\sigma_G^2}{\sqrt{nKT}} \right)}_{\text{order of} \ \Phi} $$
> - For DELTA, the convergence rate of the practical algorithm is
>   $$\min_{t \in[T]} E\|\nabla f(x_t)\|^2 \leq  \mathcal{O}\left(\frac{f^0-f^*}{\sqrt{nKT}}\right) +
>         \underbrace{ \mathcal{O}\left(\frac{\tilde{U}^2\sigma_{L,s}^2}{\sqrt{nKT}}\right) + \mathcal{O}\left(\frac{\tilde{U}^2\sigma_{L,s}^2  + 4K\tilde{U}^2\zeta_{G,s}^2}{KT}\right)}_{\text{order of } \tilde{\Phi}}$$
>   $U$ and $\tilde{U}$ are two constant bounds, which means the changing ratio of gradient norm.

---

> ### Author Response · Authors · 2022-11-15
> **Look forward to your feedback!**
>
> Dear reviewer LBTW,
>
> We thank you again for your constructive comments and helpful suggestions. Since the discussion period is ending soon, we would like to kindly remind you to check our responses and the revised version of our paper. We hope they can address your concerns and look forward to your feedback.

---

> ### Author Response · Authors · 2022-12-10
> **To reviewer LBTW**
>
> Dear reviewer LBTW,
>
> Since the discussion time is ending soon, we would like to kindly remind you to check our responses and the revised version of the paper. We hope they can address your concerns, and we would be happy to know if you could reconsider your score.

---

> > ### Comment · Reviewer_LBTW · 2022-12-12
> > **To the author**
> >
> > Thank you for addressing my concerns. Since they are all addressed, I can raise my score.
> >
> > I still agree with other reviewers that the writing of this paper should be improved. It is important to have contributions of the paper that are focused on why and how adaptive client sampling strategies in this paper perform well, compared to existing strategies. So far, this motivation is a bit unclear, except for the fact that these strategies work well in practical experiments on training large-scale neural network models.

---

> > > ### Author Response · Authors · 2022-12-12
> > > **Reply to reviewer LBTW**
> > >
> > > Thank you for raising our score.
> > >
> > > We agree that clear motivation is very important, so we have provided a toy example (Figure 4) and a discussion (Section 3.5) to show why and how DELTA performs well compared to other methods. \
> > > We will make the motivation even clearer and polish the writing in the next version.
> > >
> > > Thank you again for your constructive comments that help improve our work a lot.

---

### Official Review · Reviewer_Yzew · 2022-11-04

**Confidence:** 4
**Correctness:** 2
**Technical Novelty And Significance:** 3
**Empirical Novelty And Significance:** 2
**Recommendation:** 3

**Clarity, Quality, Novelty And Reproducibility:**

The quality of this paper should be improved. The authors should polish the writing with rigorous statements and check the correctness of the theoretical results.

**Strength And Weaknesses:**

Strength
- The idea of sampling from diverse gradient groups is a novel idea and seems to be promising.
- The authors make the proposed sampling algorithm to be unbiased.

Weakness
- The writing of this paper should be improved. There are many vague statements. For example, in the introduction, the authors explain why previous works are not good. But all the explanations are just intuitions or conjectures. They cannot be used to support the observations in Figure 2. We do not know whether these conjectures are true or not (e.g. whether cluster-IS really select clients with small gradients and whether this is the core reason causing slow convergence).
- Also, many mathematical expressions are wrong. For example, in equation 2 and 6, the function $f$ should also have some subscripts because its form changes when we sample different clients. Also, in equation (5), the authors define $E|\nabla f(x)|^2 = E|\nabla \tilde{f}(x)|^2 + \chi^2$. However, in equation (15), they wrote $E|\nabla \tilde{f}(x)|^2 = E|\nabla f(x)|^2 + \chi^2$. It is obvious these two equations are conflict with each other.
- The proof may not be correct. I didn't find any proof details for Theorem D.2, which I suspect is wrong due to the above mistakes.
- Remark 3.2 (3) is not accurate. "Chen (2020) ... with additional gradient similarity bound". This sounds like this paper did not do this and remove this assumption. But in fact, the authors define it in Assumption 3 and also use it.

**Summary Of The Paper:**

This paper proposes a new method to improve previous (cluster-based) important client sampling methods in federated learning. The new method is motivated by the insight that it would be beneficial to select clients from diverse groups. Convergence analysis are also provided and the authors claim they improve over existing ones. At last, experiments on FEMNIST and CIFAR-10 are provided to validate the performance.

**Summary Of The Review:**

I suspect that this paper has mistakes in the proof. The theoretical results may not be valid.

---

> ### Author Response · Authors · 2022-11-11
> **Response to reviewer Yzew (2/2)**
>
>
> > ### Q3: In equation (2) and (6), the function $f(x)$ should also have some subscripts because its form changes when we sample different clients.
> >
> Thanks for your constructive suggestion, *we have added subscript $S_t$ in equations (2) and (6) as $f_{S_t}(x_t)$ and $\tilde{f}_{S_t}(x_t)$*, where the subscript $S_t$ shows that $f$ and $\tilde{f}$ changes with different client sets.
>
> The reason that we used $f(x_t)= \frac{1}{n}\sum_{i\in S_t} F_i(x_t)$ in equation (2) and $\tilde{f}(x_t)=\frac{1}{n}\sum_{i\in S_t}\frac{1}{mp_i^t} F_i(x_t)$ in equation (6) in the original submission is that the subscript $t$ can show the variation of function $f(x_t)$ and $\tilde{f}(x_t)$ with round $t$. \
>  That is, different round $t$ leads to a different participated client subset $S_t$ and different client participation leads to the change of $f(x_t)$ and $\tilde{f}(x_t)$.
>
>
>
>
>
> > ### Q4:  The proof may not be correct. I didn't find any proof details for Theorem D.2, which I suspect is wrong due to the above mistakes.
> >
> Thanks for your comment. *We would like to clarify that the proof details for Theorem D.2 were provided in Appendix D.2 and D.3 in our original submission.*
>
> Sorry for the misunderstanding caused by our abuse of $f(x)$ in the proof of Theorem D.2 in our original submission; and thank you very much for reminding us that the description given before Appendix D.2 of the original submission, i.e.,  "To simplify the expression, in the following proof section, we use $\nabla f(x_t)$ instead of $\nabla \tilde{f}(x_t)$...", is not appropriate and also not clear enough that one may miss this reminder about the reuse of $f(x)$.
>
> In the revised version, we have revised $f(x)$ as $\tilde{f}(x)$ in Appendix D.2 and D.3 -- the proof of Theorem D.2.
>
>
>
>
>
>
>
>
> > ### Q5: Remark $3.2$ (3) is not accurate. "Chen (2020) ... with additional gradient similarity bound". This sounds like this paper did not do this and remove this assumption. But in fact, the authors define it in Assumption 3 and also use it.
> >
> Thank you for pointing out the inaccurate statements. Inspired by your comment, we have revised the expression from "Chen et al. [1] provides the convergence rate of nonconvex FL under the additional assumption of gradient similarity bound." to "Chen et al. [1] provides the convergence rate of nonconvex FL **under a stronger assumption**."
>
> The Assumption 9 in [1] is "$\sum_{i=1}^n w_i\left\|\nabla f_i(x)-\nabla f(x)\right\|^2 \leq \rho,$ for some $\rho \geq 0$". We agree it is a similar assumption to our assumption 3, but Assumption 9 in [1] is a more restrictive assumption compared with our Assumption 3 ($E{ \|{ \nabla F_i(x) }\|^2 } \leq (A^2+1)\|\nabla f(x)\|^2 + \sigma_G^2$), as proved in [2].
>
> [1]Chen W, Horvath S, Richtarik P. Optimal client sampling for federated learning[J]. arXiv preprint arXiv:2010.13723, 2020.\
> [2]Koloskova A, Loizou N, Boreiri S, et al. A unified theory of decentralized sgd with changing topology and local updates[C]//International Conference on Machine Learning. PMLR, 2020: 5381-5393.
>
> We hope the above responses address your concerns. Please let us know if you have other questions. We are happy to further answer the questions.

---

> ### Author Response · Authors · 2022-11-11
> **Response to reviewer Yzew (1/2)**
>
> Dear reviewer Yzew, we would like to thank you for your time spent reviewing our paper and for providing constructive comments. Please kindly find our responses to your raised questions below:
>
> > ### Q1: The writing of this paper should be improved. There are many vague statements. For example, in the introduction, the authors explain why previous works are not good. But all the explanations are just intuitions or conjectures. They cannot be used to support the observations in Figure 2. We do not know whether these conjectures are true or not (e.g. whether cluster-IS really select clients with small gradients and whether this is the core reason causing slow convergence).
>
> Thank you for pointing out the writing issues. Motivated by your comment, we conducted experiments to support our statement. The additional experimental results are in Appendix A of the revised version.
>
> For the experiment setting, we apply a logistic regression model on the non-iid MNIST. 10 clients are selected from 200 clients to participate in training in each round. More details can be found in Appendix A.
>
>
>
> We have provided the **experimental results** of:
>
> -  The gradient comparison of IS and cluster-based IS.
> -  The accuracy and loss performance comparison between vanilla cluster-based IS and cluster-based IS without small gradients.
>
> The experimental results can be found in Appendix A of our revised version. Our results show:
>
> - **Cluster-based IS samples clients with small gradients.** Specifically, both the average gradient and minimal gradient norm of cluster-based IS are smaller than that of IS after about a half rounds.
> - **Cluster-based IS selecting clients from the cluster with small gradients will slow convergence.** Specifically, for cluster-based IS, replacing clients with small gradients with clients with large gradients improves both the loss and accuracy performance.
>
> We also present the results of the experiments here:\
> The average gradient comparison between cluster-based IS and IS in [The link](https://i.postimg.cc/PJD6L0C9/gradient-comp.jpg).\
> The minimal gradient norm comparison between cluster-based IS and IS in [The link](https://i.postimg.cc/6pZ9xrN0/gradient-comp-min.jpg).\
> The experiments results of small gradients causing cluster-based IS to converge slowly: Accuracy:[The link](https://i.postimg.cc/SRLWP2xN/acc-com.jpg); Loss:[The link](https://i.postimg.cc/BZVVQcV2/loss-com.png).
>
> As suggested by the reviewer, **we have rewritten the statements on the cluster-based IS in the revised version to avoid ambiguity.** Specifically, we replace our original expression from 1) to 2).
>
> 1) "cluster-based IS suffers from a slow convergence since it keeps sampling clients from small gradient clusters."
> 2) "vanilla cluster-based IS does not work well because the high-dimensional gradient is too complicated to be a good clustering feature and can bring about poor clustering results, as pointed out by [1]. In addition, clustering is known to be susceptible to biased performance if the samples are chosen from a group that is clustered based on a biased opinion, as shown in [2][3]."
>
>
> [1]Shen G, Gao D, Song D X, et al. Fast Heterogeneous Federated Learning with Hybrid Client Selection[J]. arXiv preprint arXiv:2208.05135, 2022.\
> [2]Sharma G. Pros and cons of different sampling techniques[J]. International journal of applied research, 2017, 3(7): 749-752.\
> [3]Thompson S K. Adaptive cluster sampling[J]. Journal of the American Statistical Association, 1990, 85(412): 1050-1059.
>
>
> > ### Q2: In equation (5), the authors define $E|\nabla f(x)|^2=E|\nabla \tilde{f}(x)|^2+\chi^2$. However, in equation (15), they wrote $E|\nabla \tilde{f}(x)|^2=E|\nabla f(x)|^2+\chi^2$. It is obvious these two equations are conflict with each other.
>
> *For equation (15), the reviewer comments that $E\|\nabla \tilde{f}(x_t)\|^2=  E\|\nabla f(x_t)\|^2+ \chi^2$. But in our paper, equation (15) is $E\|\nabla \tilde{f}(x_t)\|^2 = \|\nabla f(x_t)\|^2 + \chi^2$,  in which there is no expectation over $\nabla f(x_t)$. Thus, we argue that there is no conflict between equation (5) and equation (15).*
>
> We thank the reviewer for reminding us to write equation (15) in a more readable way. To improve the readability, we have further enriched the derivation of equation (15) in the revised version.

---

> ### Author Response · Authors · 2022-11-15
> **Look forward to your feedback!**
>
> Dear reviewer Yzew,
>
> We thank you again for your constructive comments and helpful suggestions.
> Since the discussion period is ending soon, we would like to kindly remind you to check our responses and the revised version of our paper. We hope they can address your concerns and look forward to your feedback.

---

> ### Author Response · Authors · 2022-12-10
> **To reviewer Yzew**
>
> Dear reviewer Yzew,
>
> Since the discussion time is ending soon, we would like to kindly remind you to check our responses and the revised version of the paper. We hope they can address your concerns, and we would be happy to know if you could reconsider your score.

---

### Author Response · Authors · 2022-11-11
**Response to all reviewers:**

We thank all reviewers for their time and efforts in reviewing our paper. **We revised our paper based on the suggestions of all four reviewers using blue lines.** In detail:
- We have provided the additional convergence analysis without Assumption 4 for FedIS in Appendix C. We would like to clarify that the analysis for DELTA does not rely on Assumption 4 in our original submission.
- We have provided the theoretical analysis of the impacts of the practical algorithm and given its convergence rate in Appendix F.
- We have added experimental results to compare the wall-clock convergence time between different sampling methods, including FedAvg, cluster-based sampling, FedIS, and DELTA in Table 2.
- We have added experiments to enhance our observation of the drawbacks of vanilla cluster-based IS in Appendix A.
- We have added a detailed description of the sampling process for all the baselines, i.e., FedAvg, Power-of-Choice, and FedIS, in Appendix G.1.
- We have addressed the abuse of notation $f$ by replacing $f$ with $\tilde{f}$ in the proof of Theorem D.2 in Appendix D.2 and Appendix D.3.
- We have fixed the typos pointed out by the reviewers and revised the paper to improve clarity and readability.

---

### Decision · Program_Chairs · 2023-01-20

**Decision:**

Reject

**Justification For Why Not Higher Score:**

The improvements over the baseline are based on an assumption that full gradient is available in each communication round, which is exactly not the case in the partial participation setting the authors study. So, the theoretical improvements are meaningless. Moreover, stronger baselines exist (with better rates than what the authors obtain), and these have not been compared against. These are fatal issues in my view and the paper can't be accepted.

**Justification For Why Not Lower Score:**

N/A

**Metareview: Summary, Strengths And Weaknesses:**

This paper proposes a new method to improve previous (cluster-based) important client sampling methods in federated learning. The new method is motivated by the insight that it would be beneficial to select clients from diverse groups. Convergence analysis are also provided and the authors claim they improve over existing ones. At last, experiments on FEMNIST and CIFAR-10 are provided to validate the performance.

Key issue with the work: the optimal sampling depends in information not available in the partial participation setting: "Corollary 3.4 says that the optimal sampling depends on the full gradient, which is not available." Thus, the claimed theoretical benefits are valid for a method that can't be implemented. The statements related to theoretical superiority to the mentioned works are thus not fully justified; this key limitation is not mentioned in the right places. Moreover, the new rates are inferior to some recent advances/results in the field, example: https://arxiv.org/abs/2205.15580 and the references therein. These works were not mentioned, and this is a serious omission as they deal with the same problem: improving results for partial participation in the nonconvex regime. Assumption 3 is not justified beyond stating it is commonly used. This assumption (as well as other commonly used assumptions) was recently criticized in https://arxiv.org/abs/2002.03329 .

Other than this, I appreciate that the authors responded to the reviewers in detail, and that they adjusted their manuscript based on some of the criticism raised.

However, I believe that the paper cannot be accepted in its current form due to the above issues which I think are of a significant nature, especially since ICLR is a top venue in the field.

AC


**Summary Of Ac-Reviewer Meeting:**

no meeting was held